# Single-cell morphological and topological atlas reveals the ecosystem diversity of human breast cancer

Shen Zhao[1,7], De-Pin Chen [2,7], Tong Fu[1,7], Jing-Cheng Yang[1,3,7], Ding Ma [1], Xiu-Zhi Zhu[1], Xiang-Xue Wang[2], Yi-Ping Jiao[2], Xi Jin[1], Yi Xiao[1], Wen-Xuan Xiao[1], Hu-Yunlong Zhang[1], Hong Lv[4], Anant Madabhushi [5,6], Wen-Tao Yang [4] ✉, Yi-Zhou Jiang[1] ✉, Jun Xu [2] ✉ & Zhi-Ming Shao [1] ✉

Digital pathology allows computerized analysis of tumor ecosystem using whole slide images (WSIs). Here, we present single-cell morphological and topological profiling (sc-MTOP) to characterize tumor ecosystem by extracting the features of nuclear morphology and intercellular spatial relationship for individual cells. We construct a single-cell atlas comprising 410 million cells from 637 breast cancer WSIs and dissect the phenotypic diversity within tumor, inflammatory and stroma cells respectively. Spatially-resolved analysis identifies recurrent micro-ecological modules representing locoregional multicellular structures and reveals four breast cancer ecotypes correlating with distinct molecular features and patient prognosis. Further analysis with multiomics data uncovers clinically relevant ecosystem features. High abundance of locally-aggregated inflammatory cells indicates immune-activated tumor microenvironment and favorable immunotherapy response in triple-negative breast cancers. Morphological intratumor heterogeneity of tumor nuclei correlates with cell cycle pathway activation and CDK inhibitors responsiveness in hormone receptor-positive cases. sc-MTOP enables using WSIs to characterize tumor ecosystems at the single-cell level.

Tumors are increasingly recognized as an "ecosystem" composed of tumor, immune and stroma cells together with the complex cell-to-cell interactions[1]. The cancer biological behavior, prognosis and treatment response depend not only on the traits of tumor cells but also on other cell components and their interplay with the tumor cells[2,3]. For instance, tumor-infiltrating lymphocytes (TILs) play a key role in anti-tumor immune response and is associated with the prognosis of breast cancer patients and the tumor response to chemotherapy and immunotherapy[4,5]. Cancer-associated fibroblasts, which are the main cell type in cancer stroma, can regulate cancer progression via heterotypic interactions with the nearby tumor cells and contribute to immunosuppressive microenvironment and drug resistance[6,7]. Dissecting the tumor ecosystem composition and the relationship between different cell types is of great significance for understanding the cancer biology and improving the cancer management.

[1]Key Laboratory of Breast Cancer in Shanghai, Department of Breast Surgery, Fudan University Shanghai Cancer Center, Shanghai, China. [2]Institute for Artificial Intelligence in Medicine, School of Artificial Intelligence, Nanjing University of Information Science and Technology, Nanjing, China. [3]Greater Bay Area Institute of Precision Medicine, Guangzhou, China. [4]Department of Pathology, Fudan University Shanghai Cancer Center, Shanghai, China. [5]Wallace H Coulter Department of Biomedical Engineering, Georgia Institute of Technology and Emory University, Atlanta, GA, USA. [6]Atlanta Veterans Affairs Medical Center, Atlanta, GA, USA. [7]These authors contributed equally: Shen Zhao, De-Pin Chen, Tong Fu, Jing-Cheng Yang. ✉e-mail: yangwt2000@163.com; yizhoujiang@fudan.edu.cn; xujung@gmail.com; zhimingshao@fudan.edu.cn

Nowadays, histopathology remains the "gold standard" for cancer diagnosis. In addition to diagnostic information, the hematoxylin and eosin (H&E)-stained pathological slides provide a wealth of information on tumor ecosystem including cell composition, morphology and spatial organization. The advent of digital pathology and computerized image analysis allows high-throughput characterization of tumor ecosystem from pathological whole slide images (WSIs). Previous studies developed prognostic or predictive models based on the cell morphology or spatial arrangement features in several cancer types[8-14]. These studies identified certain features that were associated with specific clinical outcomes (usually the prognosis or therapy response) for model development, but lacked the comprehensive depiction of the tumor ecosystem by integrating the multidimensional features. Wang et al. and Yoo et al. performed unsupervised clustering based on the image features and classified the hepatocellular carcinoma and colorectal cancers into "imaging subtypes"[15,16]. However, both studies focused solely on the features of tumor cells and

lymphocytes and ignored those of stroma cells, which are important components in the tumor ecosystem. In addition, the above studies generally integrated the cell-level features into patient-level for modeling or clustering. A comprehensive atlas characterizing the phenotypic diversity within each cell type in the tumor ecosystem on the single-cell level remains lacking.

In this work, we present a single-cell morphological and topological profiling (sc-MTOP) framework to characterize the tumor ecosystem at the single-cell resolution and use this approach to investigate the phenotypic diversity of human breast cancer ecosystem and its clinical relevance (Fig. 1a). The framework first employs Hover-Net to segment the nuclei on WSIs and predict their cell types[17]. Then, for individual cells, nuclear morphological and texture features are extracted based on the nuclear contour. A graph-based method is devised to model the intercellular spatial relationship and a series of topological features are extracted (Fig. 1b). We apply the sc-MTOP framework to 637 breast cancer WSIs covering all immunohistochemistry (IHC) subtypes and

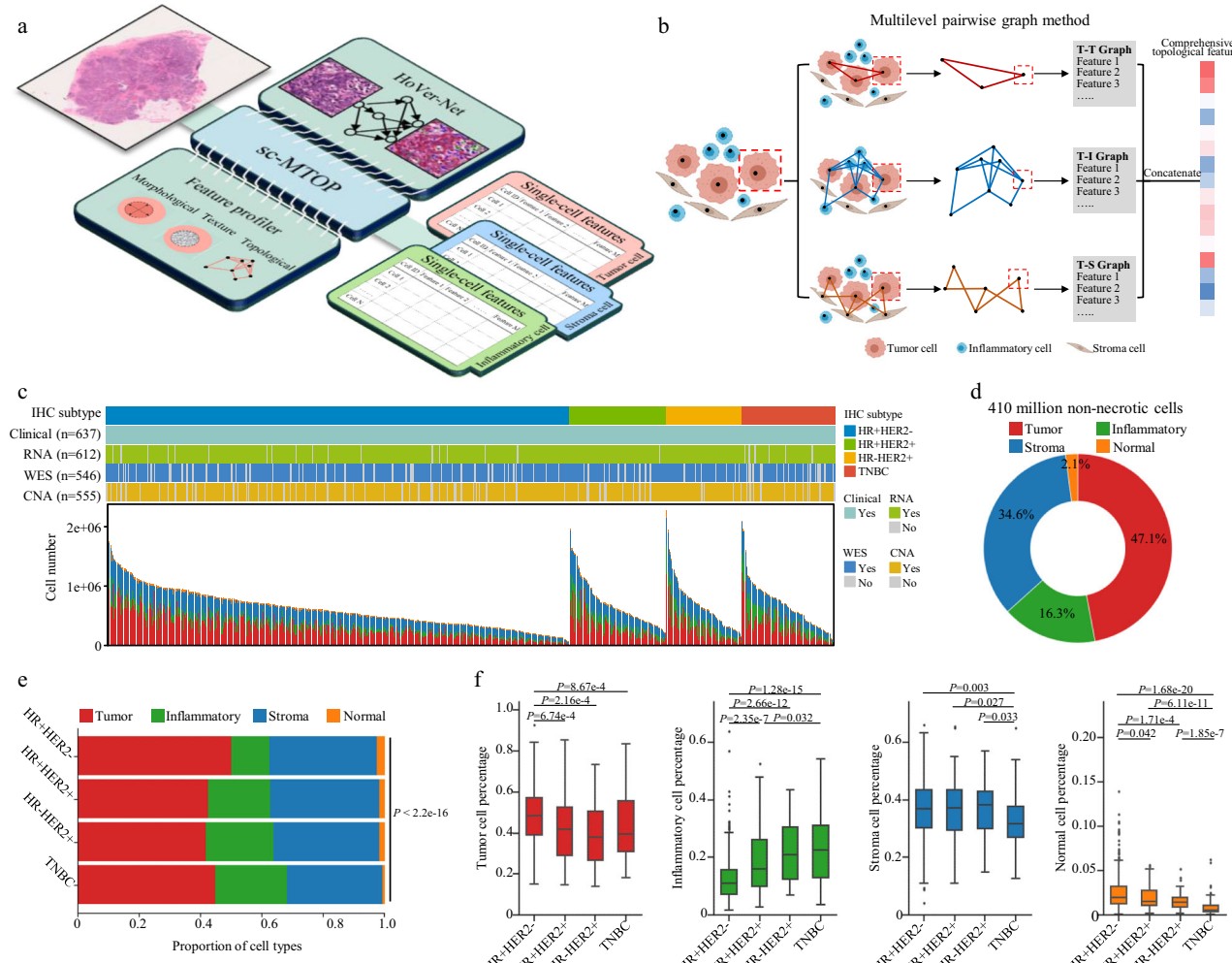

**Fig. 1 | Single-cell morphological and topological profiling and the generated dataset. a** Schematic diagram of single-cell morphological and topological profiling (sc-MTOP). It first employs Hover-Net to implement nuclear segmentation and classification on WSIs. Then, for individual cells, nuclear morphological, texture and topological features are extracted based on the nuclear contour and the intercellular spatial relationship. Created with BioRender.com. **b** Schematic diagram of multilevel pairwise graph method to model the spatial relationship of cells within the same and across cell types (taking a tumor cell as an example). T, I and S denote tumor, inflammatory and stroma cell respectively. Created with BioRender.com. **c** Cohort information about the cell composition of whole slide images and patients' clinical and multiomics data. Each column represents a patient. **d** Cell composition of the entire sc-MTOP dataset. **e** Cell composition according to the

breast cancer IHC subtypes (HR+HER2-: *n* = 405; HR+HER2+: *n* = 85; HR-HER2+: *n* = 66; TNBC: *n* = 81). *P* value is calculated using the chi-square test. **f** Boxplots of the percentage of tumor, inflammatory, stroma and normal cells of WSIs according to the breast cancer IHC subtypes (HR+HER2-: *n* = 405; HR+HER2+: *n* = 85; HR-HER2+: *n* = 66; TNBC: *n* = 81). The center lines of boxplots indicate the median values; box limits show upper and lower quartiles; whiskers extend from box limits to the farthest data point within 1.5 × interquartile range; points beyond whiskers are outliers. *P* values are calculated using the two-sided Mann-Whitney U test with false discovery rate-correction for multiple testing. sc-MTOP single-cell morphological and topological profiling; IHC immunohistochemistry; WES whole-exome sequencing; CNA copy number alteration.

establish a spatially resolved single-cell dataset containing 410 million cells. Based on the dataset, we depict a single-cell morphological and topological landscape dissecting the phenotypic diversity of breast cancer ecosystem and uncover ecosystem features with clinical relevance. Finally, to facilitate data sharing and the use of sc-MTOP framework by other researchers, an online platform is developed which provides access to our dataset and can realize real-time analysis of user-uploaded WSIs.

## Results

### sc-MTOP and the generated dataset

We propose the sc-MTOP framework to characterize the tumor ecosystem on the single-cell level based on digital pathology. The framework takes a H&E-stained pathological WSI at ×40 magnification as input. It first employs Hover-Net for simultaneous nuclear segmentation and classification. Then, feature profiling is performed for individual tumor, inflammatory and stroma cells, which are the main cellular components of tumor ecosystem[18]. The features include three categories: a) morphological features describing the nuclear shape and contour; b) texture features characterizing the local pixel distribution patterns within the nuclear contour; c) topological features characterizing the intercellular relationships based on graph algorithm. In topological feature extraction, to comprehensively characterize the spatial relationships across different cell types, we propose a multilevel pairwise method for graph construction. In brief, for each pair of cell types, a graph is constructed to model the cellular relationship. Thus, each tumor, inflammatory and stroma cell appears in three types of graph (e.g. each tumor cell appears in tumor-tumor, tumor-inflammatory and tumor-stroma graphs). For each cell, the topological features are computed separately based on each graph that it appears and are concatenated to constitute its comprehensive topological features (Fig. 1a, b and Methods). Our framework highlights the feature profiling of individual cells, which allows analysis of the phenotypic diversity among cells and its potential clinical relevance. The multilevel pairwise graph method enables comprehensive dissection of the spatial relationships between different cell types.

To depict a panoramic landscape of human breast cancer ecosystem and explore the clinical relevance of ecosystem features, we applied sc-MTOP to 637 WSIs of a retrospectively collected cohort with multiomics profiling data. This cohort included 637 consecutive breast cancer patients who received standard treatment at Fudan University Shanghai Cancer Center (FUSCC) from January 1, 2013 to December 31, 2014 (Methods). The samples covered all breast cancer IHC subtypes including 405 hormone receptor-positive HER2 negative (HR+HER2-), 85 HR+HER2+, 66 HR-HER2+ and 81 triple-negative breast cancers (TNBCs) (Fig. 1c and Supplementary Table S1). A total of 410,575,052 non-necrotic cells were identified, including 193,216,681 (47.1%) tumor cells, 66,853,146 (16.3%) inflammatory cells, 141,896,644 stroma cells (34.6%) and 8,608,581 normal breast gland cells (2.1%) (Fig. 1d). The average cell number (±standard variation) per sample were $303,322 \pm 207,555$ for tumor cells, $104,950 \pm 104,107$ for inflammatory cells, $222,758 \pm 124,843$ for stoma cells and $13514 \pm 14786$ for normal breast gland cells.

The cell classification results were validated through the comparison with co-detection by indexing (CODEX), correlation analysis and gene set enrichment analysis (GSEA). We performed successive CODEX and H&E-staining for two cases (Methods, Supplementary Fig. S1a) and analyzed the consistency between the CODEX and Hover-Net cell classification results. We marked 25 matched regions $(100 \times 100 \, \mu m)$ on the CODEX images and WSIs and quantified the number of tumor, inflammatory and stroma cells based on the CODEX images and Hover-Net cell classification respectively. It was found that the estimated cell number of tumor, stroma and inflammatory cells showed high correlation between the CODEX images and Hover-Net cell classification (Spearman correlation coefficient: 0.99 for tumor cells, 0.96 for inflammatory cells and 0.95 for stroma cells)

(Supplementary Fig. S1b–d). Second, we calculated the cell percentages of tumor, inflammatory and stroma cells and correlated them with the estimated cell abundance based on the multiomics data (Methods). Tumor cell percentage was positively associated with tumor purity estimated by the ASCAT algorithm[19]. Inflammatory cell percentage showed positive correlation with TILs, CD3E mRNA expression, immune signature scores and the estimated abundance of CD8+ T cells, while the stroma cell percentage was positively correlated with the stroma signature scores and the estimated abundance of fibroblasts, endothelial cells and adipocytes (Supplementary Fig. S2a). Third, we performed GSEA according to the samples' cell composition. Immune-related gene sets were enriched in samples with high inflammatory cell percentage, while gene sets of extracellular matrix and cell-matrix adhesions were enriched in samples with high stroma cell percentage (Supplementary Fig. S2b, c). These results supported the validity of Hover-Net-based cell classification in breast cancer WSIs.

We next investigated the difference in the cell composition among the IHC subtypes of breast cancer. HR+HER2- breast cancers comprised a higher percentage of tumor cells and a lower percentage of inflammatory cells. Both HR-HER2+ and TNBCs had a higher percentage of inflammatory cells. TNBCs were characterized by a lower percentage of stroma cells. In addition, the normal cell percentage was highest in HR+HER2- and was lowest in TNBCs (Fig. 1e, f). As for clinicopathological features, high tumor grade was associated with high inflammatory cell percentage and low percentage of stroma, normal and tumor cells. High T category samples comprised a low percentage of normal cells. N category showed no significant correlation with the samples' cell composition (Supplementary Fig. S2d–f). No statistically significant difference in recurrence-free survival (RFS) was observed between the groups stratified by the percentage of the four cell types (Supplementary Fig. S2g–j).

### Single-cell atlas of inflammatory cells

Based on the sc-MTOP data, we aimed to dissect the diversity of breast cancer ecosystem at multiple levels. First of all, at the single-cell level, we aimed to characterize the phenotypic diversity of inflammatory, tumor and stroma cells respectively through Leiden clustering based on the nuclear features[20].

Inflammatory cells were classified into 10 clusters (named INF0-INF9) with distinct topological features (Fig. 2a). Four clusters were locally aggregated inflammatory cells characterized by high I-I_Degrees, positive I-I_ClusteringCoefficient, low I-I_MinEdgeLength and low I-I_MeanEdgeLength. Among them, both INF1 and INF6 comprised inflammatory cells with connections to tumor cells. INF1 cells were further distinguished by their connections to stroma cells. INF0 and INF7 comprised locally aggregated inflammatory cells without connection to tumor cells and their difference was that INF0 cells had connections to stroma cells. The other six clusters comprised inflammatory cells that were spatially scattered or only formed small I-I subgraphs characterized by low I-I_Nsubgraph and no I-I_ClusteringCoefficient (Fig. 2b, c).

We next analyzed the association between inflammatory cell clusters with breast cancer IHC subtypes, clinicopathological characteristics and patient prognosis. After adjusting for the total number of inflammatory cells, five clusters of scattered inflammatory cells were found to be enriched in HR+HER2- breast cancers. HER2+ breast cancers had higher proportion of INF0 cells, which indicated a large number of inflammatory cells residing near the stroma cells without infiltration into the tumor nest. By contrast, TNBCs were enriched for INF6 and INF7 cells, which were locally aggregated inflammatory cells without connection to stroma cells (Fig. 2d, e). These data suggested that breast cancers of different IHC subtypes had distinct spatial distribution patterns of inflammatory cells. Besides, breast cancers of low T category had more INF7 cells. Grade I-II breast cancers had more INF2, INF3, INF4, INF5 and INF9 cells, while those of grade > II were

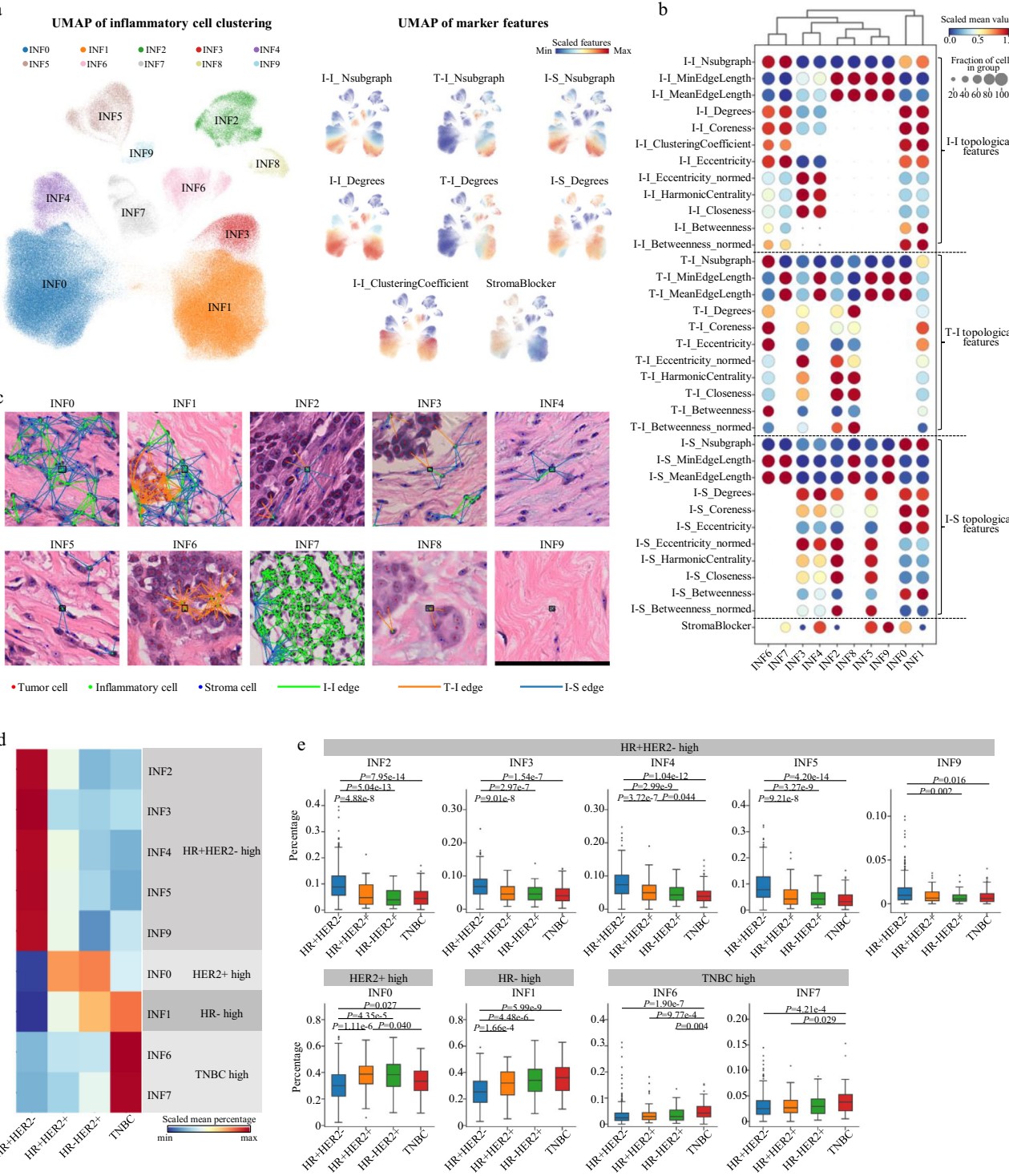

**Fig. 2 | Single-cell atlas of inflammatory cells. a** UMAP plot of unsupervised clustering results of 329,572 inflammatory cells sampled from 637 breast cancer whole slide images. The scaled values of selected features are shown on the right. **b** Dotplot of topological features according to the inflammatory cell clusters. The scaled mean values within each cluster (visualized by dot color) and the fraction of cells with positive values in the cluster (visualized by dot size) are shown. Note: If a cell is not connected to any other cells in a certain graph, the edge length-related features (MinEdgeLength and MeanEdgeLength) are set to 100 (pixel) which is the upper limit of distance between two connected cells. **c** Local nuclear graph visualization of the inflammatory cells of different clusters. A representative cell of each cluster is shown in the center of the images with a black bounding box. The edges between inflammatory and inflammatory cells (I-I edges), between tumor and inflammatory cells (T-I edges) and between inflammatory and stroma cells (I-S edges) are shown. Scale bar: 100 μm. **d** Comparison of the abundance of inflammatory cell clusters among the breast cancer IHC subtypes. Heatmap shows the scaled mean percentage of inflammatory cell clusters that are significantly different among the IHC subtypes. **e** Boxplots of the percentage of inflammatory cell clusters according to the IHC subtypes (HR+HER2-: n = 405; HR+HER2+: n = 85; HR-HER2+: n = 66; TNBC: n = 81). The center lines of boxplots indicate the median values; box limits show upper and lower quartiles; whiskers extend from box limits to the farthest data point within 1.5 × interquartile range; points beyond whiskers are outliers. P values are calculated using the two-sided Mann-Whitney U test with false discovery rate-correction for multiple testing.

enriched for INF0, INF1 and INF6 cells, which indicated more locally aggregated inflammatory cells (Supplementary Fig. S3a). As for prognosis, high abundance of INF1 and INF7 was associated with favorable prognosis (Supplementary Fig. S3b).

## Abundance of locally aggregated inflammatory cells indicates immunotherapy response

The immune cell infiltration pattern reflected the anti-tumor immunity status and was reported to correlated with immunotherapy response[10]. Thus, we explored the association between the abundance of inflammatory cell clusters and anti-tumor immune response. We first correlated the abundance of the inflammatory cell clusters with the microenvironment cell abundance which was estimated using single sample GSEA (ssGSEA) based on the RNA sequencing (RNA-seq) data (Methods)[21]. It was found that the abundance of INF0, INF1, INF6 and INF7 cells showed significant positive association with abundance of a variety of immune cell subsets (Fig. 3a). Besides, the abundance of these four clusters was positively correlated with the biomarkers or gene signatures that were reported to predict favorable immunotherapy response (Fig. 3b, Methods). Interestingly, these four clusters were all characterized by high I-I_Nsubgraph, I-I_Degrees and positive I-I_ClusteringCoefficient, which indicated local aggregation of inflammatory cells. These results suggested that high abundance of locally aggregated inflammatory cells was associated with the overall activation of tumor immune microenvironment and possibly with a favorable response to immunotherapy in breast cancer.

We next validated this finding in an independent immunotherapy cohort with available treatment response information (Supplementary Table S2 and Methods). This validation cohort was from the NCT04129996 clinical trial, where patients with advanced TNBC were treated with a PD-1 inhibitor-based regimen as first-line treatment[22]. Patients were classified as responders or non-responders according to whether or not they achieved objective response. We included 24 cases with biopsy specimen before the PD-1 inhibitor-based immunotherapy and collected the paraffin-embedded H&E-stained WSIs of the pretreatment specimens. We applied the sc-MTOP pipeline and extrapolated the inflammatory cell clustering from the FUSCC discovery cohort to the WSIs of the validation cohort (Methods). All of the ten inflammatory cell clusters were identified in this validation cohort with comparable proportions and feature profiles (Supplementary Fig. S4). We calculated an aggregated inflammatory cell abundance score (AIC score) as the sum of the abundance of INF0, INF1, INF6 and INF7 cells and examined its association with immunotherapy response (Fig. 3c, d). We found that AIC score was significantly higher in responders than in non-responders ($P = 0.045$) (Fig. 3e). In receiver operating characteristics (ROC) analysis, AIC score achieved an area under the curve (AUC) of 0.83 for identifying the responders, comparable to that of PD-L1 IHC score which was an established biomarker for immunotherapy response in advanced TNBC[23] (Fig. 3f). In addition, high AIC score indicated long progression-free survival (PFS) (log-rank $P = 9.12e-4$) and overall survival (OS) (log-rank $P = 0.026$) (Fig. 3g, h, Supplementary Fig. S5). Therefore, in this cohort of limited sample size, our data suggested that the high abundance of locally aggregated inflammatory cells can indicate better responsiveness to immunotherapy in patients with TNBC.

## Single-cell atlas of tumor cells and stroma cells

Adopting the same analytic approach as for inflammatory cells, we characterized the phenotypic diversity of tumor and stroma cells respectively. Tumor cells were classified into twelve clusters (named TUM0-TUM11) with different topological and morphological features (Fig. 4a–c). Except TUM9 and TUM10, tumor cells of the other clusters which accounted for 98.7% of all tumor cells were locally connected characterized by the positive value of T-T_Degrees. TUM0, TUM5 and TUM8 consisted of tumor cells with connections to stroma cells but no connection to inflammatory cells, and TUM0 was further characterized

by larger T-T_Nsubgraph and T-S_Nsubgraph, which indicated a large tumor cell nest with surrounding stroma cells. TUM1 and TUM7 comprised tumor cells with connections to both inflammatory and stroma cells, while TUM1 was further characterized by larger T-T_Nsubgraph, T-T_Degrees and T-S_Nsubgraph. TUM2, TUM3, TUM4 and TUM11 comprised tumor cells at the center of tumor nest with no connection to either inflammatory or stroma cells. Both TUM2 and TUM11 cells had relatively large nuclear size indicated by their large Area and AreaBbox. TUM11 cells was further characterized by long T-T_MinEdgeLength and no T-T_ClusteringCoefficient, which reflected a loose tumor cell distribution. By contrast, TUM3 cells had small nuclear size. TUM4 cells were characterized by large CellEccentricity, Elongation and low Circularity, which indicated spindle-shaped tumor nuclei. TUM6 was the only cluster with connections to inflammatory cells but no connection to stroma cells. We also examined the association of the tumor cell clusters with breast cancer IHC subtypes, clinicopathological characteristics and patient prognosis. The four clusters that comprised tumor cells with connections to inflammatory cells (TUM1, TUM6, TUM7 and TUM9) were enriched in HR-HER2+ and TNBCs, while TUM0 and TUM5 which were two clusters including tumor cells with connections to stroma cells were enriched in HR+HER2- breast cancers. Among the four clusters that comprised tumor cells with only tumor-tumor connections, TUM2 and TUM11 cells were enriched in HER2+ breast cancers; TUM3 cells were enriched in HR+HER2- breast cancers; TUM4 cells showed a tendency of enrichment in TNBCs (Supplementary Fig. S6a, b). Besides, elderly patients had more TUM8 and TUM10 cells. TUM0, TUM3, TUM5 and TUM10 cells were enriched in the grade I-II breast cancers, while TUM1, TUM4, TUM6, TUM7, TUM9 and TUM11 cells were enriched in grade > II breast cancers (Supplementary Fig. S6c). As for prognosis, high abundance of TUM0 and TUM5 was correlated with poor prognosis (Supplementary Fig. S6d).

Stroma cells were classified into nine clusters (named STR0-STR8) (Fig. 4d–f). STR1, STR2, STR4 and STR5 comprised stroma cells that had connections to tumor cells. Among these four clusters, only STR1 cells had connections to inflammatory cells. Compared with STR5, both STR2 and STR4 cells were locally connected stroma cells and STR2 cells were further distinguished by large T-S_Nsubgraph and S-S_Degrees, which indicated more stroma cells surrounding the tumor cells. STR3, STR6 and STR7 consisted of stroma cells with no connection to either tumor or inflammatory cells. STR7 comprised separated stroma cells with no connection to any cells. Compared with STR6, STR3 cells had larger S-S_Nsubgraph and S-S_Degrees, indicating more locally aggregated stroma cells. STR0 and STR8 were locally connected and separated stroma cells respectively with connections to inflammatory cells but no connection to tumor cells. These stroma cell clusters also displayed distinct enrichment patterns among the IHC subtypes. The clusters that comprised stroma cells with connections to inflammatory cells (STR0, STR1 and STR8) were enriched in HR-HER2+ and TNBCs, while STR2, STR3, STR4, STR6 and STR7 cells were enriched in HR+HER2- breast cancers (Supplementary Fig. S7a, b). Besides, STR4, STR5, STR6, STR7 and STR8 cells were enriched in elderly patients, while young patients had more STR1 cells. Tumor of low T category had more STR7 and STR8 cells. STR0 and STR1 cells were enriched in grade > II breast cancers, while grade I-II breast cancers had more STR2, STR3, STR4, STR6 and STR7 cells, which were stroma cells with few connections to inflammatory cells (Supplementary Fig. S7c). As for prognosis, high abundance of STR2, STR4 and STR5 was associated with poor prognosis (Supplementary Fig. S7d).

## Morphological intratumor heterogeneity of tumor nuclei indicates cell cycle pathway activity and CDK inhibitor response in HR+ breast cancer models

We noticed that unsupervised clustering of tumor cells identified four clusters with similar topological features but different morphological and texture features (TUM2, TUM3, TUM4 and TUM11). This indicated

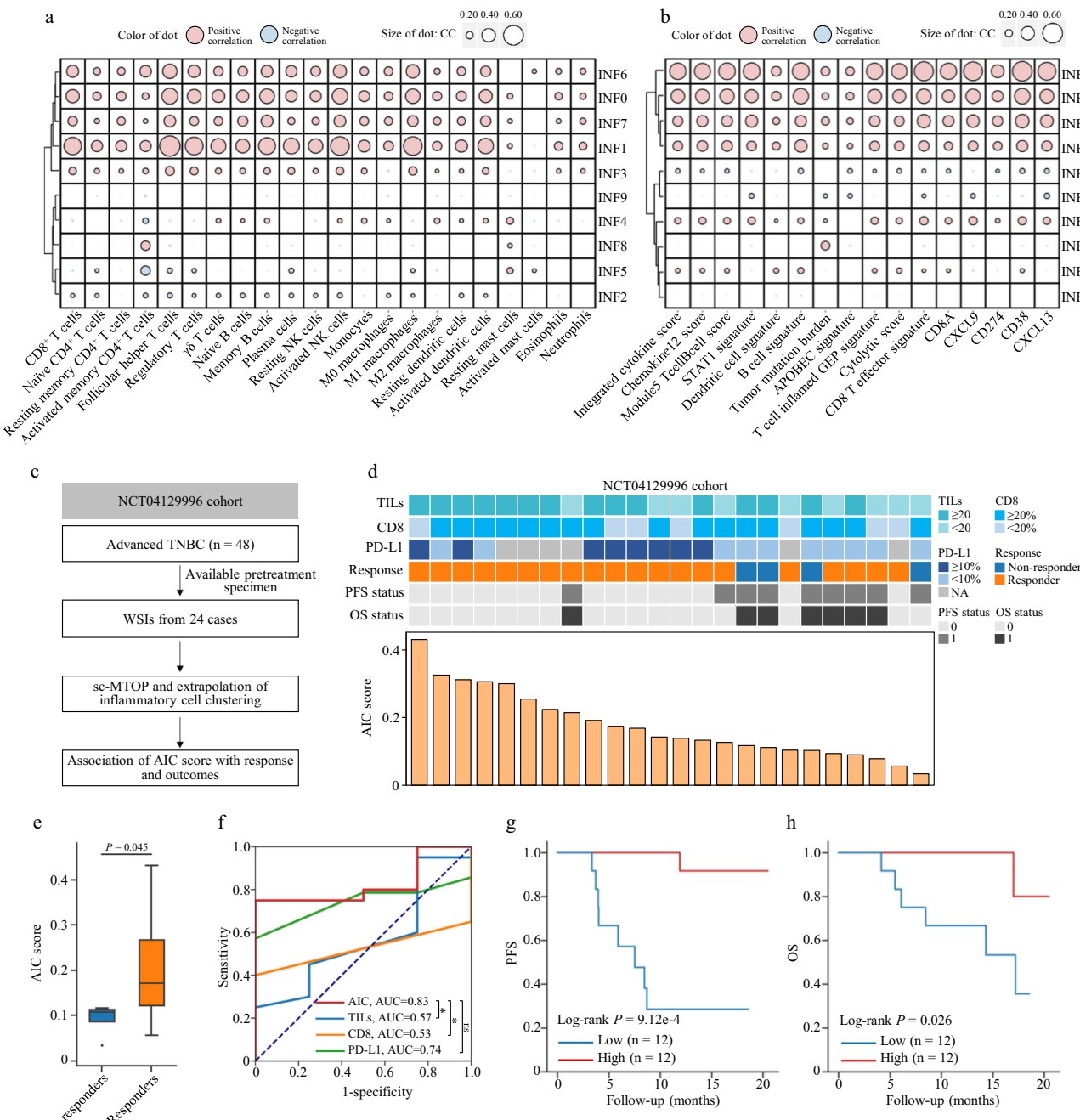

**Fig. 3 | Abundance of locally aggregated inflammatory cells indicates immunotherapy response.** Correlation between the abundance of inflammatory cell clusters and **a** abundance of microenvironment immune cells and **b** biomarkers indicating favorable immunotherapy response reported in the previous studies. Two-sided Spearman correlation analysis is performed with $P$ values corrected for multiple testing. Red/blue dots represent positive/negative correlation respectively. The size of the dots is proportional to the Spearman correlation coefficient. The outlined dots indicate false discovery rate corrected $P$ values < 0.05. Rows are ordered by hierarchical clustering. **c** Diagram of using the NCT04129996 cohort to investigate the association between the abundance of aggregated inflammatory cells and immunotherapy response. **d** NCT04129996 cohort information in TILs, CD8 IHC score, PD-L1 IHC score, treatment response, PFS status, OS status and AIC score. **e** Comparison of AIC score between the responders (patients who achieved

objective response, $n = 20$) and non-responders ($n = 4$). The center lines of boxplots indicate the median values; box limits show upper and lower quartiles; whiskers extend from box limits to the farthest data point within 1.5 × interquartile range; points beyond whiskers are outliers. $P$ value is calculated using the two-sided Mann-Whitney U test. **f** Receiver operating characteristic curves for using the AIC score, TILs, CD8 IHC score and PD-L1 IHC score to identify the responders. $P$ values for the comparison of AUC are calculated using the two-sided bootstrap test (AIC score vs. TILs: $P = 0.022$; AIC score vs. CD8 IHC score: $P = 0.012$; AIC score vs. PD-L1 IHC score: $P = 0.455$). *$P < 0.05$; ns, not significant. **g**, **h** Kaplan–Meier curves of progression-free survival and overall survival of patients with high- and low-AIC score in the NCT04129996 cohort. CC correlation coefficient, AIC score aggregated inflammatory cell abundance score, TILs tumor-infiltrating lymphocytes, AUC area under the curve, PFS progression-free survival, OS overall survival.

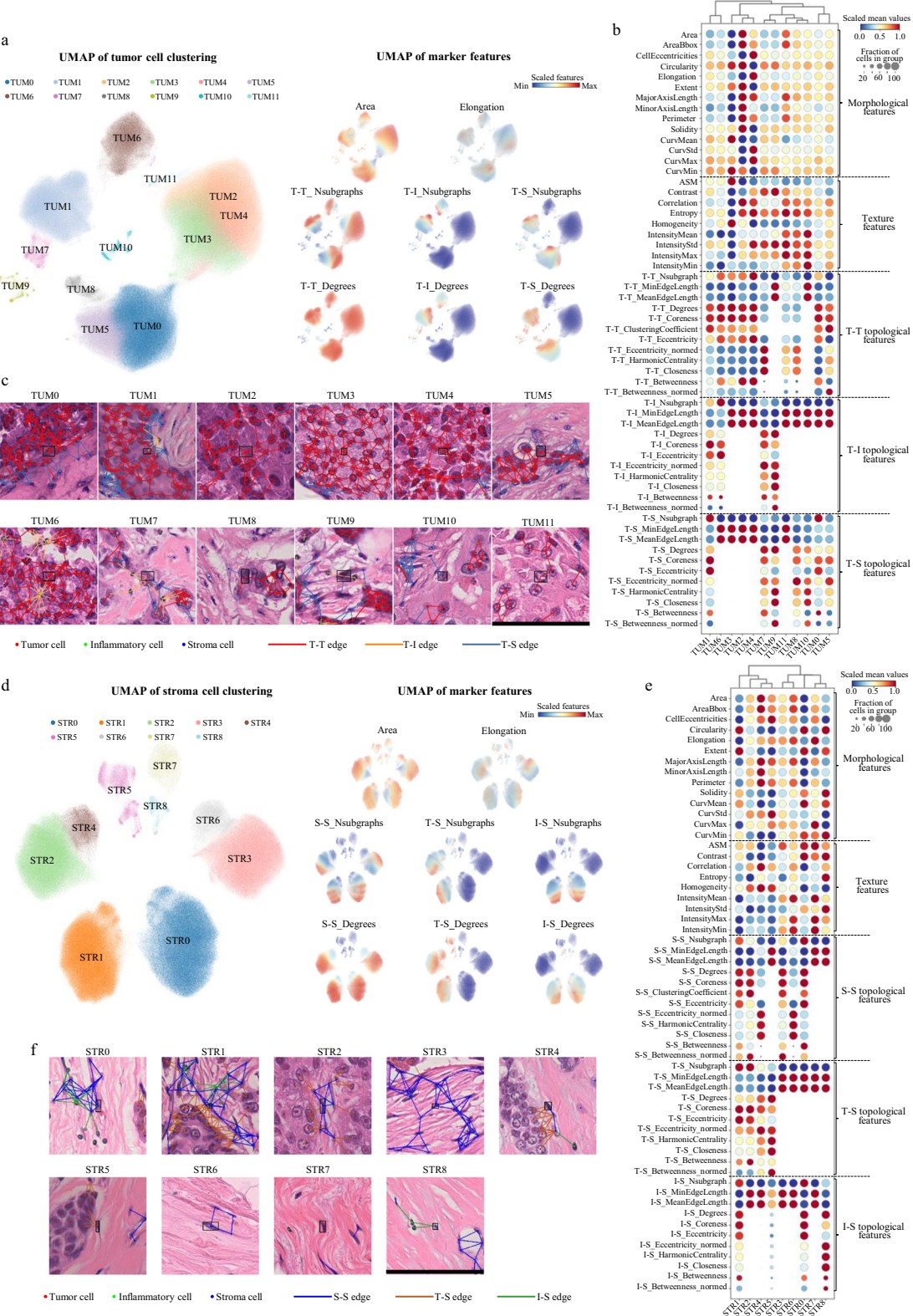

high variation of nuclear morphology among tumor cells and inspired us to investigate morphological heterogeneity of tumor nuclei in depth. Based on our sc-MTOP data, we quantified morphological intratumor heterogeneity (MITH) of tumor nuclei for each sample by computing the mean value of the standard variation of morphological and texture features among its tumor nuclei (Fig. 5a, b, Supplementary Fig. S8a and Methods). A moderate correlation (Spearman correlation coefficient: 0.20, $P = 2.67e-6$) was observed between MITH and genetic intratumor heterogeneity measured by mutant-allele tumor heterogeneity[24,25] (Fig. 5c). High MITH was associated with aggressive tumor characteristics including high T category, high tumor grade, HR-negative and HER2-positive status (Fig. 5d). We also correlated MITH with the tumor cell cluster abundance. The abundance of TUM4 which comprised the spindle-shaped tumor cells showed the highest positive correlation with MITH (Spearman correlation coefficient: 0.40, false discovery rate-corrected $P = 2.24e-24$) (Supplementary Fig. S8b).

**Fig. 4 | Single-cell atlas of tumor cells and stroma cells. a** UMAP plot of unsupervised clustering results of 963,226 tumor cells sampled from 637 breast cancer whole slide images. The scaled values of selected features are shown on the right. **b** Dotplot of features according to the tumor cell clusters. The scaled mean values within each cluster (visualized by dot color) and the fraction of cells with positive values in the cluster (visualized by dot size) are shown. Note: If a cell is not connected to any other cells in a certain graph, the edge length-related features (MinEdgeLength and MeanEdgeLength) are set to 100 (pixel) which is the upper limit of distance between two connected cells. **c** Local nuclear graph visualization of the tumor cells of different clusters. A representative cell of each cluster is shown in center of the images with a black bounding box. The edges between tumor and tumor cells (T-T edges), between tumor and inflammatory cells (T-I edges), and

between tumor and stroma cells (T-S edges) are shown. Scale bar: 100 μm. **d** UMAP plot of unsupervised clustering results of 705,367 stroma cells sampled from 637 breast cancer whole slide images. The scaled values of selected features are shown on the right. **e** Dotplot of features according to the stroma cell clusters. The scaled mean values within each cluster (visualized by dot color) and the fraction of cells with positive values in the cluster (visualized by dot size) are shown. **f** Local nuclear graph visualization of the stroma cells of different clusters. A representative cell of each cluster is shown in center of the images with a black bounding box. The edges between stroma and stroma cells (S-S edges), between tumor and stroma cells (T-S edges) and between inflammatory and stroma cells (I-S edges) are shown. Scale bar: 100 μm.

We next explored the molecular basis and clinical relevance of MITH. GSEA showed significant enrichment of cell cycle-related gene sets in high MITH samples. Interestingly, subgroup analysis revealed that this enrichment was even more significant in the HR+ breast cancers but was not observed in the HR- breast cancers (Fig. 5e and Supplementary Fig. S8c, e). The Ki-67 index and multigene proliferation score (MGPS)[26] were also higher in high-MITH samples only in HR+ but not in HR- breast cancers (Fig. 5f and Supplementary Fig. S8d, f). These results suggested that high MITH was associated with the activation of proliferation and cell cycle pathways in HR+ breast cancers. Based on this, we further investigated which part of cell cycle pathway was upregulated in high-MITH HR+ breast cancers (Fig. 5g). We first performed IHC staining of phosphorylated Rb protein (phospho-Rb) and found that phospho-Rb was significantly higher in high-MITH than in low-MITH HR+ breast cancers (Fig. 5h). Furthermore, high MITH was associated with high mRNA expression of CCNE1, CCNE2, CDK2, E2F1, E2F2, E2F3 and high E2F target signature score (Methods). By contrast, there is no difference in the expression of CCND1, CCND2, CCND3, CDK4 or CDK6 between the high- and low-MITH subgroups (Fig. 5i and Supplementary Fig. S8g). Genomic analysis showed that high-MITH was associated with high frequency of *TP53* mutation, which has been reported to lead to the upregulation of CDK2[27,28] (Supplementary Fig. S8h). These results indicated that in HR+ breast cancers, high MITH was associated with the Cyclin E/CDK2-dependent activation of cell cycle pathway. Since the Cyclin E/CDK2 activation has been reported as an important mechanism of CDK4/6 inhibitor resistance[29], we hypothesized that MITH may be a biomarker for CDK4/6 inhibitor resistance in HR+ breast cancers.

We attempted to validate our hypothesis using drug sensitivity test of patient-derived organoids (PDOs). We established PDOs using the tumor surgical specimens from HR+ breast cancers in our center. Meanwhile, the H&E-stained tumor slides made from the same specimen as the corresponding PDOs were collected and scanned to generate WSIs. sc-MTOP was applied to these WSIs and MITH was evaluated based on the sc-MTOP data. Both the genetic alterations and MITH of the primary tumor were preserved in the corresponding PDOs (Supplementary Fig. S9). The PDOs were treated with CDK4/6 inhibitor Abemaciclib or CDK2/4/6 inhibitor PF-06873600 and the relative viability was assessed after the treatment. We investigated the association between MITH and drug response (Fig. 5j). It was found that PDOs from high-MITH samples were less sensitive to CDK4/6 inhibitor than those from low-MITH samples (Comparison of relative viability, $P = 7.36e-4$). By contrast, no significant difference in response to CDK2/4/6 inhibitor was observed between the PDOs from the high- and low-MITH samples (Comparison of relative viability, $P = 0.859$) (Fig. 5k). These results supported our assumption that in HR+ breast cancers, high MITH indicated CDK4/6 inhibitor resistance and CDK2/4/6 inhibitor may be an effective therapeutic option for the high-MITH HR+ breast cancers.

## Recurrent micro-ecological modules characterize the spatial diversity of breast cancer ecosystem

The above analysis characterized the phenotypic diversity of breast cancer ecosystem on the single-cell level and established a cellular taxonomy for tumor, inflammatory and stroma cells respectively. Considering that tumor is a complex ecosystem where cells organize in certain spatial patterns to perform specific functions, we next aimed to identify recurrent patterns of locoregional cellular organization and further decipher the spatial diversity of breast cancer ecosystem. We first tessellated each WSI into square domains (referred to as spots) and for each spot calculated the number of cells of each cell clusters. Then, we revealed colocalization of cell clusters through spatial correlation. The micro-ecological modules (MEMs) were defined based on the hierarchical clustering of the spatial correlation matrix (Fig. 6a and Methods). Eight MEMs were identified (Fig. 6b): i) Module1 tumor core (Module1_TC) which included TUM2, TUM3 and TUM4; ii) Module2 loosely distributed tumor cells (Module2_LT) including TUM10 and TUM11; iii) Module3 discrete inflammatory cell infiltration (Module3_DI) including INF8 and TUM6; iv) Module4 colocalization of tumor cells and stroma cells (Module4_TS) including STR5, STR4, TUM0, TUM8, STR2 and TUM5; v) Module5 low cellularity (Module5_LC) including STR6, STR7, STR8 and INF9; vi) Module6 colocalization of stroma and inflammatory cells (Module6_SI) including INF4, INF5, STR0 and STR3; vii) Module7 colocalization of tumor, stroma and inflammatory cells (Module7_TSI) including INF2, INF3, STR1 and TUM1; viii) Module8 local inflammatory cell aggregation (Module8_IA) including INF0, INF1, INF6, INF7, TUM7 and TUM9. For each MEM, a score was calculated for each spot as the percentage of cells belonging to this module. To characterize the spatial patterns of the eight MEMs, spatial auto-correlation was measured using Moran's I index[30]. Except for the Module2_LT and Module3_DI, all the other MEMs exhibited organized spatial patterns with median Moran's I index ranging from 0.39-0.54, which reflected the spatial variability of these modules across the tumor ecosystem (Fig. 6c).

To gain further insights into the ecosystem spatial diversity, we classified all spots into the eight MEMs according to their maximum MEM score and those with all MEM scores of 0 were assigned to a new module, Module0 no cell (Module0_NC) (Fig. 6d). Then, we first validated our MEM-based ecosystem characterization by evaluating its the consistency with the histological patterns. Histological regions of tumor, stroma, immune cell aggregates and immune infiltrates in tumor were manually annotated by a specialist breast pathologist. All of the histological regions were enriched with appropriate MEMs (Fig. 6e, f). Furthermore, regions of the same histological category can be further classified according to the MEMs. For example, the histological tumor region 1 and tumor region 2, although both were tumor regions, displayed different locoregional cellular composition and organization. The tumor region 1 was mainly composed of Module1_TC characterized by a tumor nest with few stroma cells, while the tumor region 2 was a mixture of Module1_TC and Module4_TS with more stroma cells and tumor-stroma interactions. Similarly, the diversity

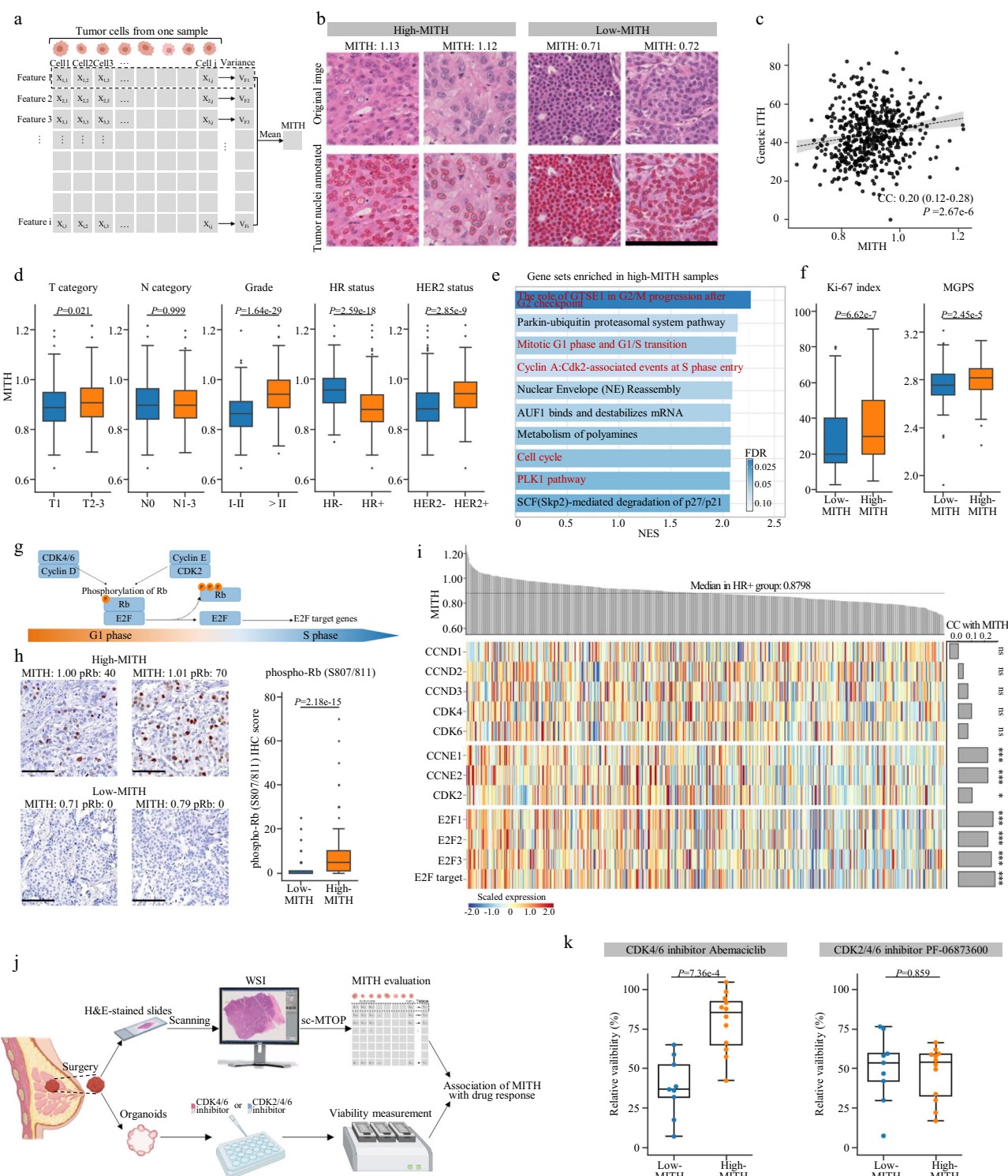

of regions of stroma and immune infiltration can also be mapped according to the MEMs (Fig. 6g).

We explored using the patch-based deep learning method to identify the recurrent patterns of locoregional cellular structures and compared it with our sc-MTOP-based method. We examined two deep learning models: self-supervised learning model by Ciga et al.[31] and KimiaNet by Riasatian et al.[32]. We extracted deep features of the same image "spots" as used in the aforementioned MEM analysis and defined self-supervised learning model-based MEM (sslMEM) classification and KimiaNet-based MEM (KimiaMEM) classification through clustering on the corresponding features (Supplementary Fig. S10a, Methods). Both

sslMEM and KimiaMEM classifications showed higher inter-MEM variations and lower intra-MEM variations compared to random classification, suggesting that they can indicate the locoregional cellular composition to some extent, but their performance did not attain that of our sc-MTOP-based MEM classification (Supplementary Fig. S10b). Besides, the sslMEMs and KimiaMEMs showed less congruency with histological patterns than our sc-MTOP-based MEMs (Supplementary Fig. S10c, refer to Fig. 6e, f for comparison). Furthermore, in sc-MTOP-based MEMs, high percentage of Module8_IA was associated with favorable prognosis, while high percentage of Module4_TS was correlated with poor prognosis. By comparison, no sslMEMs or

**Fig. 5 | Morphological intratumor heterogeneity of tumor nuclei indicates cell cycle pathway activity and CDK inhibitor response in HR+breast cancer.**
**a** Calculation of MITH. Created with BioRender.com. **b** Representative local images of high- and low-MITH tumors. Scale bar: 200 μm. **c** Correlation between MITH and genetic intratumor heterogeneity. Fitted linear regression line is shown. Two-sided Spearman correlation analysis is performed. **d** Association between MITH and clinicopathological features. *P* values are calculated using the two-sided Mann-Whitney U test. **e** Gene sets enriched in high-MITH samples in HR+ breast cancers revealed by GSEA. Cell cycle-related gene sets are marked in red. The NES and FDR output by GSEA are presented. **f** Comparison of Ki-67 index and MGPS between high- and low-MITH samples in HR+ breast cancers. *P* values are calculated using the two-sided Mann-Whitney U test. **g** Diagram depicting how Cyclin D/CDK4/6 and Cyclin E/CDK2 regulate G1/S phase progression and activate transcription of E2F target genes. **h** Comparison of phospho-Rb (S807/811) IHC score between high- ($n = 164$) and low-MITH ($n = 163$) samples in HR+ breast cancers. Representative images of IHC staining are presented. Scale bar: 100 μm. *P* value is calculated using

the two-sided Mann-Whitney U test. **i** Heatmap of the mRNA expression of CCND1, CCND2, CCND3, CDK4, CDK6, CCNE1, CCNE2, CDK2, E2F1, E2F2, E2F3 and E2F target signature score in HR+ breast cancers. Their correlation with MITH is indicated by bar plot on the right through two-sided Spearman correlation analysis. ***$P < 0.001$; *$P < 0.05$; ns, not significant. **j** Experiment design to evaluate CDK inhibitor response of PDOs stratified by MITH. Created with BioRender.com. **k** Relative viability of PDOs treated with 1 μM CDK4/6 inhibitor Abemaciclib and 0.4 μM CDK2/4/6 inhibitor PF-06873600 (High-MITH: $n = 12$; low-MITH: $n = 9$). *P* values are calculated using two-sided Mann–Whitney U test. For Fig. 5d, f, h and k, the center lines of boxplots indicate median values; box limits show upper and lower quartiles; whiskers extend from box limits to the farthest data point within $1.5 \times$ interquartile range; points beyond whiskers are outliers. In Fig. 5e, f, h and k, high- and low-MITH subgroups are defined using the median MITH value (0.8798) of HR+ breast cancers. MITH morphological intratumor heterogeneity, CC correlation coefficient, NES normalized enrichment score, FDR false discovery rate, MGPS multigene proliferation score, IHC immunohistochemistry.

---

KimiaMEMs showed significant association with patient prognosis (Supplementary Fig. S10d). These data indicated that compared with the deep learning method, MEMs identified by our sc-MTOP-based method can better represent the patterns of local multicellular structures in tumor ecosystem and provide prognostic relevant information.

## Micro-ecological module-based breast cancer ecotypes associated with molecular features and patient prognosis

Based on the decomposition of tumor ecosystem by MEMs, we aimed to establish a draft taxonomy of breast cancer ecosystem based on digital pathology (Methods). Hierarchical clustering of the tumor MEM composition identified four ecotypes. Tumors of Ecotype 1 ($n = 163$, 25.6%) were enriched for Module4_TS; Ecotype 2 ($n = 158$, 24.8%) for Module7_TSI and Module8_IA; Ecotype 3 ($n = 149$, 23.3%) for Module6_SI and Module5_LC; and Ecotype 4 ($n = 167$, 26.2%) for Module1_TC (Fig. 7a and Supplementary Fig. S11a, b). The cell clusters enriched in each ecotype coincide with the MEMs enriched in that ecotype (Supplementary Fig. S11c).

The molecular alterations can influence the biological behavior of tumor cells and leave footprint in the phenotype of tumor ecosystem[33,34]. Therefore, we explored the molecular features associated with these four ecotypes using the multiomics data. On the transcriptional level, GSEA was performed to reveal the enriched gene sets in each ecotype. It was found that Ecotype 1 showed enrichment of estrogen-related and cell cilium-related gene sets. Ecotype 2 was enriched for immune-related gene sets, which indicated an immune-activated tumor microenvironment. Ecotype 3 displayed enrichment of gene sets associated with focal adhesion and extracellular matrix. Ecotype 4 was enriched for pathways of electron transport chain and oxidative phosphorylation (Fig. 7b). On the genomic level, difference in oncogenic pathway alterations and copy number alterations (CNAs) was compared across the four ecotypes. Among the known oncogenic signaling pathways, *TP53* pathway alteration was significantly enriched in Ecotype 2 (Supplementary Fig. S12a). As for CNAs, Ecotype 2 had the highest frequency of *ERBB2* (17q12, 50.4%) amplification, which was in line with the high proportion of HER2+ breast cancers of this ecotype. Ecotype 3 was enriched with the amplification of *KLF8* (Xp11.21, 45.8%), a transcription factor playing a regulatory role in epithelial-mesenchymal transition. Ecotype 4 displayed the highest frequency of *ATP6V0B* (1p34.2, 23.4%) and *COX18* (4q13.3, 19.3%) amplification, which were involved in oxidative phosphorylation (Supplementary Fig. S12b).

We next compared the difference in tumor clinicopathological characteristics among the four ecotypes. Ecotype 1 and Ecotype 3 comprised more lymph node-positive breast cancers and no significant difference was observed in T category among the four ecotypes. Ecotype 2 was characterized by high tumor grade. Ecotype 1 had a high proportion of HR+HER2- breast cancers, while Ecotype 2 comprised more HR- breast cancers. (Fig. 7c). Subgroup analysis according

to IHC subtypes showed that only in the HR-HER2+ subgroup, Ecotype 1 and Ecotype 3 breast cancers were significantly associated with higher rates of lymph node metastasis. The higher tumor grade of Ecotype 2 and Ecotype 4 breast cancers was most pronounced in HR+HER2- and TNBC subgroups (Supplementary Fig. S12c). As for prognosis, patients with Ecotype 2 breast cancer had the best RFS, while those with Ecotype 1 breast cancer had the worst RFS (log-rank $P = 0.025$) (Fig. 7d). Subgroup analysis showed that patients with Ecotype 1 breast cancer represented a subpopulation with poorer prognosis among the HR+ cases, while those with Ecotype 2 breast cancer constituted a subpopulation with better prognosis among the HR- cases (Supplementary Fig. S12d). The poor prognosis of Ecotype 1 patients may be explained by their enrichment of Module4_TS and the corresponding TUM0, TUM5, STR5, STR4 and STR2 cells, which correlated with poor prognosis. The better prognosis of Ecotype 2 patients may be attributed to their enrichment of Module8_IA and the corresponding INF0 and INF7 cells, which were associated with favorable prognosis. Multivariate Cox analysis demonstrated that the ecotype classification was an independent prognostic factor (Fig. 7e and Supplementary Table S3).

To facilitate the potential application of our ecosystem classification to new cases, we developed a classifier to distinguish samples' tumor ecotypes using their MEM composition as input. We trained a support vector machine (SVM) model using 80% cases from the FUSCC cohort and tested its performance in the remaining 20% cases (Methods, Supplementary Fig. S13a). It achieved a weighted F1 score of 0.98 in the test set (Supplementary Fig. S13b). In summary, based on the MEM composition, we characterized the inter-patient ecosystem diversity and identified four distinct breast cancer ecotypes. These four ecotypes were associated with different tumor molecular features and patient prognosis. An SVM classifier was developed to realize the ecotype identification of samples based on its MEM composition.

## Reproducibility evaluation of sc-MTOP algorithm on a second slide of the same patient

To evaluate the reproducibility of our sc-MTOP algorithm on a second slide of the same patient, we a) randomly selected 50 patients from the FUSCC cohort, b) collected and scanned a second slide from each, c) applied sc-MTOP and subsequent analytic pipeline including the cell clustering extrapolation, MEM identification and ecotype classification, and d) assessed the consistency of the major metrics and markers of potential clinical relevance between these second slides and the original used ones. The cell type and cell cluster percentage showed high correlation between the second slides and the original used slides, with most correlation coefficient values higher than 0.60 (Spearman correlation test, all false discovery rate-corrected $P < 0.01$, median correlation coefficient=0.81) (Supplementary Fig. 14a). The

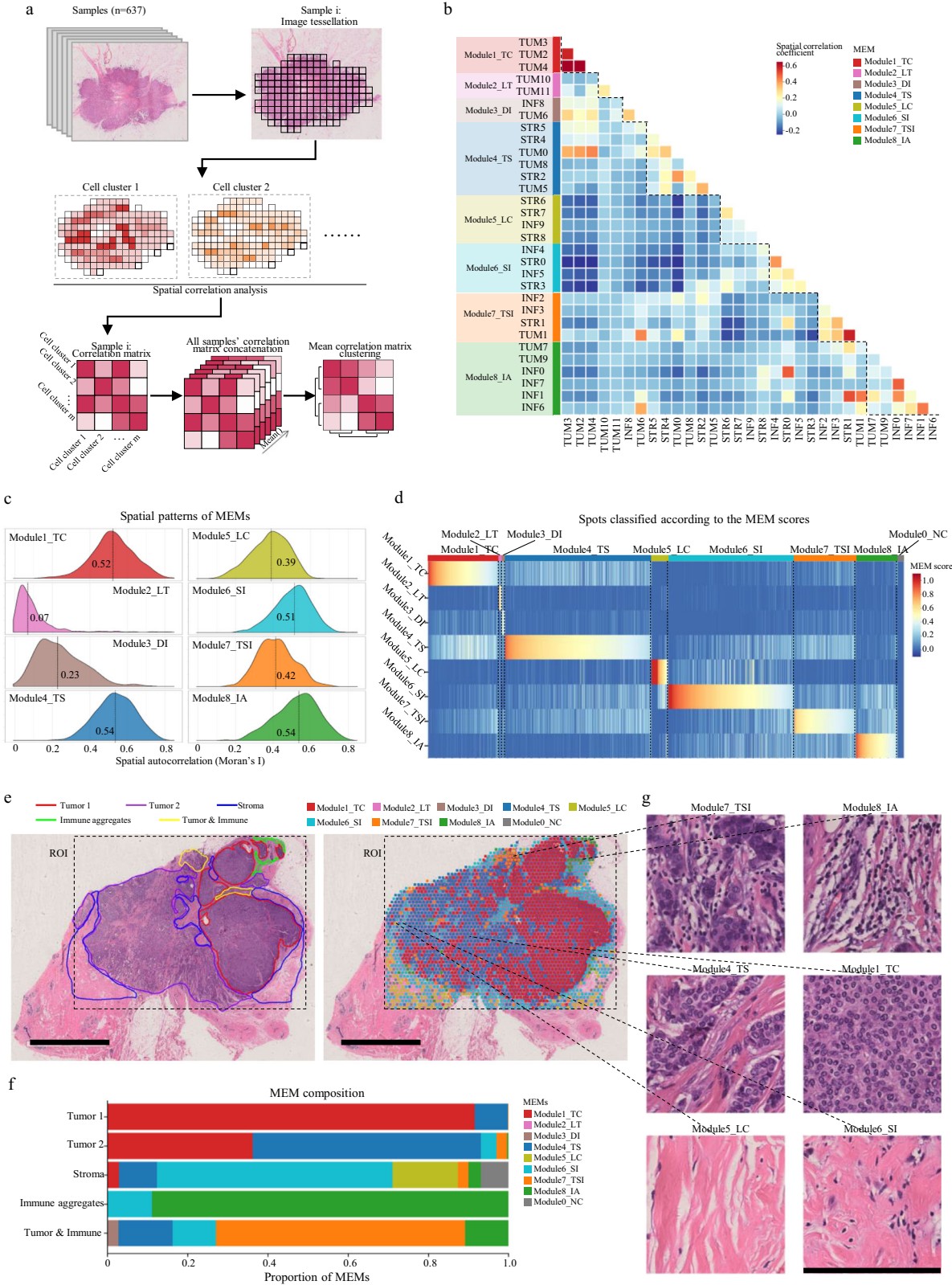

ecotype classification results showed high consistency between the second slide and the original used (accuracy: 88.0%, F1 score: 0.87) (Supplementary Fig. S14b). As for the clinically-relevant features, both AIC score (Spearman correlation coefficient: 0.91 (0.84-0.95), $P = 7.24e{-}20$) and MITH (Spearman correlation coefficient: 0.76 (0.61–0.86), $P = 2.02e{-}10$) were significantly correlated between the two groups of slides (Supplementary Fig. S14c, d). These results

demonstrated high reproducibility of the results of our algorithm on different slides from the same patient.

## Online platform for data management and real-time sc-MTOP analysis

To facilitate data sharing and the use of sc-MTOP by other researchers, we developed an online platform for data management and real-time

**Fig. 6 | Recurrent micro-ecological modules characterize the spatial diversity of breast cancer ecosystem. a** Diagram for the identification of micro-ecological modules (MEMs). **b** Hierarchical clustering of spatial correlation matrix of tumor, inflammatory and stroma cell clusters identifies eight micro-ecological modules: i) Module1_TC, Module1 tumor core; ii) Module2_LT, Module2 loosely distributed tumor cells; iii) Module3_DI, Module3 discrete inflammatory cell infiltration; iv) Module4_TS, Module4 colocalization of tumor cells and stroma cells; v) Module5_LC, Module5 low cellularity; vi) Module6_SI, Module6 colocalization of stroma and inflammatory cells; vii) Module7_TSI, Module7 colocalization of tumor, stroma and inflammatory cells; viii) Module8_IA, Module8 local inflammatory cell aggregation. Rows and columns are ordered by hierarchical clustering. **c** Spatial patterns

of MEMs based on Moran's I statistics. **d** Heatmap of MEM scores of all spots. Spots (columns) are classified into the eight MEMs according to its maximum MEM score. Spots where all module scores are zero are classified as Module0_NC. **e** Thumbnail of a whole slide image with manually histological annotation (left) and with MEM mapping (right). In histological annotation, a pathologist manually delineated the regions of different histological patterns including two types of tumor regions (red and purple), stroma (blue), immune cell aggregation in stroma (green) and immune infiltration in tumor (yellow). Scale bar: 4 mm. **f** Composition of MEMs of different histological regions. **g** Mapping the spatial diversity of histological regions of immune infiltration, tumor and stroma based on MEMs. Scale bar: 200 μm. MEM micro-ecological module, ROI regions of interest.

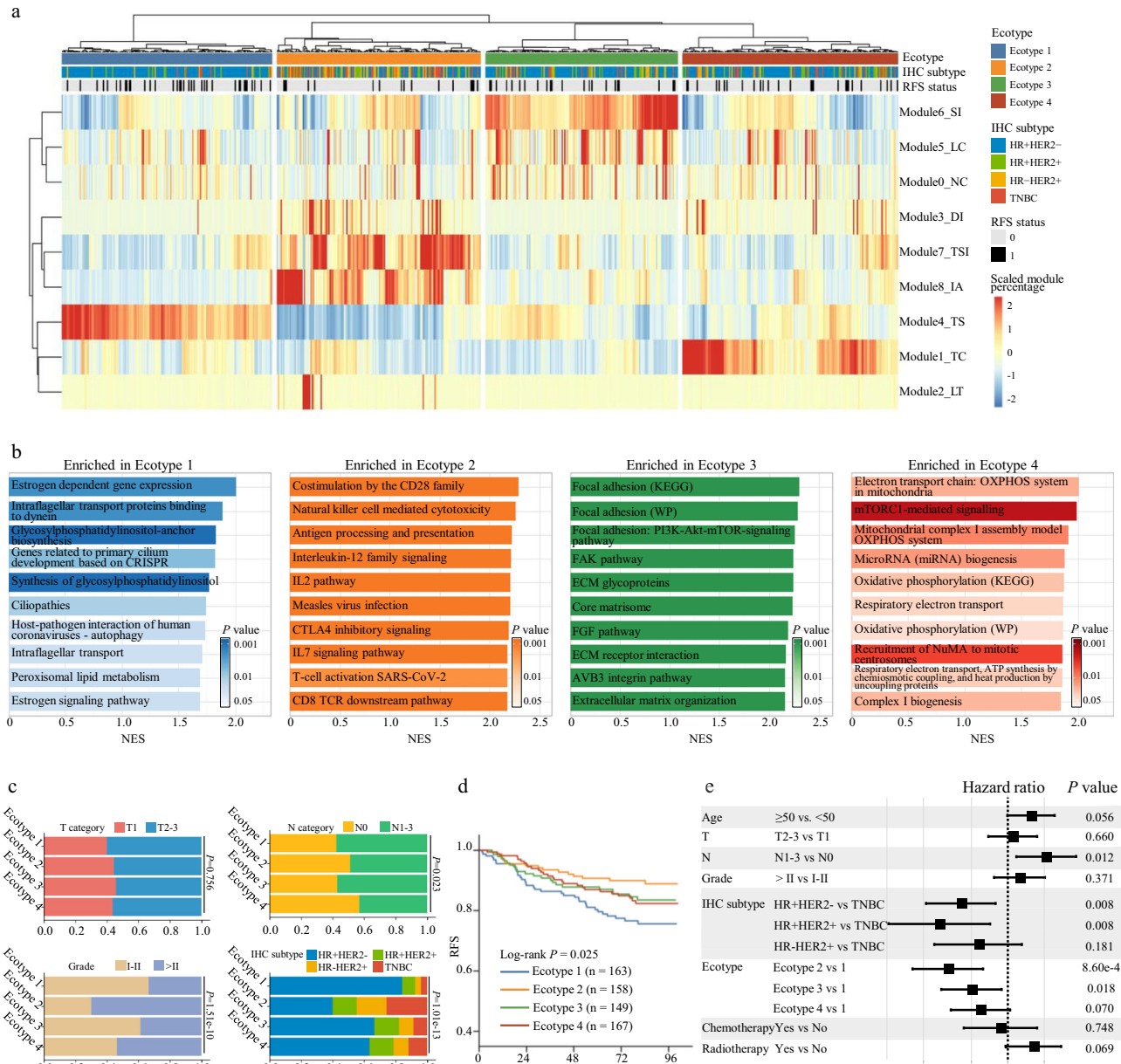

**Fig. 7 | Micro-ecological module-based breast cancer ecotypes associated with molecular features and patient prognosis. a** Hierarchical clustering of micro-ecological module composition identifies four breast cancer ecotypes. **b** Gene sets enriched in the samples of each ecotype revealed by gene sets enrichment analysis. The NES and *P* value output by GSEA are presented. **c** Association between breast cancer ecotypes and tumor T category, N category, grade and IHC subtypes.

*P* values are calculated using the chi-square test. **d** Kaplan–Meier curves of recurrence-free survival according to the breast cancer ecotypes. **e** Multivariate Cox regression analysis reveals breast cancer ecotype as an independent prognostic factor of recurrence-free survival (*n* = 537). Squares and whiskers represent point estimates and the 95% confidence interval of hazard ratios. RFS recurrence-free survival, IHC immunohistochemistry, NES normalized enrichment score.

analysis (http://sc-mtop.biosolver.cn/). The platform comprises two modules. The first is a data management and sharing module. All WSIs of the FUSCC cohort have been uploaded and can be viewed using an online digital pathology viewer. The sc-MTOP data together with the clinical and multiomics information are presented in company with the WSI of the corresponding patient. The second module is an online analysis module, which supports real-time analysis for user-uploaded WSIs. We have modularized and deployed the sc-MTOP framework and the subsequent analytic pipeline including the cell clustering extra-polation, MEM identification and ecotype classifier as an online tool. For each uploaded WSI, sc-MTOP is executed on a cloud server and the raw output data are allowed for download. In addition, an ecosystem portrait is automatically generated which presents the clinically rele-vant ecosystem features of this sample including the cellular compo-sition, AIC score, MITH and the tumor ecotype. This portrait may provide valuable information for prognostic evaluation and ther-apeutic decision-making. We have provided a test account for users to access our platform and have made a video to illustrate its functions (Supplementary Movie 1).

## Discussion

In this study, we presented sc-MTOP framework to characterize indi-vidual cells on WSIs by extracting the nuclear morphological and topological features. By applying it to a large breast cancer cohort, we depicted an extensive single-cell atlas including 410 million cells and characterized the phenotypic diversity of breast cancer ecosystem on multiple levels. Furthermore, integrative analysis with clinical and multiomics data identified ecosystem features that may serve as bio-markers informing treatment responses (Fig. 8).

Over the last few years, there has been growing interest in ana-lyzing the tumor ecosystem using different technologies. Based on the bulk RNA-seq, computational methods have been developed to pro-vide rough estimates of cell types within the tumor ecosystem[35,36]. Single-cell RNA-seq, on the other hand, provides transcriptional pro-files of individual cells, enabling the characterization of the tumor ecosystem in terms of the composition, transcriptional patterns and functional states of microenvironmental cells. However, a major lim-itation of this technology is the loss of cellular spatial information to understand cellular organizations and interactions within the tissue context[37]. More recently, several technologies have been developed for the gene expression measurement in a spatially-resolved manner. Multiplex immunohistochemistry technologies such as image mass cytometry and CODEX allowed targeted detection of specific proteins in individual cells in situ[38,39]. Nanostring GeoMx/CosMx technology

permitted simultaneous RNA and protein expression detection and increased the number of markers that can be detected[40]. These methods can provide important insight into how individual cells of specific types and functional properties organized in space to form the tumor ecosystem[41,42]. However, the high cost, material requirements and technological complexity limited their use in large cohort and clinical implementation. By comparison, our sc-MTOP framework are unable to classify cells into specific functional subsets with molecular annotations, but it provides a different perspective for characterizing the tumor ecosystem, which focuses on the nuclear morphological and spatial topological features. It can be easily applied to the entire WSIs to realize panoramic characterization of tumor ecosystem at a single-cell resolution with the cell spatial information preserved. In addition, sc-MTOP only requires broadly available H&E-stained tumor slides as raw materials, which facilitated its validation and optimization in large clinical cohorts and the potential implementation in clinical practice at a much lower cost.

In this study, we applied the sc-MTOP algorithm to 637 WSIs and obtained raw feature data over 410 million cells. To comprehensively dissect the phenotypic diversity of breast cancer ecosystem, we adopted an analytic workflow mainly based on clustering analyses. From the raw, abstract and massive feature data, clustering analyses generated visualizable and biologically interpretable data including cell clusters, MEMs and ecotypes, which enabled the characterization of the phenotypic diversity of breast cancer ecosystem on multiple levels. First, at the single cell level, the cell clusters generated by fea-ture clustering dissected the morphological and topological feature diversity of individual tumor, inflammatory and stroma cells respec-tively. Second, the MEMs identified through cell cluster co-localization analysis achieved intuitive representations of distinct local multi-cellular structures and were used to map the spatial diversity of tumor ecosystem. Third, on the patient level, the ecotypes identified through the clustering of samples' MEM composition reflected the inter-patient diversity of breast cancer ecosystem (Fig. 8).

Immune infiltration in the tumor microenvironment is an impor-tant biomarker of the host anti-tumor immunity. The density of TILs has been reported in a variety of solid tumors as a prognostic bio-marker and a predictive biomarker for immunotherapy response[4,5,43,44]. Recently, Wang et al. developed a model based on quantitative fea-tures of TILs density and spatial arrangement to predict treatment response to immunotherapy in patients with non–small cell lung cancer and gynecological cancer[10]. The study leveraged a supervised machine-learning workflow of feature extraction, feature selection and model development. By comparison, in our study, based on data from

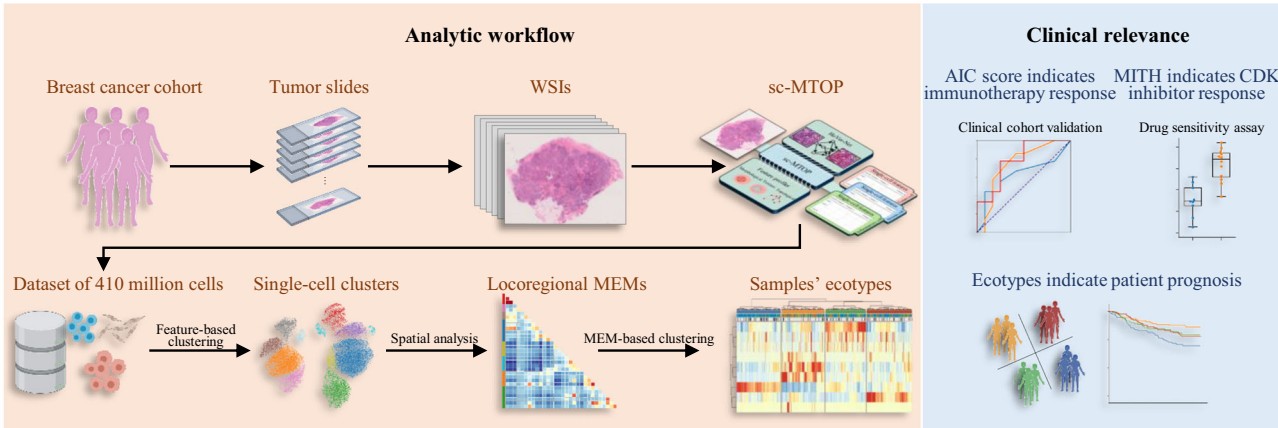

**Fig. 8 | Summary of the present study.** Left panel: The analytic workflow of generating and using sc-MTOP data to dissect the ecosystem diversity of human breast cancer at multiple levels. Right panel: The ecosystem features with ther-apeutic and prognostic implications. Created with BioRender.com. sc-MTOP single-cell morphological and topological profiling, MEM micro-ecological module, AIC score aggregated inflammatory cell abundance score, MITH morphological intra-tumor heterogeneity.

nuclear segmentation and feature extraction, we adopted a data-driven analytic pipeline. We first revealed the topological feature diversity of inflammatory cells through unsupervised clustering and characterized the samples' immune infiltration patterns using the abundance of different inflammatory cell clusters. Then, integrated analysis with multiomics data revealed the association between the cluster abundance and the immune activation status of tumor micro-environment and molecular biomarkers for immunotherapy response. Based on the results, we built an AIC score measuring the abundance of locally aggregated inflammatory cells and validated its association with immunotherapy response in an external clinical cohort. Our unsupervised clustering-based analytic approach maximized the use of our sc-MTOP dataset and the corresponding multiomics data[45]. Further studies are required to compare these two approaches for treatment response prediction and investigate whether an integrated method can lead to a better result.

Intratumor heterogeneity refers to the diversity of cancer cell populations in their genetic, phenotypic or behavioral characteristics within the same tumor[46]. Genetic ITH has been extensively studied and reported to play an important role in driving cancer progression and drug resistance, but the tumor nuclear morphological heterogeneity has rarely been investigated[47]. Andor et al. used the CellProfiler software to quantify histologic ITH in several cancer types based on H&E-stained pathological images and observed moderate correlation between histologic and genetic ITH. However, they did not analyze histologic ITH in breast cancer and did not associate the histologic ITH with biological processes[48]. In our study, we presented a method based on sc-MTOP data to quantify tumor nuclear MITH, which had the following advantages over Andor et al.'s method: a) nuclear type classification allowed MITH evaluation on only tumor nuclei which avoided computational bias due to the mixing of other cell types; b) more morphological and texture descriptors were included for the comprehensive characterization of MITH; c) calculated based on the entire WSI rather than a limited local image. More importantly, we revealed that high MITH indicated cell cycle pathway activation in HR+ breast cancers and such activation was mainly due to the upregulation of Cyclin E/CDK2, which was an important mechanism of CDK4/6 inhibitor resistance[49]. The drug response test using PDOs further validated that high MITH was an indicator for CDK4/6 inhibitor resistance in HR+ breast cancers and CDK2/4/6 inhibitor may be an effective option for the patients with high-MITH HR+ breast cancer.

The sc-MTOP data can be used not only for the characterization of single-cell features, but also for mapping the spatial diversity of tumor ecosystem. Here, following a spatially resolved analytical approach, we identified nine recurrent MEMs based on the spatial relationship of different cell clusters. These MEMs represented distinct local multi-cellular structures and showed good congruency with histological patterns. We also tried to use the pre-trained deep learning models to extract features of local image patches and identify deep feature-based MEMs through unsupervised clustering. Although the models were developed under weakly supervised or unsupervised settings, their generated feature-based MEMs can also indicate the cellular composition of patches and showed partial congruency with the histological patterns. This reflected the good robustness and generalizability of these deep learning models. However, without specific labels for supervised learning, convolutional neural network models tend to focus more on image texture features rather than the shape or topological features[50]. This may explain why they underperformed our sc-MTOP-based method in characterizing the local multicellular structures. Moreover, further studies could be planned by integrating our data with spatial omics data for the multimodal characterization of tumor ecosystem and the discovery of biologically meaningful tissue structures[51].

Four breast cancer ecotypes were identified through the unsupervised clustering of MEM composition of WSIs. These four ecotypes shared remarkable similarity with the conserved pan-cancer micro-environment subtypes based on transcriptomic analysis proposed by Bagaev et al.[52]. Their fibrotic subtype corresponded to our Ecotype 1 which was enriched with Module4 colocalization of tumor cells and stroma cells; immune-enriched non-fibrotic subtype to our Ecotype 2 enriched with Module8 local inflammatory cell aggregation; immune-enriched fibrotic subtype to our Ecotype 3 enriched with Module6 colocalization of stroma and inflammatory cells; and desert subtype to our Ecotype 4 enriched with Module1 tumor core. More interestingly, the prognostic features of their four subtypes were also consistent with the corresponding ecotypes with immune-enriched non-fibrotic subtype showing the best prognosis and the fibrotic subtype having the worst. This similarity further demonstrated the robustness and validity of the breast cancer ecosystem that we proposed based on digital pathology. Besides, while the transcriptomics data were not always accessible in clinical practice, our digital pathology-based classification may serve as a clinically applicable method for ecosystem-based patient stratifications.

Our study has some limitations. First, given the inherent limitation of H&E-stained WSIs, the current nuclear classification algorithms for digital pathology can only classify nuclei into the main cell types usually including tumor, stroma, inflammatory and normal nuclei, and are unable to further classify them into different functional subgroups[12,17,53,54]. This limited more in-depth characterization of the tumor ecosystem at higher biological granularity. Second, our study found that AIC score and MITH can indicate response to immunotherapy and CDK4/6 inhibitors, respectively. These findings were mainly based on the multiomics data analysis and were only validated in cohorts with limited samples. More experimental and prospectively collected evidence from large clinical cohorts should be added before their clinical translation. Third, The WSIs analyzed in our study were scanned using the NanoZoomer scanner. Limited by equipment constraints, we have not yet optimized and adapted the algorithm for slides from different scanners. Further study is needed to verify or optimize the reproducibility of sc-MTOP across the slides from different scanners, before applying it and the relevant conclusions to WSIs from other scanners.

In conclusion, our study presented sc-MTOP, an analysis framework for the characterization of tumor ecosystem based only on the routine pathological WSIs. We depicted a single-cell morphological and topological atlas, systematically characterizing the phenotypic diversity of breast cancer ecosystem. We also identified ecosystem features including AIC score, MITH and ecotypes that were associated with the response to certain treatment and patient prognosis in breast cancer. These features could potentially serve as broadly applicable markers to inform treatment decision-making and aid prognostic evaluation. The sc-MTOP framework and the large single-cell dataset with matched clinical and multiomics data may serve as a valuable resource for further investigation.

## Methods

### Ethics approval
This study was approved by the Fudan University Shanghai Cancer Center (FUSCC) Ethics Committee and complied with all relevant ethical regulations. All patients provided written informed consent.

### Data source
Our study used two cohorts. The discovery cohort is a retrospective cohort that included consecutive breast cancer patients who received standard treatment at the Department of Breast Surgery, FUSCC from January 1, 2013 to December 31, 2014 according to the following criteria: a) female patients diagnosed with unilateral invasive carcinoma of no special type; b) no evidence of distant metastasis at diagnosis; and c) treated by surgical resection without prior anti-tumor treatment. The paraffin-embedded H&E-stained tumor slides were

collected and scanned at 40× to generate digital WSIs using Nano-Zoomer digital pathology scanner (Hamamatsu Photonics, Japan). All of the slides, WSIs and the corresponding pathological reports were re-reviewed by pathologists from FUSCC and the cases with slides or WSIs of poor quality (large areas of debris, folds, pen marks or blurred regions) were excluded. The final discovery cohort encompassed 637 WSIs from 637 patients. The ER, PR, and HER2 status were independently confirmed by two pathologists at FUSCC based on immunochemical analysis and in situ hybridization and ≥1% positively stained cells was used as the cutoff for ER and PR positivity[55]. Patient clinicopathological characteristics were outlined in Supplementary Table S1. The median length of follow-up was 83.1 months (interquartile range, 70.9–90.6 months). RFS was defined as the time from diagnosis to the first relapse, a contralateral breast cancer or death due to any cause. Besides the clinical and prognostic information, next-generation DNA and RNA sequencing were performed on the tumor samples which derived from the same specimen as the tumor slides. The whole-exome sequencing (WES) data were available for 546 patients, CNA data for 555 patients and RNA-seq data for 612 patients.

Another cohort was used for the validation of our findings concerning the association between the abundance of locally aggregated inflammatory cells and immunotherapy response. This cohort derived from the NCT04129996 trial, in which patients with locally advanced or metastatic immunomodulatory TNBC (defined as CD8 IHC staining ≥10%) were enrolled and treated with a PD-1 inhibitor (camrelizumab)-based regimen as first-line treatment[22]. The primary endpoint was objective response rate assessed by investigators according to RECIST v1.1[56]. Patients who received treatment and achieved objective response (including complete response and partial response) were categorized as responders and those who did not achieve objective response (including stable disease and progressive disease) were categorized as non-responders. The secondary endpoints were PFS and OS. PFS was defined as the time from the date of the first study dose to the first recording of tumor progression or to the date of death due to any reason. OS was defined as the time from the date of the first study dose to the date of death due to any reason. 24 cases with available pretreatment biopsy specimen before the PD-1 inhibitor-based treatment were included (Supplementary Table S2). The paraffin-embedded H&E-stained WSIs were collected and reviewed by pathologists following the same criteria as the FUSCC discovery cohort.

## Single-cell morphological and topological profiling (sc-MTOP)

sc-MTOP consisted of two steps:

a. Nuclear segmentation and classification:
Hover-Net was used to locate nuclei on WSIs and predict their cell types[17]. Hover-Net is a deep convolutional neural network for simultaneous nuclear segmentation and classification on WSIs. The model pre-trained on the public PanNuke dataset was used, which can identify five nuclear types including tumor (specifically refers to breast cancer cells in this study), inflammatory, stromal, normal (non-neoplastic epithelial) and dead (necrotic) nuclei[57]. For each WSI, Hover-Net output the information on centroid, contour and nuclear type of detected nuclei for the following feature extraction.

b. Feature extraction:

We focused on tumor, inflammatory and stroma cells which were the major and functional cellular components of breast cancer ecosystem and can be accurately identified by Hover-Net[17,18]. Three categories of features were extracted for individual nuclei.

i. Morphological features were computed to describe the nuclear shape and the characteristics of contour. The coordinate of nuclear contour was used to generate a mask. Based on this mask, the following 14 morphological features were extracted:

Area, AreaBbox, CellEccentricities, Circularity, Elongation, Extent, MajorAxisLength, MinorAxisLength, Perimeter, Solidity, Curv-Mean, CurvMax, CurvMin, CurvStd.

ii. Texture features were calculated to characterize the local pixel distribution patterns within the nuclear contour. By converting the color image to grayscale and extracting the nuclear patch according to the bounding box, gray-level co-occurrence matrix (GLCM) was calculated for each nucleus and the following five features were extracted from the GLCM: ASM, Contrast, Correlation, Entropy, Homogeneity. Besides, four statistics of pixel intensity within the nuclear contour were calculated including IntensityMean, IntensityStd, IntensityMax and IntensityMin.

iii. Topological features were extracted to characterize the inter-cellular relationship at the single-cell resolution through a graph-based method. The nuclei detected by Hover-Net were denoted by vertices and the potential cellular interactions were represented by edges. In order to comprehensively study the spatial relationship between different cell types, we proposed a multi-level pairwise method for graph construction based on the igraph package. In detail, for each pair of cell types, a graph was constructed denoted as Graph_C1-C2, where C1 and C2 denoted the cell types. We focused on tumor, inflammatory and stroma cells and six graphs were constructed: Graph_Tumor-Tumor (Graph_T-T), Graph_Inflammatory-Inflammatory (Graph_I-I), Graph_Stroma-Stroma (Graph_S-S), Graph_Tumor-Inflammatory (Graph_T-I), Graph_Tumor-Stroma (Graph_T-S) and Graph_Inflammatory-Stroma (Graph_I-S). For the construction of each graph, based on the hypothesis that spatially close cells are more likely to interact with each other[58], $k$-nearest neighbor algorithm was used for edge configuration. Each cell is connected to its five nearest cell according to the Euclidean distance and the edges longer than a threshold of 100 pixel (25 μm) were deleted[59,60]. Formally, the edge set of Graph_C1-C2 can be written as follow:

$$E = \left\{ \left( V_i, V_j \right) \middle| C_{V_i} = C1, C_{V_j} = C2, V_j \in \mathrm{KNN}(V_i), D\left( V_i, V_j \right) < T \right\} \quad (1)$$

where the $C_{V_i}$ and $C_{V_j}$ denoted the types of nucleus $V_i$ and $V_j$ respectively, $D(V_i, V_j)$ denoted the Euclidean distance between $V_i$ and $V_j$, and $T$ denoted the threshold for edge configuration and was set to 100 pixel (25 μm) in our study.

After the construction of multilevel pairwise graphs, each tumor, inflammatory and stroma cell appeared in three types of graphs (e.g. each tumor cell appeared in Graph_T-T, Graph_T-I and Graph_T-S). For each cell, the topological features were computed separately for each graph it appeared and were concatenated to constitute its final comprehensive topological features.

For each type of graph, topological features included the following four categories: i) Nsubgraph, the cell number of the subgraph where the cell was located; ii) Edge length-related features: MinEdgeLength and MeanEdgeLength, the minimum and average edge lengths of a cell. If a cell was not connected to any other cells, the edge length-related features were set to 100 (pixel) which was the upper limit of distance between two connected cells. iii) Features characterizing the position of a cell in the subgraph where it was located including Degrees, Coreness, Eccentricity, Eccentricity_normed, Harmonic Centrality, Closeness, Betweenness, Betweenness_normed and ClusteringCoefficient. In all heterotypic graphs (Graph_T-I, Graph_T-S and Graph_I-S), ClusteringCoefficient for all cells were zero or not available and were therefore excluded from analysis. For cells with no edges to any other cells, Closeness and ClusteringCoefficient were set as zero. iv) For inflammatory cells, a specific

feature, StromaBlocker, was calculated to measure the barrier between the inflammatory cells and tumor cells constituted by the stroma cells.

Detailed descriptions of all morphological, texture and topological features were presented in Supplementary Table S4. An example of sc-MTOP results for one sample was provided in Supplementary Materials.

## CODEX for the validation of cell classification accuracy

We validated the accuracy of the Hover-Net model for nuclear classification using the paired H&E-stained WSIs and images of co-detection by indexing (CODEX)[39]. First, we retrieved the FFPE blocks of tumor specimen and made tissue slides of two cases. Second, CODEX experiment and imaging was performed on the PhenoCycler-Fusion system (Akoya Biosciences, Menlo Park, CA) following the standard protocol. An inventoried PhenoCycler antibody panel was used (Supplementary Table 5). After this, we washed out the flow cells on the slides and performed H&E staining. We scanned the H&E-stained slides to generate WSIs and applied the Hover-Net model to obtain the cell classification results. To examine the accuracy of Hover-Net cell classification, we marked a total of 25 matched regions (100 × 100 μm) on the WSIs and the corresponding CODEX images, quantified the number of tumor, inflammatory and stroma cells respectively and analyzed their consistency. For CODEX cell quantification, we used pan-cytokeratin as the marker for tumor cells, SMA for stroma cells and CD3e/CD20 for inflammatory cells. CODEX related image analysis were performed using QuPath (version 0.3.0)[61].

## Dataset management

For each WSI, sc-MTOP generated four dataframes. For each type of tumor, inflammatory and stroma cells, one dataframe stored the features for all cells belonging to this type and each cell was identified by a unique cell ID together with the centroid's spatial coordinates. The other one dataframe stored the edge information for this sample and characterized each edge by the connected cell IDs.

## Data processing, dimension reduction and cell clustering

The Scanpy toolkit was used for single-cell data analysis[62]. First, for the analysis of each type of tumor, inflammatory and stroma cells, 0.5% cells were sampled from each WSI and were combined into an AnnData object[63]. The AnnData object comprised a data matrix of cell features together with the additional annotations including their nuclear spatial coordinates and the information on their donor patient. Second, the values of each feature were logarithmized and scaled to unit variance using the scanpy.pp.log1p() and scanpy.pp.scale() functions respectively. Third, dimensionality reduction of the data was performed by running principle component analysis. Finally, Leiden clustering implemented by the scanpy.tl.leiden() function was performed to classify the cells based on their features[20]. UMAP was used for the visualization of cell clusters in a two-dimensional space[64].

## Cell clustering extrapolation

The inflammatory cell clustering from the FUSCC discovery cohort was extrapolated to the samples of the NCT04129996 cohort. First, for each WSI of the NCT04129996 cohort, 10% inflammatory cells were sampled and an AnnData object was generated and preprocessed as in the FUSCC discovery cohort. Second, this newly-generated external AnnData object was concatenated to the inflammatory cell AnnData object of the FUSCC discovery cohort which had saved the cell clustering results and this yielded an integrated AnnData object. Third, Leiden clustering was performed on this integrated AnnData object. In the initiation process of clustering, each cell from the FUSCC discovery

cohort was assigned to an initial cluster the same as it was in the FUSCC AnnData object, while each cell from the NCT04129996 cohort was assigned to a random initial cluster. In the alteration process of clustering, the clusters of the cells from the FUSCC discovery cohort were fixed by using the is_membership_fixed parameter. Finally, each cell from the NCT04129996 cohort would obtain a cluster result after the clustering of the integrated AnnData object.

## Sample processing for genomic DNA and total RNA extraction

The specimens used for DNA and RNA sequencing were macro-dissected fresh frozen tumor tissues. For quality control, we filtered out the samples with an excessively high proportion of stromal tissue. Genomic DNA was extracted and purified from fresh frozen samples and peripheral blood cells using TGuide M24 (Tiangen). The DNA purity and concentration were evaluated by measuring the absorbance at 260 nm (A260) and 280 nm (A280) on a NanoDrop 2000 spectrophotometer (Thermo Fisher Scientific). Only DNA samples with A260/A280 ratios between 1.6–1.9 were considered pure and used for subsequent experiments. Total RNA was extracted and purified from tissues stored in RNAlater solution using the miRNeasy Mini Kit (Qiagen #217004). RNA integrity was assessed on the Agilent 4200 Bioanalyzer (Agilent Technologies). RNA concentrations were quantified by a NanoDrop ND-8000 spectrophotometer (Thermo Fisher Scientific).

## Generation of WES data

The exome sequencing reads were aligned with BWA-mem (version 202010.02). The resulting BAM files were processed using Sentieon Genomics tools (version 202010.02)[65]. NGSCheckMate (version 1.0.0), FastQ Screen (version 0.12.0), FastQC (version 0.11.8) and Qualimap (version 2.0.0) were used to evaluate the quality of sequencing data[66–68]. Somatic mutations were determined through the following steps: First, mutations were called using three different callers, VarScan2 (version 2.4.2), TNseq (version 202010.02) and TNscope (version 202010.02)[65,69,70]. Then, spurious mutation calls caused by sequencing artifacts were filtered out. After that, only mutations consistently called by at least two of the three callers were kept. Additional filtering based on bam-readcount (https://github.com/genome/bam-readcount) was performed to reduce false positive calls, requiring mutations to meet: a) variant allele frequency ≥5%; b) sequencing depth in the region ≥ 8; c) number of mutation-supporting reads ≥4.

## Generation of CNA data

OncoScan CNA Assay (Affymetrix) was used to detect genome-wide CNAs. Chromosome Analysis Suite (ChAS, version 4.1) software (Thermo Fisher Scientific) was used for data analysis. Probe-level output of ChAS was analyzed using ASCAT (version 2.4.3) to generate segmented copy number calls and estimated tumor ploidy and purity[19]. GISTIC2.0 (version 2.0.22) was used to generate the gene level CNAs based on the ASCAT segments[71].

## Generation of RNA-seq data

RNA libraries were constructed through ribosomal RNA depletion methods using Ribo-off rRNA Depletion Kit (Human/Mouse/Rat) (Vazyme #N406) and VAHTS Universal V8 RNA-seq Library Prep Kit for Illumina (Vazyme #NR605). The libraries were sequenced on Illumina NovaSeq platforms with paired-end reads of 150 bp. The raw sequence data were demultiplexed and converted to FASTQ files with adapter and low-quality sequences quantified. Sequencing reads were aligned to the hg38 human reference genome. We obtained the Fragments Per Kilobase of transcript per Million mapped reads (FPKM) using StringTie (version 1.3.4) and Ballgown (version 2.14.149). To focus on genes with robust expression values for subsequent analysis, genes with FPKM of 0 in more than 30% samples were filtered out.

## Gene set enrichment analysis (GSEA)

GSEA was performed using the GSEA software (v4.1.0)[72]. The Canonical pathways gene sets (v7.5.1) were employed as the gene datasets. 1000 rounds of permutation were used.

## Estimation of tumor microenvironment cell abundance with RNA-seq data

Using the RNA-seq data, the ESTIMATE immune score and xCell immune score were calculated to estimate the overall abundance of immune cells. The ESTIMATE stromal score and xCell stromal score were calculated to estimate the overall abundance of stroma cells[73,74]. For the abundance estimation of more specific microenvironment cell subsets, previously established microenvironment signatures were retrieved[75], which contained 22 immune cell signatures from CIBER-SORT and one each for fibroblast and endothelial cell from MCP-counter (Supplementary Table S6)[36]. The abundance of these 24 microenvironment cell subsets was evaluated through ssGSEA ("GSVA" function in R) based on the RNA-seq data[21,75].

## Evaluation of immunotherapy response biomarkers

Biomarkers or gene signatures that were predictive of immunotherapy response were retrieved from two previous studies (Supplementary Table S7)[76,77]. Integrated cytokine score, Chemokine12 score, Module5 TcellBcell score, STAT1 signature, dendritic cell signature and B cell signature were retrieved and evaluated according to Wolf et al.[76]. Based on the WES data, tumor mutation burden was calculated using the R package "maftools" (version 2.6.05)[78]. The APOBEC mutational signature was calculated using SigProfiler[79]. T cell inflamed gene expression signature, cytolytic score and CD8 T effector signature (from the POPLAR trial) were assessed based on the RNA-seq data[34,80,81]. The expression value of individual genes including CD8A, CXCL9, CD274, CD38 and CXCL13 were directly retrieved from the RNA-seq data.

## Evaluation of tumor infiltrating lymphocytes (TILs)

TILs were manually evaluated using the WSIs used in our study according to the recommendations by the International TILs Working Group[82].

## IHC staining and evaluation

In the NCT04129996 cohort, IHC staining of CD8 and PD-L1 was performed on paraffin-embedded tissue slides that derived from the same specimen as the analyzed WSIs. CD8 IHC score was measured as the percentage of positive cells (positive cells divided by all nucleated cells)[83], while PD-L1 IHC score was measured as the proportion of tumor area (area containing viable tumor cells, their associated intratumor stroma and contiguous peritumoral stroma) occupied by PD-L1-positive immune cells[5,84]. In the FUSCC cohort, IHC staining of phospho-Rb (S807/811) was performed on paraffin-embedded tissue slides and phospho-Rb IHC score was measured as the percentage of positive tumor cells (positive tumor cells divided by all tumor cells). For all three markers, the IHC staining results were independently evaluated by two experienced pathologists who were blinded to the patients' clinical information. Discrepancies between them were resolved by discussion and consensus. The following antibodies were used: anti-CD8 (clone SP57, Ventana, #790-4460, undiluted), anti-PD-L1 (clone SP142, Abcam, #ab228462, 1:500 dilution) and anti-phospho-Rb (Ser807/811) (clone D20B12, Cell Signaling Technology, #8516, 1:200 dilution). All antibodies used were validated by their manufactures.

## Tumor nuclear morphological intratumor heterogeneity (MITH)

Based on the nuclear morphological and texture features, we measured the tumor nuclear MITH for each individual sample. First, values of each morphological and texture feature were normalized across all tumor nuclei from all samples. Then, for each individual sample, MITH was calculated as the mean value of the features' standard deviation across the tumor nuclei that belong to this sample:

$$MITH = \frac{1}{N}\sum_{i=1}^{N}\sqrt{\frac{\sum_{j=1}^{M}\left(x_{ij} - \bar{x}_i\right)^2}{M}} \quad (2)$$

where $N$ is the number of features, $M$ is the number of tumor nuclei belonging to the sample, $x_{ij}$ is the normalized value of feature $i$ of cell $j$ and $\bar{x}_i$ is the mean value of the normalized feature $i$ of the sample.

## Genetic intratumor heterogeneity

Genetic intratumor heterogeneity was measured by mutant-allele tumor heterogeneity (MATH) calculated based on the WES data[24,25].

## Multigene proliferation score (MGPS)

MGPS was evaluated using the RNA-seq data according to the previous study[26]. It was calculated as the average expression of 772 cell cycle-regulated genes identified by Whitfield et al.[85].

## E2F target signature score

E2F target signature score was calculated through ssGSEA using the MSigDB Hallmark gene sets (HALLMARK_E2F_TARGETS) which included the genes encoding cell cycle-related targets of E2F transcription factors[21,86].

## PDO collection and culture

PDOs were established from the breast tumor surgical specimens according to the protocol in the previous study[87]. Breast cancer tissue was cut into 1–3 mm³ pieces and was digested in collagenase (Sigma). The digested tissue suspension was sequentially sheared and strained over 100 μm filter. The strained suspension was centrifuged, suspended and seeded in 24-well plates with breast cancer organoid medium containing cold basement membrane extract (BME) (prepared according to Sachs et al.[87]). Plates were incubated at 37 °C/5% CO₂ in ambient O₂ conditions. Organoids in good condition were split, strained over 70 μm filter, and cultured for 5–7 days in the growth medium before drug treatment. The consistency was examined between the PDOs and the corresponding primary tumors in MITH and genomic alterations (evaluated through targeted sequencing with a well-established panel[88]).

## Drug response testing of PDOs

For drug response testing, organoids were harvested and diluted to 75 organoids/μL in the medium. 384-well plates were coated with 10 μL BME per well using a multidrop dispenser. 30 μL of the organoid suspension was then added to each well. Abemaciclib (1 μM final concentration, S5716, Selleck), PF-06873600 (0.4 μM final concentration, S8816, Selleck) and DMSO controls were then added in duplicate. After 7 days of drug treatment, cell viability was measured by adding 40 μL of CellTiter-Glo 3D Reagent (G9683, Promega) per well and reading luminescence after 5 min of shaking and 25 min of incubation in darkness at room temperature.

## Identification of micro-ecological modules (MEMs)

Based on our spatially-resolved single-cell atlas, we tried to identify recurring spatial patterns of local multicellular organization. The following analytic workflow was adopted: First, the single-cell clustering results for tumor, inflammatory and stroma cells were extrapolated to all cells from each sample using the aforementioned cell clustering extrapolation method. Second, the WSIs were tessellated into non-overlapping square domains with slide length of 200 μm, referred to as spots. All spots were characterized by the cell number of all tumor, inflammatory and stroma cell clusters. This yielded a $N_s \times N_c$ matrix

($N_s$ = number of spots and $N_c$ = number of cell clusters) for each sample. Third, spatial correlation analysis was performed to reveal the colocalization of cell clusters in a sample-wise manner. Multivariate spatial correlation algorithm developed based on Moran's autocorrelation by Wartenberg et al. was adopted and the spatial correlation between two cell clusters $x$ and $y$ was calculated as[89]:

$$C = \frac{N_s}{W} \frac{\sum_{i=1}^{N_s}\sum_{j=1}^{N_s} w_{ij}(x_i - \bar{x})(y_j - \bar{y})}{\sqrt{\sum_{i=1}^{N_s}(x_i - \bar{x})^2}\sqrt{\sum_{i=1}^{N_s}(y_i - \bar{y})^2}} \tag{3}$$

where $x_i$ and $y_j$ are cell number of cluster $x$ in spot $i$ and cluster $y$ in spot $j$ respectively, $\bar{x}$ and $\bar{y}$ denote the average cell number of these two cell clusters across all spots of the sample, $w_{ij}$ denotes the spatial weight between spots $i$ and $j$, $N_s$ is the total number of spots of the sample, and $W$ is the sum of $w_{ij}$. We assign $w_{ij} = 1$ if spot $j$ is among the nearest neighbors of spot $i$, and $w_{ij} = 0$ otherwise[90]. This analysis was performed for each pair of cell clusters and the resulting $N_c \times N_c \times N_p$ correlation matrix ($N_c$ = number of cell clusters and $N_p$ = number of patients) was reduced by mean to a $N_c \times N_c$ correlation matrix. Finally, hierarchical clustering of the correlation matrix was performed using the scipy.cluster.hierarchy.fcluster function with the Ward linkage method. MEMs were identified based on the clustering results. After the identification of MEMs, for each module, a module score was assigned to each spot which was calculated as the percentage sum of cell clusters belonging to this module. All spots were classified into the MEMs according to their maximum module score.

### Characterization of spatial patterns of MEMs using Moran's I index

To characterize the spatial variability of our defined MEMs, Moran's I index is used, which is a commonly used statistic to measure the degree of spatial autocorrelation[30]. For each module of each sample, it is calculated as:

$$I = \frac{N_s}{W} \frac{\sum_{i=1}^{N_s}\sum_{j=1}^{N_s} w_{ij}(x_i - \bar{x})(x_j - \bar{x})}{\sum_{i=1}^{N_s}(x_i - \bar{x})^2} \tag{4}$$

where $x_i$ and $x_j$ are the module scores of spots $i$ and $j$, $w_{ij}$ is spatial weight between spots $i$ and $j$, $\bar{x}$ is the mean value of the module score across all spots of the sample, $N_s$ is the total number of spots, and $W$W is the sum of all weights $w_{ij}$. We assign $w_{ij} = 1$ if spot $j$ is among the four nearest neighbors of spot $i$, and $w_{ij} = 0$ otherwise[90]. The Moran's I index value ranges from −1 to 1. A value close to 1 indicates a spatially aggregated pattern, a value close to 0 indicates a random spatial pattern, and a value close to −1 indicates a chess board-like pattern.

### Comparison between the deep learning methods and our sc-MTOP method in characterizing the tumor ecosystem

We examined two deep learning models: self-supervised learning model by Ciga et al.[31] and KimiaNet by Riasatian et al.[32]. First, we extracted the deep features from the same image "spots" as used in our MEM analysis, keeping the size, coordinates and magnification consistent. Then, we performed K-means clustering on the patches' deep features and set the cluster number to 9, matching the number of categories of our sc-MTOP-based MEM classification. Finally, we compared the yielded self-supervised learning model-based MEM (sslMEM) classification and KimiaNet-based MEM (KimiaMEM) classification with our sc-MTOP-based MEM classification in terms of the ability to characterize the locoregional cellular structures by calculating the inter-MEM and intra-MEM variations[91]. In detail, first, for each spot we calculated the proportion of tumor, stroma and inflammatory cells. Then, for each cell type, intra-MEM variation was measured as the

mean of the standard deviation of its proportion across all spots within each MEM:

$$\frac{1}{n}\sum_{j=1}^{n} \text{STD}_{i=1}^{r}(p_{i,j}) \tag{5}$$

where $n$ is the number of MEMs, $r$ is the number of spots belonging to MEM $j$, $p_{i,j}$ is the cell type proportion of spot $i$, $\text{STD}_{i=1}^{r}(\cdot)$ is an operator for calculating the standard deviation of cell type proportions across the $r$ spots belonging to the MEM $j$. Inter-MEM variation was calculated as the standard deviation of the mean proportion of the cell type of all spots belonging to the same MEM:

$$\text{STD}_{j=1}^{n}\left(\frac{1}{r}\sum_{i=1}^{r} p_{i,j}\right) \tag{6}$$

An ideal MEM classification should have both lower intra-MEM variation, indicating they can identify and summarize the recurrent patterns of local multicellular structures, and higher inter-MEM variation, indicating they can clearly distinguish different local multicellular structures. A random nine-class classification was used as a negative control for this analysis.

### Tumor ecotypes based on the MEM composition

Tumor ecotypes were identified using hierarchical clustering of the composition of MEMs. For each sample, the percentage of spots belonging to each MEM was calculated, which generated a percentage vector for the sample's MEM composition. Then, hierarchical clustering was performed based on the samples' MEM percentage vectors using the scipy.cluster.hierarchy.fcluster with the Ward linkage method. The optimal cluster number was determined according to Calinski–Harabasz index and Silhouette coefficient[92,93].

### Somatic mutation and CNA associations with breast cancer ecotypes

To investigate the association between somatic mutations and breast cancer ecotypes, we compared the frequency of somatic alterations in oncogenic pathways among the four ecotypes (R OncogenicPathways from package maftools)[94]. P values were calculated by Chi-Square Test. As for CNA, each level of averaged event was converted into ordinal scale. Ordinal regression model (R clm from package ordinal) with proportional odds link function was fitted by using the tumor ecotypes as a categorical independent variable and the ordinal scale as the dependent variable[52]. Log odds ratio was used to measure the effect of ecotypes on CNA events.

### Development of a breast cancer ecotype classifier

Our ecosystem classification was based on hierarchical clustering of the 637 samples of the FUSCC cohort. To realize the identification of tumor ecotypes of new samples, we developed a breast cancer ecotype classifier based on the MEM composition data. We first split the 637 samples into a training set (80%) and a test set (20%). Stratified sampling was adopted to balance the proportions of the four ecotypes between the training and test sets. Then, we trained an SVM classifier in the training set and evaluated its classification accuracy in the test set using the weighted F1 score[95].

### Online platform for data management and real-time analysis

To facilitate data sharing and the use of sc-MTOP framework by other researchers, an online platform was developed (http://sc-mtop.biosolver.cn/). The platform comprises two modules. The first is a data management module for digital pathology and multiomics data, which is developed based on cbioportal[96]. The uploaded digital WSIs of the FUSCC cohort are available for online view under up to ×40 magnification using a digital pathology viewer developed with the

OpenSeadragon v2.4.2 package. Besides the WSI data, the sc-MTOP raw data together with the clinical and multiomics data are also correlated and shown aside. The mongodb and mysql databases are used to store these comprehensive data and manage their relationships for integrated online presentation. The second module of the platform is an online analysis module for real-time analysis of newly-uploaded WSIs. We have designed a plugin system to make the entire sc-MTOP framework and the subsequent analytic pipeline modularized with Clojure language. After the upload of WSI, the sc-MTOP framework will be executed on a cloud server and the output results can be downloaded. In addition, an ecosystem portrait is generated for the sample by evaluating the clinically relevant ecosystem features including the cellular composition, AIC score, MITH and the tumor ecotypes. A user-friendly interface is developed based on the Vue.js framework and JavaScript language[97]. A detailed data policy will be made and uploaded to the website when the platform fully opens to the public.

## Statistical analysis

Statistical analysis was performed using Python (version 3.8.12) and R (version 3.6.3). Spearman correlation was used to investigate the relationship between continuous variables. Continuous variables were compared between groups using the Mann-Whitney U test with false discovery rate (FDR)-correction for multiple testing[98]. All tests were two-sided, and an FDR-corrected $P < 0.05$ was considered statistically significant. Predictive accuracy of biomarkers was measured by ROC analysis using AUC. Survival curves were plotted using the Kaplan-Meier method and survival differences were compared between groups using the log-rank test and Cox proportional hazards models.

## Reporting summary

Further information on research design is available in the Nature Portfolio Reporting Summary linked to this article.

## Data availability

All WSIs and sc-MTOP data of the FUSCC cohort have been uploaded to the sc-MTOP platform (http://sc-mtop.biosolver.cn/). Example data of sc-MTOP results for one sample are provided in Supplementary Data 1. Raw WES, CNA and RNA-seq data have been deposited on the GSA database with GSA-Human accession number HRA005104. The data are available under controlled access for research purposes only. Access can be obtained by approval via the Data Access Committee of the GSA-human database (for detailed instructions, please refer to: https://ngdc.cncb.ac.cn/gsa-human/document/GSA-Human_Request_Guide_for_Users_us.pdf). The approximate response time for accession requests is four weeks. Once the access has been granted, the data will be available for one week and should be used for research purposes only. Source data are provided with this paper.

## Code availability

The source codes of sc-MTOP are provided in Supplementary Software and are available at Github (https://github.com/fuscc-deep-path/sc_MTOP) and Zenodo (https://doi.org/10.5281/zenodo.8364420)[99].

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

## Acknowledgements

This work was supported by grants from the National Key Research and Development Program of China (Grant number 2021YFF1201300 to Y.Z.J.), the National Natural Science Foundation of China (Grant numbers 82103039 to S.Z., 91959207 and 92159301 to Z.M.S.), the Shanghai Key Laboratory of Breast Cancer (12DZ2260100), the SHDC Municipal Project for Developing Emerging and Frontier Technology in Shanghai Hospitals (SHDC12021103) and China Postdoctoral Science Foundation (Grant number 2022M720790 to S.Z.).

## Author contributions

Y.Z.J., Z.M.S., J.X. and W.T.Y. designed the study. S.Z., D.P.C. and T.F. collected materials, analyzed the data and wrote the manuscript. S.Z., D.P.C. and J.C.Y. developed the algorithm and built the sc-MTOP platform. D.M., X.Z.Z., X.X.W., Y.P.J., X.J., Y.X., W.X.X., H.Y.Z., H.L. and A.M. participated in data analysis and manuscript revision. All authors approved the manuscript. S.Z., D.P.C., T.F. and J.C.Y. contributed equally to this work.

## Competing interests

The authors declare no competing interests.
