## [Peer Review File · Nature Communications]

REVIEWER COMMENTS

Reviewer #1 (Remarks to the Author): Expert in artificial intelligence, digital pathology, and computational genomics

Summary:

This work proposes an analysis of breast cancer histopathology slides based on cellular structures. The authors combine morphological and topological features of each cell and apply hierarchical clustering to identify different sub-clusters of inflammatory, tumor, and stromal cells. Afterwards, they demonstrate that various molecular subtypes of breast cancer are enriched with different proportions of the identified sub-clusters. Furthermore, they try to link this information to patient outcome and genomic information.

Overall, this reviewer commends the extensive work and validations that have been carried out. Below, I have provided some recommendations to improve the manuscript.

Major Comments:

- 1- I think the major weakness of the paper in the ecotype section is its comparison to other deep learning-based models. Are the current state-of-the-art patch-based models (e.g., Attention MIL, etc.) able to capture the same ecotype information? To be more specific, what happens if the authors extract deep learning-based features from the patches and then do the same analysis? I think the work will benefit from such an analysis on one of the tasks that have been performed in the paper. This could be a good starting point: <https://arxiv.org/pdf/2011.13971.pdf>
- 2- Analysis on Page 15: majority of this analysis (e.g., linking to clinical parameters) is performed on all the samples together. However, I believe the manuscript will be enhanced if this analysis is repeated within the context of the major molecular subtypes of breast cancer.
- 3- Lines 682-685, authors need to explain how each of the features are calculated.
- 4- Page 5: is there a reason why the authors have excluded necrotic cells? Necrotic cells have been shown to correlate with cancer outcomes.
- 5- I believe the manuscript can benefit from some editorial revisions (e.g., to improve clarity on some places or fix some minor grammatical issues). As an example, page 13, some of the details can be moved to the methods section.

Reviewer #2 (Remarks to the Author): Expert in breast cancer tumour microenvironment and digital pathology

In this paper Zhao and others develop the scMTOP framework to characterize individual cells on WSIs by extracting nuclear morphological and topological features from breast cancer. They use a combination of AI-technology on WSI's, gene expression, mutation and clinical data. The idea of just using the information of an HE slide is good, and widely applicable. However, there are a few pitfalls in the study design and interpretation of the data. Please find below suggestions.

1. The cohort could be better described. What type of breast cancer, collected in which setting? How can the results help in clinical translation? What is the aim of the paper, purely method development or also to answer clinical/biological question?
2. The authors propose to cluster a number of features and then cluster these clusters again (to create MeMs) and then cluster these clusters again (to generate ecotypes). The authors do not provide an explanation of this approach and it is also not clear what method is used for clustering. As this is such a vital part of the work this should be mentioned and explained. Especially why clustering of clustering makes sense, would not it be better to try to relate cell morphological and topological features immediately to subtypes/ecotypes or outcomes?
3. "Based on scMTOP we aimed to dissect the diversity of the breast cancer ecosystem at multiple levels" (line 170). Although there is a wealth of information in WSI combined with RNAseq data, there are also many other technologies around which can give a more profound picture of the breast cancer ecosystem. Think of single cell transcriptomics of different tissue components (tumor, stroma, infiltrate), multiplex immunohistochemistry technologies like Cytof, Codex or Nanostring GeoMx/CosMx. These would give you information on the cellular level about specific immune or other cell sets. The authors could have discussed more the advantages and disadvantages of their study in light of other novel technological developments.
4. How was the RNAseq done? Was it enriched for tumor cells? It is always problematic to compare the single cell results (scMTOP) with bulk sequencing (RNAseq). Especially in heterogeneous samples.
5. Line 212: "...may serve as a biomarker for immunotherapy response." I think this is too much speculation, and too much indirect assumptions.
6. The data on NCT04129996 should be presented with caution. Numbers are low and no multivariate analysis has been performed.
7. Throughout the paper the analyses of the various clusters (infiltrate, tumor, stroma) there are only associations assessed with the IHC subtypes. It would be interesting to see also associations with other clinical variables, like age, grade, histology.
8. Line 284: please include 'models' in the subheading, as the CDK inhibitor data is on breast cancer models and not on patients.

9. It is currently unclear how the MITH score is calculated. Figure 5a and the formula tell the same story, however, in the text it says that the features' variance is normalized. Therefore, the standard deviation would also be normalized and the formula doesn't make sense.
10. Line 428: For the ecotype classifier the authors build an SVM classifier but do not show how to use it and why they choose an SVM classifier as it is quite an old method.
11. The results on prognosis seem counter-intuitive, as the Ecotype 1 has most ER+ patients and a poor prognosis. In general ER+ breast cancer has a better prognosis than TN or HER2+. Also many ER+ were N+. Could this be a confounder? In addition, some more analyses with associations on clinical variables would be appreciated. Also a better description of the cohort would be good. What is the effect of treatment? Why is grade not taken into account into the multivariate model for the ecotypes?
12. The authors show a lot of correlation values in the results and figures. However, they often do not provide confidence intervals and thus we cannot see if the correlations are significantly different from 0.
13. The authors mention that their pipelines are unsupervised. When in fact the biggest part of the models are supervised models
14. Were all WSI scanned by one scanner? How does the method perform on other scanners? And how is the reproducibility on a second slide of the same patient?

Reviewer #3 (Remarks to the Author): Expert in breast cancer tumour microenvironment, digital pathology, and single-cell and spatial omics

In this study, the authors developed an algorithm which uses whole slide images (WSI) to identify malignant, immune and stromal cells, using the texture and topological features of each single cell / nucleus. The authors further divide these three main cell types into more distinct spatially relevant cells. Then identify on the WSI micro-ecological modules (MEM) which through unsupervised clustering form tumor ecotypes. Authors emphasize that their algorithm is clinically relevant through the association of ecotypes with prognostics or by showing that the spatially relevant cell types may guide response to therapies.

Major comments

- The main validation, one would have liked to see is the accuracy of the algorithm to classify immune, stromal and malignant cells using paired HE stains and either spatial transcriptomics or proteomics.

- What about endothelial cells and adipocytes? they are an important part of breast tumors. However, the authors barely mention them. Would these be counted as stromal cells? if yes are they making a specific STR subtype?
- What about the polynuclear immune cells? I would have thought these would had been possible to identify on the WSI, as they have a specific shape. Would these make a specific INF subtype?
- The authors use organoids to investigate whether the MITH is associated with resistance to CDK4/6 inhibitors. Since the authors also have WES data it would have been interesting to potentially associate MITH with genomic heterogeneity or the resistance to CDK4/6 inhibitors to genomic alterations.
- When the authors use organoids to investigate CDK4/6 inhibitors it is necessary to show if such organoids have the same tumor / genetic alterations than the primary tumors. Can the same MITH seen at the patient level been reproduced on PDOs?
- It is surprising that only five of the INF subtypes (1,7,0,6, 3) in Figure 3A correlate with immune cell types, what are the others IFN subtypes correlated with? I understand that the correlation is made only with immunomodulatory subtypes, but it would be good to understand what the other are 'made of'?
- It seems that the ecotypes and the MEMs are very correlated (Fig7a), while the authors associate the ecotypes with prognosis it would had been interesting to see the same for the individual MEMs and single cell subtypes, it could help to understand how the ecotypes are linked to prognosis, indeed while the cox analysis show an independent prognostic value, it is not explained or hypothesized how/why that may be.
- Is it complicated to link ecotypes to IFN, TUM and STR? would it be possible to retrieve the cellular subtypes composition of each ecotype?
- Is the step of MEM necessary to obtain ecotypes? Going from 8 modules to 4 ecotypes through unsupervised clustering almost seem unnecessary?
- It would had been super interesting to deconvolute the RNAseq data with CIBERSORTx and a reference of well annotated scRNA-seq data to understand if the IFN, STR and TUM correspond/correlate with to some more granular cell phenotypes, if it is the case all the biological downstream interpretation of prognosis and response to treatment would had been easier and had made much more sense.

Minor comments

- I would not call, malignant or cancer cells tumor cells, is not any cell coming for a tumor a tumor cell?

Reviewer #1 (Remarks to the Author): Expert in artificial intelligence, digital pathology, and computational genomics.

This work proposes an analysis of breast cancer histopathology slides based on cellular structures. The authors combine morphological and topological features of each cell and apply hierarchical clustering to identify different sub-clusters of inflammatory, tumor, and stromal cells. Afterwards, they demonstrate that various molecular subtypes of breast cancer are enriched with different proportions of the identified sub-clusters. Furthermore, they try to link this information to patient outcome and genomic information.

Overall, this reviewer commends the extensive work and validations that have been carried out.

Below, I have provided some recommendations to improve the manuscript.

Response: We sincerely appreciate your recognition and valuable suggestions on our manuscript. Following your recommendations, we have conducted additional analysis and revised our manuscript. The revisions have been marked in red in the file “Revised Manuscript”. Here are our point-by-point responses to your comments:

Comment 1. I think the major weakness of the paper in the ecotype section is its comparison to other deep learning-based models. Are the current state-of-the-art patch-based models (e.g., Attention MIL, etc.) able to capture the same ecotype information? To be more specific, what happens if the authors extract deep learning-based features from the patches and then do the same analysis? I think the work will benefit from such an analysis on one of the tasks that have been performed in the paper. This could be a good starting point: <https://arxiv.org/pdf/2011.13971.pdf>

Response: Following your advice, we made a comprehensive comparison between the patch-based deep learning method and our sc-MTOP-based method in capturing the tumor ecotype information and providing clinically relevant information. We extracted deep features from

the same image patches, termed “spots” in our method, using the pre-trained ResNet model proposed in the paper you suggested. This model was developed specifically for image feature extraction through self-supervised contrastive learning on a large collection of pathology images spanning multiple organs and cancer types¹. Then, we performed unsupervised clustering on patches’ deep features and also identified nine deep learning-based MEMs (DeepMEMs) for comparison (Fig. R1a). The results were summarized as follows:

a) sc-MTOP-based MEMs outperformed DeepMEMs in representing the recurrent patterns of locoregional cellular structures in tumor ecosystem:

We quantified intra-MEM variation and inter-MEM variation to examine which of sc-MTOP-based MEMs and DeepMEMs can better represent the multicellular structures in the tumor ecosystem². In detail, first, for each spot we calculated the proportion of tumor, stroma and inflammatory cells. Then, for each cell type, intra-MEM variation was measured as the mean of the standard deviation of its proportion across all spots within each MEM:

$$\frac{1}{n} \sum_{j=1}^n STD_{i=1}^r(p_{i,j})$$

where n is the number of MEMs, r is the number of spots belonging to MEM j , $p_{i,j}$ is the cell type proportion of spot i , $STD_{i=1}^r(\cdot)$ is an operator for calculating the standard deviation of cell type proportions across the r spots belonging to the MEM j . Inter-MEM variation was calculated as the standard deviation of the mean proportion of the cell type of all spots belonging to the same MEM:

$$STD_{j=1}^N \left(\frac{1}{r} \sum_{i=1}^r p_{i,j} \right)$$

An ideal MEM classification should have both lower intra-MEM variation, indicating they can identify and summarize the recurrent patterns of local multicellular structures, and higher inter-MEM variation, indicating they can clearly distinguish different local multicellular structures.

It was found that compared with the DeepMEMs, sc-MTOP-based MEMs showed high inter-

MEM variations and low intra-MEM variations (Fig. R1b, also see Supplementary Fig. S10b), which indicated that sc-MTOP-based MEMs can better represent the recurrent locoregional multicellular structures.

b) sc-MTOP-based MEMs show higher consistency with histological patterns and can better portray the diversity within the same histological category.

In our original manuscript, we demonstrated that sc-MTOP-based MEMs showed high congruency with histological patterns and can portray the diversity within the same histological category (Fig. 6e-g). According to your suggestion, we performed the same analysis on DeepMEMs. Compared with sc-MTOP-based MEMs, DeepMEMs showed less consistency with histological regions (Fig. R1c-d, also see Supplementary Fig. S10c-d). Specifically, each histological region annotated by the pathologists had evident correspondence with one or two sc-MTOP-based MEMs, while there was no such correspondence between histological regions and DeepMEMs. In addition, sc-MTOP-based MEMs can better characterize the difference between tumor region 1 and tumor region 2 than the DeepMEMs.

c) sc-MTOP-based MEMs are prognostic relevant.

In sc-MTOP-based MEMs, high percentage of Module8_IA was significantly associated with a better prognosis ($P = 0.013$), while high percentage of Module4_TS was significantly associated with a worse one ($P < 0.001$) (Fig. R1e, also see Supplementary Fig. S10e). By comparison, no DeepMEMs showed a significant association with patient prognosis (Fig. R1f, also see Supplementary Fig. S10f).

In summary, compared with the deep learning method, MEMs identified by our sc-MTOP-based method can better represent the patterns of local multicellular structures in tumor ecosystem and provide prognostic relevant information. We have incorporated these data in our Revised Manuscript (page 15, line, 7-24)

Fig. R1. Comparison between the deep learning-based method and our sc-MTOP-based method in capturing ecotype information and providing clinical relevant information.

a. Diagram for the comparison between the deep learning-based MEMs and our sc-MTOP-based MEMs.

b. The intra-MEM variation and inter-MEM variation of the tumor, inflammatory and stroma cell proportions according to the sc-MTOP-based MEMs and DeepMEMs.

c. Sankey plot showing the correspondence between histological patterns annotated by pathologists (middle), sc-MTOP-based MEMs (upper) and DeepMEMs (lower).

d. Thumbnail of a whole slide image with manually histological annotation (left), sc-MTOP-based MEM mapping (middle) and DeepMEM mapping (right). In histological annotation, a pathologist manually delineated the regions of different histological patterns including two types of tumor regions (red and purple), stroma (blue), immune cell aggregation in stroma (green) and immune infiltration in tumor (yellow). Scale bar: 4 mm.

e-f. Forest plot showing the univariate Cox regression analysis of relapse-free survival for e) sc-MTOP-based MEMs and f) DeepMEMs modeled as log-transformed module percentages. Squares and whiskers represent point estimates and the 95% confidence interval of hazard ratios.

Abbreviations: WSI, whole slide image; sc-MTOP, single-cell morphological and topological profiling; MEM, micro-ecological module; RFS, recurrence-free survival.

Comment 2. Analysis on Page 15: majority of this analysis (e.g., linking to clinical parameters) is performed on all the samples together. However, I believe the manuscript will be enhanced if this analysis is repeated within the context of the major molecular subtypes of breast cancer.

Response: Thanks for your suggestion. We have repeated the analysis of linking ecotypes to clinical parameters within each IHC subtypes and presented the results in our Revised Manuscript

Fig. R2. Association between breast cancer ecotypes and tumor T category, N category and grade within each IHC subtype.

P values are calculated using the chi-square test. *, *P* < 0.05; ns, not significant.

(page 17, line 1-5):

Subgroup analysis according to IHC subtypes showed that only in the HR-HER2+ subgroup, Ecotype 1 and Ecotype 3 breast cancers were significantly associated with higher rates of lymph node metastasis. The higher tumor grade of Ecotype 2 and Ecotype 4 breast cancers was most pronounced in HR+HER2- and TNBC subgroups (Fig. R2, also see Supplementary Fig. S12c).

Comment 3. Lines 682-685, authors need to explain how each of the features are calculated.

Response: We have added the algorithm that we used for the calculation of each feature in Supplementary Table S4.

Comment 4. Page 5: is there a reason why the authors have excluded necrotic cells? Necrotic cells have been shown to correlate with cancer outcomes.

Response: First, necrotic cells are not the major and functional cellular components of breast cancer ecosystem³. Second, unlike tumor, inflammatory and stroma cells, necrotic cells often undergone karyorrhexis and karyolysis, which makes it difficult to accurately locate them through nuclear segmentation. Several other studies on single-cell digital pathology also did not include necrotic cells in their analysis⁴⁻⁶. We have added the following sentence to explain this issue in our revised manuscript (page 26, line 27-29):

We focused on tumor, inflammatory and stroma cells which were the major and functional cellular components of breast cancer ecosystem and can be accurately identified by Hover-Net^{3,7}.

Comment 5. I believe the manuscript can benefit from some editorial revisions (e.g., to improve clarity on some places or fix some minor grammatical issues). As an example, page 13, some of the details can be moved to the methods section.

Response: Thanks for your suggestions. We have provided a concise summary of the MEM

identification methods in the Results section and relocated the detailed descriptions to the Methods section. (Revised Manuscript, page 14, line 1-4)

We have also carefully read through the paper and made some editorial revisions to improve the clarity of expression. For example, we included the word “models” to the subheading of the morphological intratumor heterogeneity section to make its meaning more precise (Revised Manuscript, page 11, line 28).

Reviewer #2 (Remarks to the Author): Expert in breast cancer tumour microenvironment and digital pathology

In this paper Zhao and others develop the scMTOPT framework to characterize individual cells on WSIs by extracting nuclear morphological and topological features from breast cancer. They use a combination of AI-technology on WSI's, gene expression, mutation and clinical data. The idea of just using the information of an HE slide is good, and widely applicable. However, there are a few pitfalls in the study design and interpretation of the data. Please find below suggestions.

Response: We are very grateful for your recognition of our idea of using H&E-stained slides to explore breast cancer ecosystem and your professional critiques on our manuscript. We highly value your concerns regarding the study design, data analysis and interpretation and have revised our manuscript to clarify the study cohort, study aim, rationale of analytic procedures and interpret our data with caution. We have also studied all other comments, performed additional analysis and incorporated the results into our Revised Manuscript. The revisions have been marked in red in the file “Revised Manuscript”. Here are our point-by-point responses to your comments:

Comment 1.1 The cohort could be better described. What type of breast cancer, collected in which setting?

Response: We specified the histological types of breast cancers, the settings how cases were

collected, and the inclusion/exclusion criteria in the Methods section (Revised manuscript, page 25, lines 3-9):

The discovery cohort is a retrospective cohort that included consecutive breast cancer patients who received standard treatment at the Department of Breast Surgery, Fudan University Shanghai Cancer Center (FUSCC) from January 1, 2013 to December 31, 2014 according to the following criteria: a) female patients diagnosed with unilateral invasive carcinoma of no special type; b) no evidence of distant metastasis at diagnosis; and c) treated by surgical resection without prior anti-tumor treatment.

We added a brief cohort description to the Results section to help readers better understand the study population (Revised manuscript, page 5, lines 25-29):

..., we applied sc-MTOP to 637 whole slide images (WSIs) of a retrospectively collected cohort with multiomics profiling data. This cohort included 637 consecutive breast cancer patients who received standard treatment at Fudan University Shanghai Cancer Center (FUSCC) from January 1, 2013 to December 31, 2014.

Comment 1.2 How can the results help in clinical translation?

Response: We identified ecosystem features including AIC score, MITH and ecotypes that were associated with the response to certain treatment and patient prognosis in breast cancer. These features could potentially serve as broadly applicable markers to inform treatment decision-making and aid prognostic evaluation. We have clarified the potential translational significance of our results in Revised Manuscript (Revised Manuscript, page 24, lines 16-20):

...We also identified ecosystem features including AIC score, MITH and ecotypes that were associated with the response to certain treatment and patient prognosis in breast cancer. These features could potentially serve as broadly applicable markers to inform treatment decision-making and aid prognostic evaluation.

Comment 1.3 What is the aim of the paper, purely method development or also to answer

clinical/biological question?

Response: We have clarified the aim of our study in our revised manuscript (Revised manuscript, page 4, lines 4-6):

Our study aims to develop a digital pathology-based approach to characterize the tumor ecosystem at the single-cell resolution and use this approach to investigate the phenotypic diversity of human breast cancer ecosystem and its clinical relevance.

Comment 2.1 The authors propose to cluster a number of features and then cluster these clusters again (to create MeMs) and then cluster these clusters again (to generate ecotypes). The authors do not provide an explanation of this approach and it is also not clear what method is used for clustering.

Response: The Leiden algorithm, an unsupervised clustering method recommended for single-cell data clustering, was used to categorize cells into cell clusters based on their features^{8,9}. Hierarchical clustering was used to generate micro-ecological modules (MEMs) based on the spatial correlation matrix of cell clusters, and to generate ecotypes based on the samples' MEM composition. We have described the clustering methods used in our revised manuscript (Revised manuscript, page 7, line 17-19; page 14, line 1-4; page 15, line 30; page 29, line 26-27; page 33, line 24-26; page 35, line 9-15).

Comment 2.2 As this is such a vital part of the work this should be mentioned and explained. Especially why clustering of clustering of clustering makes sense.

Response: We have discussed the rationality and significance of performing multiple rounds of clustering in Discussion of our Revised Manuscript (page 21, line 5-17):

In this study, we applied the sc-MTOP algorithm to 637 WSIs and obtained raw feature data over 410 million cells. To comprehensively dissect the phenotypic diversity of breast cancer ecosystem, we adopted an analysis workflow mainly based on clustering analysis, which generated

visualizable and biologically interpretable data including cell clusters, MEMs, and ecotypes from the primitive, abstract and massive feature data. These generated data enabled the characterization of the phenotypic diversity of breast cancer ecosystem at multiple levels. First, at the single cell level, the cell clusters generated by feature clustering characterized the phenotypic diversity of tumor, inflammatory and stroma cells respectively. Second, the MEMs obtained through cell cluster co-localization analysis can achieve a more intuitive and effective representation of local areas and can be used to map the spatial diversity of tumor ecosystem. Finally, the ecotypes identified through the clustering of samples' MEM composition reflected the inter-patient ecosystem diversity of human breast cancer.

Comment 2.3 would not it be better to try to relate cell morphological and topological features immediately to subtypes/ecotypes or outcomes?

Response: We tried to directly use cell features to generate ecotypes and compared this feature-based ecotype classification to our MEM-based ecotype classification in dissecting ecotype diversity and stratifying patient outcomes. We first obtain patient-level features by averaging cell-level features of tumor, stromal and inflammatory cells respectively, which is commonly used method for feature aggregation^{4,10}. This yielded 172 features for each patient. Then, we applied the hierarchical clustering, which is also used to derive the MEM-based ecotypes, to cluster patient-level feature profiles to generate feature-based ecotypes. The recommended number of clusters was reduced to two according to the Calinski-Harabasz index and Silhouette coefficient (Fig. R3a, refer to Supplementary Fig. S11a for comparison). While feature-based Ecotype 1 showed some correspondence to the MEM-based Ecotype 2, feature-based Ecotype 2 encompassed a mixture of all four MEM-based ecotypes (Fig. R3b-c). In addition, survival analysis showed that feature-based ecotype classification cannot stratify patient outcomes (Log-rank $P = 0.411$) (Fig. R3d, refer to Fig. 7d-e for comparison). In summary, we believe that our MEM-based ecotype classification was superior to feature-based ecotype classification in

characterizing the ecotype diversity and stratifying patient prognosis.

Comment 3. “Based on scMTOPT we aimed to dissect the diversity of the breast cancer ecosystem at multiple levels” (line 170). Although there is a wealth of information in WSI combined with RNAseq data, there are also many other technologies around which can give a more profound picture of the breast cancer ecosystem. Think of single cell transcriptomics of different tissue

Fig. R3. Feature-based ecotype classification and its comparison to MEM-based ecotype classification.

a. Selection of cluster number for hierarchical clustering based on Calinski-Harabasz index and Silhouette coefficient.

b. Hierarchical clustering of patient-level feature profiles across 637 patients identifies two feature-based ecotypes.

c. Sankey plot showing the correspondence between the feature-based ecotype classification and MEM-based ecotype classification.

d. Kaplan-Meier curves of recurrence-free survival according to the feature-based ecotypes.

Abbreviations: MEM, micro-ecological module; RFS, recurrence-free survival.

components (tumor, stroma, infiltrate), multiplex immunohistochemistry technologies like Cytof, Codex or Nanostring GeoMx/CosMx. These would give you information on the cellular level about specific immune or other cell sets. The authors could have discussed more the advantages

and disadvantages of their study in light of other novel technological developments.

Response: Thanks for your suggestions. We surveyed other technologies for studying tumor ecosystem including single-cell RNA-seq, multiplex immunohistochemistry and multiplex in situ molecular imaging. **In the Discussion section, we discussed the advantages and disadvantages of our sc-MTOP method compared to the above technologies:**

Over the last few years, there has been growing interest in analyzing the tumor ecosystem using different technologies. Based on the bulk RNA-seq, computational methods have been developed to provide rough estimates of cell types within the tumor ecosystem^{11,12}. Single-cell RNA-seq, on the other hand, provides transcriptional profiles of individual cells, enabling the characterization of the tumor ecosystem in terms of the composition, transcriptional patterns and functional states of microenvironmental cells. However, a major limitation of this technology is the loss of cellular spatial information to understand cellular organizations and interactions within the tissue context¹³. More recently, several technologies have been developed for the gene expression measurement in a spatially-resolved manner. Multiplex immunohistochemistry technologies such as image mass cytometry and CODEX allowed targeted detection of specific proteins in individual cells in situ^{14,15}. Nanostring GeoMx/CosMx technology permitted simultaneous RNA and protein expression detection and increased the number of markers that can be detected¹⁶. These methods can provide important insight into how individual cells of specific types and functional properties organized in space to form the tumor ecosystem^{17,18}. However, the high cost, material requirements and technological complexity limited their use in large cohort and clinical implementation. By comparison, our sc-MTOP framework are unable to classify cells into specific functional subsets with molecular annotations, but it provides a new perspective for characterizing the tumor ecosystem, which focuses on the nuclear morphological and spatial topological features. It can be easily applied to the entire WSIs to realize panoramic

characterization of tumor ecosystem at a single-cell resolution with the cell spatial information preserved. In addition, sc-MTOP only requires broadly available H&E-stained tumor slides as raw materials, which facilitated its validation and optimization in large clinical cohorts and the potential implementation in clinical practice at a much lower cost.

Comment 4.1 How was the RNAseq done? Was it enriched for tumor cells?

Response: We have described the RNA-seq method in detail in the Supplementary Methods.

In brief, fresh frozen tumor tissues were macro-dissected and subjected to RNA extraction using miRNeasy Mini Kit. Ribosomal RNA depletion methods were used for library construction. The libraries were then sequenced on Illumina NovaSeq platforms. **We did not perform any experimental procedures to enrich the tumor cells in the sequencing samples.**

Comment 4.2 It is always problematic to compare the single cell results (scMTOP) with bulk sequencing (RNAseq). Especially in heterogeneous samples.

Response: We will clarify the rationality of our analysis integrating the sc-MTOP data and bulk RNA-seq data from the following two perspective:

a) Our analysis focused on the inter-patient variation, which is not affected by the difference in detection resolution between the sc-MTOP and bulk RNA-seq methods.

We agree with your opinion that direct comparison between single-cell data and bulk RNA-seq data is problematic in some cases, because bulk RNA-seq averages gene expression across the tissue, which obscures the intra-sample gene expression heterogeneity across cell types¹⁹. Nevertheless, when the analysis focused on inter-patient variations, the results of bulk RNA-seq are often comparable with single cell results²⁰. Our analysis of the correlation between inflammatory cell clusters from sc-MTOP and mRNA signatures from bulk RNA-seq focused on the variation of INF clusters across patients and its correlation with patient-level signatures. Thus, we believe that the results are not affected by the difference in detection resolution between the

two methods or the presence of intra- sample heterogeneity.

b) Previous studies provided the rationale for associating digital pathology data and bulk RNA-seq data.

Several previous studies correlated the digital pathology data with bulk RNA-seq data. Lu et al. identified several cellular morphology-related features with prognostic implications and investigated the features' molecular underpinning by correlating them with biological pathway scores based on the bulk RNA-seq data²¹. Failmezger et al. developed two stromal cell-related features based on digital pathology and revealed their association with immunosuppression and the molecular basis through differential gene expression analysis of bulk RNA-seq data⁴. Desbois et al. developed a gene expression-based classifier for immune phenotypes by comparing the bulk RNA-seq data with the quantitative metrics of T cells based on digital pathology²². These studies indicated that correlating digital pathology data with bulk RNA-seq data is feasible and can yield valuable insights.

Comment 5. Line 212: "...may serve as a biomarker for immunotherapy response." I think this is too much speculation, and too much indirect assumptions.

Response: We have revised this sentence as follows (Revised Manuscript, page 9, line 1-4):

These results suggested that high abundance of locally aggregated inflammatory cells was associated with the overall activation of tumor immune microenvironment and possibly with a favorable response to immunotherapy in breast cancer.

We believe that this revised sentence is a reasonable summary and assumptions based on the existing data.

Comment 6. The data on NCT04129996 should be presented with caution. Numbers are low and no multivariate analysis has been performed.

Response: We performed multivariate analysis including variables of patient age, disease status (recurrent/metastatic or *de novo* stage IV), visceral metastases (yes or no) and our AIC score (categorized as high or low according to the median). The analysis showed that high AIC score remained associated with better PFS and OS ($P = 0.007$ for PFS and 0.033 for OS) (Fig. R4, also see Supplementary Fig. S5).

We agree that the data from trial NCT04129996 are limited by the small sample size and should be interpreted with caution. We have modified the descriptions of the conclusions related to AIC scores in the Results section and mentioned this sample size limitation in the Discussion section:

(In Results) Therefore, in this cohort of limited sample size, our data suggested that the high abundance of locally aggregated inflammatory cells can indicate better responsiveness to immunotherapy in patients with TNBC. (Revised Manuscript, page 9, line 25-28)

(In Limitation) Our study found that AIC score and MITH can indicate response to immunotherapy and CDK4/6 inhibitors, respectively. These findings were mainly based on the multiomics data analysis and were only validated in cohorts with limited samples. More experimental and prospectively collected evidence from large clinical cohorts should be added before their clinical translation. (Revised Manuscript, page 24, line 3-7)

Figure R4. Multivariate Cox analysis of a) PFS and b) OS in the NCT04129996 cohort. Abbreviations: AIC score, aggregated inflammatory cell abundance score; PFS, progression-free survival; OS, overall survival.

Comment 7. Throughout the paper the analyses of the various clusters (infiltrate, tumor, stroma) there are only associations assessed with the IHC subtypes. It would be interesting to see also associations with other clinical variables, like age, grade, histology.

Response: We investigated the associations between the INF, TUM, and STR clusters and clinical variable including age, tumor T category, N category and grade (Fig R5, also see Supplementary Fig. S3, S6c and S7c) (histology was not analyzed because we only included patient with invasive carcinoma of no special type in our cohort). We have incorporated the results in our revised manuscript:

Breast cancers of low T category had more INF7 cells. Grade I breast cancers had more INF2, INF3, INF4, INF5 and INF9 cells, while those of grade II-II were enriched for INF0, INF1 and INF6 cells, which indicated more locally aggregated inflammatory cells. (Revised Manuscript, page 8, line 10-13)

Elderly patients had more TUM8 and TUM10 cells. TUM0, TUM3, TUM5 TUM10 cells were enriched in the grade I breast cancers, while TUM1, TUM4, TUM6, TUM7, TUM9, and TUM11 cells were enriched in grade II-III breast cancers. (Revised Manuscript, page 10 line 29 to page 11 line 3)

STR4, STR5, STR6, STR7 and STR8 cells were enriched in elderly patients, while young patients had more STR1 cells. Tumor of low T category had more STR7 and STR8 cells. STR0 and STR1 cells were enriched in grade II-III tumors, while grade I tumors had more STR2, STR3, STR4, STR6 and STR7 cells, which were stroma cells with few connections to inflammatory cells.

Fig. R5. Volcano plot showing the association of a) INF, b) TUM and c) STR cell clusters with clinicopathological characteristics. Fold change is calculated as the ratio of the median percentage value between groups. *P* values are calculated using the Mann-Whitney U test with false discovery rate-correction for multiple testing. FDR values smaller than 10^{-5} are set to 10^{-5} . The horizontal dotted line indicates an FDR value of 0.05.

Abbreviations: FC, fold change; FDR, false discovery rate.

Comment 8. Line 284: please include ‘models’ in the subheading, as the CDK inhibitor data is on breast cancer models and not on patients.

Response: Thanks for your suggestion. We have added the word “models” to the subheading

(Revised Manuscript, page 11, line 28).

Comment 9. It is currently unclear how the MITH score is calculated. Figure 5a and the formula tell the same story, however, in the text it says that the features' variance is normalized. Therefore, the standard deviation would also be normalized and the formula doesn't make sense.

Response: The features' variance was normalized across all tumor nuclei from all samples, while MITH was calculated for each individual sample based on the features' variance across tumor nuclei belonging to this sample (Fig. R6). MITH of a certain sample can be either higher or lower than 1, depending on whether the variance of this sample's nuclei exceeds or falls below the variance across all nuclei from all samples. To avoid misunderstanding, we have provided a clearer description of the MITH calculation in the Methods section (Revised Manuscript, page 31 line29 to page 32 line 7):

Fig. R6. Diagram of nuclear feature normalization and MITH calculation.
Abbreviations: MITH, morphological intratumor heterogeneity; Std, standard deviation.

Based on the nuclear morphological and texture features, we measured the tumor nuclear MITH for each individual sample. First, values of each morphological and texture feature were normalized across all tumor nuclei from all samples. Then, for each individual sample, MITH was calculated as the mean value of the features' standard deviation across the tumor nuclei belonging to this sample:

$$MITH = \frac{1}{N} \sum_{i=1}^N \sqrt{\frac{\sum_{j=1}^M (x_{ij} - \bar{x}_i)^2}{M}}$$

where N is the number of features, M is the number of tumor nuclei belonging to the sample, x_{ij} is the normalized value of feature i of cell j and \bar{x}_i is the mean value of the normalized feature i of the sample.

Comment 10. Line 428: For the ecotype classifier the authors build an SVM classifier but do not show how to use it and why they choose an SVM classifier as it is quite an old method.

Response: The SVM classifier is used to classify samples' tumor ecotypes using their MEM composition as input.

Ecotype classification was based on hierarchical clustering of samples' MEM composition, which cannot be used to determine the ecotypes of new cases. Therefore, we developed a classifier which can classify the tumor ecotype using the MEM composition as input. We combined this classifier with the sc-MTOP, cell clustering extrapolation and MEM identification algorithm into an end-to-end pipeline to determine the tumor ecotype from WSIs. We have revised our manuscript to clarify the use of this ecotype classifier (Revised Manuscript, page 17, line 14-16; page 18 line 23 to page 19 line 2).

SVM model achieved a sufficiently high accuracy in our task.

The SVM model that we developed achieved an F1 score of 0.98 in the test set (Supplementary Fig. S13b), which we believe is a sufficiently high accuracy. In addition, we tried other machine learning models including random forest and neural network based on the same training-test dataset split. We found that their performance was not superior to the SVM model (Table R1).

Table R1. Performance of SVM, random forest and neural network models in predicting tumor ecotypes.

Classifier	Accuracy	F1 score
SVM	98.4%	0.98

Random Forest	92.2%	0.92
Neural Network	95.3%	0.95

Comment 11.1 The results on prognosis seem counter-intuitive, as the Ecotype 1 has most ER+ patients and a poor prognosis. In general ER+ breast cancer has a better prognosis than TN or HER2+.

Response: We analyzed the association between hormone receptor (HR) status, ecotype classification and patient prognosis. As you mentioned, HR+ patients exhibited a trend toward better prognosis than HR- patients (Figure R7a). However, **within the HR+ group, the prognosis of Ecotype 1 patients was significantly worse than those of other ecotypes and was comparable to HR- patients** (Figure R7b). That is to say, the prognosis of Ecotype 1 HR+ patients was not good. Therefore, despite more HR+ cases in Ecotype 1, the overall prognosis was worse than the other ecotypes.

Comment 11.2 Also many ER+ were N+. Could this be a confounder? In addition, some more analyses with associations on clinical variables would be appreciated. What is the effect of treatment? Why is grade not taken into account into the multivariate model for the ecotypes?

Response: According to your comments, we re-performed multivariate Cox analyses, adjusting for more potential confounders including age, IHC subtype, tumor T category, N category, grade, chemotherapy (yes or no) and radiotherapy (yes or no). The results indicated that tumor ecotype classification remains an independent prognostic factor (Fig. R7c, also see Fig. 7e).

Fig. R7. Data related to comment 11.1 and comment 11.2.

a. Kaplan-Meier curves of recurrence-free survival of HR- patients and HR+ patients.

b. Kaplan-Meier curves of recurrence-free survival of HR- patients and HR+ patients stratified by the tumor ecotype. P values are calculated using the log-rank test.

c. Multivariate Cox analysis reveals breast cancer ecotype as an independent prognostic factor of recurrence-free survival. Abbreviations: RFS, recurrence-free survival; IHC, immunohistochemistry.

Comment 11.3 Also a better description of the cohort would be good.

Response: (also see response to your comment 1.1) We specified the histological types of breast cancers, the settings how cases were collected, and the inclusion/exclusion criteria in the Methods section (Revised manuscript, page 25, line 3-9):

The discovery cohort is a retrospective cohort that included consecutive breast cancer patients who received standard treatment at the Department of Breast Surgery, Fudan University Shanghai Cancer Center (FUSCC) from January 1, 2013 to December 31, 2014 according to the following criteria: a) female patients diagnosed with unilateral invasive carcinoma of no special type; b) no evidence of distant metastasis at diagnosis; and c) treated by surgical resection without prior anti-tumor treatment.

We added a brief cohort description to the Results section to help readers better understand the study population (Revised manuscript, page 5, line 25-29):

..., we applied sc-MTOP to 637 whole slide images (WSIs) of a retrospectively collected cohort with multiomics profiling data. This cohort included 637 consecutive breast cancer patients who received standard treatment at Fudan University Shanghai Cancer Center (FUSCC) from January

1, 2013 to December 31, 2014.

Comment 12. The authors show a lot of correlation values in the results and figures. However, they often do not provide confidence intervals and thus we cannot see if the correlations are significantly different from 0.

Response: According to your suggestions, we have added the confidence intervals of the correlation coefficient (Fig. 5c, Supplementary Fig. S1d, S9c, S13c-d and Source data).

Comment 13. The authors mention that their pipelines are unsupervised. When in fact the biggest part of the models are supervised models.

Response: As you mentioned, Hover-Net that we used for nuclear segmentation and classification is a supervised model. The subsequent feature-based cell clustering is unsupervised analysis. We have corrected the sentence “our study adopted an unsupervised data-driven analytic pipeline” as follows (Revised manuscript, page 21, lines 25-27):

in our study, based on data from nuclear segmentation and feature extraction, we adopted a data-driven analytic pipeline.

Comment 14.1 how is the reproducibility on a second slide of the same patient?

Response: To evaluate this reproducibility, we a) randomly selected 50 patients, b) collected and scanned a second slide from each, c) applied sc-MTOP and subsequent analytic pipeline, and d) assessed the consistency of the major metrics and markers of potential clinical relevance between these second slides and the original used ones. **The results demonstrated high reproducibility of our algorithm on a second slide of the same patient:**

The cell type and cell cluster percentage showed high correlation between the second slides and the original used slides, with most correlation coefficient values higher than 0.60 (Spearman

correlation test, all $P < 0.001$, median correlation coefficient=0.81) (Fig. R8a, also see Supplementary Fig. 14a). The ecotype classification results showed high consistency between the second slide and the original used (accuracy: 88.0%, F1 score: 0.87) (Fig. R8b, also see Supplementary Fig. S14b). As for the clinically-relevant features, both AIC score (Spearman correlation coefficient: 0.91 (0.84-0.95), $P < 0.001$) and MITH (Spearman correlation coefficient: 0.80 (0.68-0.88), $P < 0.001$) were significantly correlated between the two groups of slides (Fig. R8c-d also see Supplementary Fig. S14c-d). These results demonstrated high reproducibility of the results of our algorithm on different slides from the same patient.

Fig. R8. Reproducibility evaluation of our sc-MTOP algorithm on a second slide of the same patient.

a. Barplot and boxplot (inset) showing the correlation of the percentage of major cell types and cell clusters (TUM, STR and INF) between the original slide and the second slide from the same patient. Spearman correlation coefficient with 95% confidence interval and the corresponding P values are shown. ***, $P < 0.001$. Inset box plot displays the distribution of correlation coefficients. The center line of boxplot indicates the median values; box limits show upper and lower quartiles; whiskers extend from box limits to the farthest data point within $1.5 \times$ interquartile range; points beyond whiskers are outliers.

b. Confusion matrix of the ecotype classification results for the original slide and the second slide of the same patient.

c. Comparison of the AIC score between the original slide and the second slide from the same patient. Spearman correlation coefficient with 95% confidence interval and the corresponding P values are shown.

d. Comparison of MITH between the original slide and the second slide from the same patient. Spearman correlation coefficient with 95% confidence interval and the corresponding P values are shown.

Abbreviations: AIC score, aggregated inflammatory cell abundance score; MITH, morphological intratumor heterogeneity; CC, correlation coefficient.

Comment 14.2 Were all WSIs scanned by one scanner? How does the method perform on other scanners?

All WSIs were scanned using the NanoZoomer scanner (Hamamatsu Photonics, Japan).

Technically, our sc-MTOP algorithm can be applied to images in all formats compatible with Hover-Net, including formats of ndpi, sv5, mrxs and tif. However, previous studies have shown that deep learning algorithms applied to WSIs scanned by different scanners often yield discrepant results, which may be attributed to the differences in image style across scanners²³. Currently, there is no universally recognized standard method to fully resolve this issue²⁴. Our sc-MTOP algorithm employed a deep learning model, Hover-Net, for nuclear segmentation. It may yield varying results, when applied to WSIs from different scanners. Limited by equipment constraints, we have not yet optimized and adapted the algorithm for slides from different scanners. Therefore, we believe that before applying the sc-MTOP algorithm and relevant conclusions to WSIs from other scanners, further study is needed to verify or optimize the reproducibility of sc-MTOP on different scanners. We have discussed this limitation of our algorithm in our revised manuscript:

The WSIs analyzed in our study were scanned using the NanoZoomer scanner. Limited by equipment constraints, we have not yet optimized and adapted the algorithm for slides from different scanners. Further study is needed to verify or optimize the reproducibility of sc-MTOP across the slides from different scanners, before applying it and the relevant conclusions to WSIs from other scanners.

Reviewer #3 (Remarks to the Author): Expert in breast cancer tumour microenvironment, digital pathology, and single-cell and spatial omics

In this study, the authors developed an algorithm which uses whole slide images (WSI) to identify malignant, immune and stromal cells, using the texture and topological features of each single cell

/ nucleus. The authors further divide these three main cell types into more distinct spatially relevant cells. Then identify on the WSI micro-ecological modules (MEM) which through unsupervised clustering form tumor ecotypes. Authors emphasize that their algorithm is clinically relevant through the association of ecotypes with prognostics or by showing that the spatially relevant cell types may guide response to therapies.

Response: We are very grateful to you for capturing the main line of our research and providing an insightful summary. We want to clarify a subtle misunderstanding. Our algorithm did not use the nuclear features to identify malignant, immune and stromal cells. Rather, we employed a well-established and widely-accepted model, Hover-Net⁷, for nuclear segmentation and classification (see Revised Manuscript page 4, line 8-12) and features were extracted for individual cells based on the nuclear segmentation and classification results. To avoid this misunderstanding, we have revised the legend of Fig. 1a (schematic diagram of our algorithm) to make it clearer and more informative:

Schematic diagram of single-cell morphological and topological profiling (sc-MTOP). It first employs Hover-Net to implement nuclear segmentation and classification on WSIs. Then, for individual cells, nuclear morphological, texture and topological features are extracted based on the nuclear contour and the intercellular spatial relationship.

Besides, we highly value your comment regarding validating the accuracy of tumor, inflammatory and stromal cell classifications. We have performed validation analysis using paired HE-stained WSIs and images of co-detection by indexing (CODEX), a spatial proteomics technique. The results have been included in our Revised Manuscript (see response to your Comment 1). We have also studied all other comments, conducted additional analysis and incorporated the results into our Revised Manuscript. The revisions have been marked in red in the file “Revised Manuscript”. Here are our point-by-point responses to your comments:

Comment 1. The main validation, one would have liked to see is the accuracy of the algorithm to classify immune, stromal and malignant cells using paired HE stains and either spatial transcriptomics or proteomics.

Response: We validated the accuracy of the Hover-Net model to classify immune, stromal and malignant cells using the paired HE-stained WSIs and images of co-detection by indexing (CODEX), a spatial proteomics technique¹⁵. First, we retrieved the FFPE blocks of tumor specimen and made tissue slides of two cases. Second, CODEX experiment and imaging was performed on the PhenoCycler-Fusion system (Akoya Biosciences, Menlo Park, CA) following the standard protocol. After this, we washed out the flow cells on the slides and performed H&E staining. We scanned the H&E-stained slides to generate WSIs, applied the Hover-Net model, and analyzed the consistency between the Hover-Net cell classification results and the CODEX images (Fig. R9a, also see Supplementary Fig. S1a). Examples of paired cell classification and CODEX images were shown and manifested a high classification accuracy (Fig. R9b, also see Supplementary Fig. S1b). Furthermore, since currently there is no algorithm to match individual cells on CODEX images and WSIs, we marked 25 matched regions (100 * 100 μm) on the CODEX images and WSIs and compared the number of tumor, inflammatory and stroma cells based on the CODEX images and Hover-Net cell classification. It was found that the estimated cell numbers of tumor, stroma and inflammatory cells showed high correlation between the CODEX and Hover-Net cell classification (Spearman correlation coefficient: 0.99 for tumor cells, 0.96 for inflammatory cells and 0.95 for stroma cells) (Fig. R9c-d, also see Supplementary Fig. S1c-d). We have added the methods concerning CODEX and the above validation analysis results to our revised manuscript (Revised manuscript, page 6, line 10-18; page 28 line 25 to page 29 line 10).

Figure R9. Validation of cell classification accuracy using paired co-detection by indexing (CODEX) and H&E-stained WSIs.

a. Experiment design for paired CODEX and H&E-staining on tissue slides.

b. Examples of CODEX staining and Hover-Net cell classification results in matched regions. Scale bar for the first row images: 400µm; Scale bar for the second row images: 100µm.

c. The estimated number of tumor, inflammatory and stroma cells through CODEX and Hover-Net cell classification in matched regions (25 randomly selected 100 * 100 µm regions).

d. Comparison of the estimated number of tumor, inflammatory and stroma cells between CODEX and Hover-Net cell classification in matched regions (25 randomly selected 100 * 100 µm regions). Spearman correlation coefficient with 95% confidence interval and the corresponding P values are shown.

Abbreviations: FFPE, formalin-fixed paraffin-embedded; CODEX, co-detection by indexing; WSI, whole slide image; CC, correlation coefficient.

Comment 2. What about endothelial cells and adipocytes? they are an important part of breast tumors. However, the authors barely mention them. Would these be counted as stromal cells? if yes are they making a specific STR subtype?

Response: Endothelial cells and adipocytes are classified as stromal cells.

First, nuclei of both endothelial cells and adipocytes are labeled as stromal nuclei in the PanNuke dataset, which is used for the Hover-Net model development²⁵. Therefore, theoretically these two cell types would be classified as stromal cells by the model. Second, we visualized the classification results of several endothelial and adipocyte. As expected, they are classified as stromal cells (Fig. R10a-b, left and middle panel). Third, we estimated cell abundance of endothelial cells and adipocytes based on the RNA-seq data and found a positive correlation between them and the percentage of stroma cells (Fig. R10c). We have shown this correlation in Supplementary Fig. S2a.

Both endothelial and adipocytes are associated with certain STR subtypes, but do not constitute a specific STR subtype.

We visualized the STR classification results of several endothelial and adipocyte. Both of them are composed of several STR clusters rather than constitute a specific STR cluster (Fig. R10a-b, left and right panel). We also examine the correlation between the STR clusters and the estimated abundance of endothelial cells and adipocytes. Consistent with the visualization results, the abundance of endothelial cells showed evident positive correlation to STR3, STR6, STR7, STR0, STR8, STR2 and STR4 while the abundance of adipocytes showed significant positive correlation to STR3, STR6, STR7, STR2, STR4, STR8 and STR5 (Fig R10d).

Figure R10. Nuclear classification and stroma cell (STR) clustering results of endothelial cells and adipocytes.

a-b Example of nuclear classification and STR clustering results of a) endothelial cells and b) adipocytes. The raw images (left), nuclear segmentation results (middle) and endothelial cells or adipocytes annotated with the STR clusters (right) were shown.

c. Correlation between the tumor, inflammatory and stroma cell percentage with the estimated abundance of endothelial cells and adipocytes through single-sample gene set enrichment analysis based on the RNA-seq data.

d. Correlation between the abundance of stroma cell clusters with the estimated abundance of endothelial cells and adipocytes through single-sample gene set enrichment analysis based on the RNA-seq data.

Comment 3. What about the polynuclear immune cells? I would have thought these would had been possible to identify on the WSI, as they have a specific shape. Would these make a specific INF subtype?

Response: The majority of the polynuclear immune cells were classified as inflammatory cells by the Hover-Net model (Fig. R11a, left and middle panel; Fig. R11b). **Visualization of INF clustering results showed that these polynuclear immune cells encompassed several INF**

clusters including INF0, INF1, INF3, INF4, INF6 and INF7, rather than making a specific INF cluster (Fig. R11a, left and right panel). Consistent with the visualization results, correlation analysis indicated that the abundance of neutrophils and eosinophil was significantly associated with INF0, INF1, INF3, INF6 and INF7 (Fig. R11c).

Figure R11. Nuclear classification and inflammatory cell (INF) clustering results of polynuclear immune cells.
a Example of nuclear classification and INF clustering results of polynuclear immune cells. The raw images (left), nuclear segmentation results (middle) and multinuclear immune cells annotated with the INF clusters (right) were shown.
b. Correlation between the tumor, inflammatory and stroma cell percentage with the estimated abundance of eosinophils and neutrophils through single-sample gene set enrichment analysis based on the RNA-seq data.
c. Correlation between the abundance of inflammatory cell clusters with the estimated abundance of eosinophils and neutrophils through single-sample gene set enrichment analysis based on the RNA-seq data.

Comment 4. The authors use organoids to investigate whether the MITH is associated with resistance to CDK4/6 inhibitors. Since the authors also have WES data it would have been interesting to potentially associate MITH with genomic heterogeneity or the resistance to CDK4/6

inhibitors to genomic alterations.

Response: We performed supplementary analyses and experiments to investigate the association between MITH and genomic heterogeneity/alterations and the association between genomic alterations and CDK4/6 inhibitor response. The results were as follows:

High MITH was correlated with high genomic heterogeneity and high frequency of TP53 mutation.

Based on the WES data of our FUSCC cohort, we calculated mutant-allele tumor heterogeneity (MATH)²⁶ to measure the tumor genomic intratumor heterogeneity and found a moderate positive correlation between MITH and MATH (Spearman correlation coefficient: 0.20, $P < 0.001$) (Fig R12a, also see Fig. 5c).

We next compared genomic mutation frequencies between the high- and low-MITH cases. Overall, high-MITH was associated with high frequency of *TP53*, *GATA3* and *MAP3K1* mutation. Subgroup analysis stratified by the hormone receptor status showed that only *TP53* mutation varied significantly between high- and low-MITH cases in HR+ breast cancers (Fig. R12b-d, also see Supplementary Fig. 8h). Interestingly, previous studies reported that *TP53* mutation can lead to CDK4/6 inhibitor resistance through upregulating CDK2^{27,28}, which indicated that *TP53* mutation may be one of the upstream genomic alterations for CDK4/6 resistance of HR+ high-MITH breast cancers. We have added the sentence in our Manuscript:

Genomic analysis showed that high-MITH was associated with high frequency of *TP53* mutation, which has been reported to lead to the upregulation of CDK2^{27,28}

***TP53* mutation was associated with CDK4/6 inhibitor resistance in PDO models.**

Since the 8 PDO samples for drug response experiment reported in our original manuscript did not have genomic sequencing data, to investigate the association between genomic alterations and CDK4/6 inhibitor responsiveness, we collected another 13 HR+HER2- breast cancer specimens and a) performed targeted sequencing using a well-established panel²⁹, b) established PDOs and

conducted drug response test and c) collected the H&E-stained WSI, applied sc-MTOP and evaluated MITH. Compared with *TP53*-wildtype samples, PDOs from *TP53*-mutated samples exhibited relatively low sensitivity to CDK4/6 inhibitor ($P = 0.030$) (Figure R12e), which was consistent with the previous study conclusion^{27,28}.

In addition, we expanded the sample size of our analysis investigating the correlation between MITH and CDK inhibitor responsiveness to 21 PDO samples. Consistent with the results of 8 PDO samples reported in our original manuscript, the PDOs from high-MITH samples were less sensitive to CDK4/6 inhibitor than those from low-MITH samples (Comparison of relative viability, $P < 0.001$) and no significant difference in response to CDK2/4/6 inhibitor was observed between the two groups (Comparison of relative viability, $P = 0.859$) (Fig. R12f, also see Fig. 5k).

Based on these analyses, we have displayed the data on a) association between MITH and genomic heterogeneity, b) association between MITH and *TP53* mutation frequency and c) drug response test on the combined 21 PDOs and made the corresponding revisions to our manuscript (page 12, line 30-32; page 13, line 10-11; page 13, line 14-18).

Fig. R12. Data related to Reviewer 3, comment 4.

a. Correlation between MITH and genetic intratumor heterogeneity evaluated by mutant-allele tumor heterogeneity. Spearman correlation coefficient with 95% confidence interval and the corresponding *P* value are shown.

b-d. Genomic mutation frequencies between high- and low-MITH cases in b) entire cohort, c) HR+ group and d) HR- group. The top 10 mutated genes within each group are shown. *P* values are calculated using the chi-square test with false discovery rate-correction for multiple testing. ***, *P* < 0.001; *, *P* < 0.01

e. Relative viability of PDOs with/without *TP53* mutation treated with 1 μ m CDK4/6 inhibitor Abemaciclib. *TP53*-WT, *TP53*-wild type; *TP53*-MT, *TP53*-mutated. *P* value is calculated using the Mann-Whitney U test. *, *P* < 0.05.

f. Relative viability of PDOs treated with 1 μ m CDK4/6 inhibitor Abemaciclib (left) and 0.4 μ m CDK2/4/6 inhibitor PF-06873600 (right). The difference in relative viability between the high- and low-MITH samples is compared. *P* values are calculated using the Mann-Whitney U test. ***, *P* < 0.001; ns, not significant.

Abbreviations: MITH, morphological intratumor heterogeneity; CC, correlation coefficient.

Comment 5. When the authors use organoids to investigate CDK4/6 inhibitors it is necessary to show if such organoids have the same tumor / genetic alterations than the primary tumors. Can the same MITH seen at the patient level been reproduced on PDOs?

Response: We designed experiments to examine the consistency between primary tumors and the corresponding PDOs in MITH and genomic alterations (evaluated through targeted sequencing with a well-established panel²⁹) (Fig. R13a, also see Supplementary Fig. S9a). The results were as follow:

The PDOs preserved most of the high prevalence genomic alterations found in primary tumor

samples, including *PIK3CA*, *TP53*, and *GATA3* mutation (Fig. R13b, also see Supplementary Fig. S9b). MITH of PDOs were highly correlated with that of the corresponding primary tumor (Spearman correlation coefficient: 0.81, $P = 0.001$) (Fig. R13c-d, also see Supplementary Fig. S9c-d). These analyses demonstrated the validity of using PDOs to study the correlation between MITH and response of tumor to drugs. We have incorporated these data into our Revised Manuscript.

Fig. R13. Comparison between the primary tumors and the corresponding PDOs in genomic alterations and MITH (related to Reviewer 3, comment 5).

a. Diagram of examining the consistency between the primary tumors and the corresponding PDOs in genomic alterations and MITH.

b. Comparison of genomic alterations between the primary tumors and the corresponding PDOs (T, primary tumors; O, PDOs).

c. Comparison of MITH values between the primary tumors and the corresponding PDOs. Spearman correlation coefficient with 95% confidence interval and the corresponding P value are shown.

d. Local images of the primary tumors and the corresponding PDOs. Scale bar: 200 μ m.

Abbreviations: MITH, morphological intratumor heterogeneity; PDO, patient-derived organoid; CC, correlation coefficient.

Comment 6. It is surprising that only five of the INF subtypes (1,7,0.6, 3) in Figure 3A correlate with immune cell types, what are the others IFN subtypes correlated with? I understand that the correlation is made only with immunomodulatory subtypes, but it would be good to understand

what the other are 'made of'?

Response: Our INF classification of inflammatory cells was mainly based on their topological features, which reflected their spatial distribution patterns of their relationships with other cell types. In brief, INF0, INF1, INF6, INF7 clusters represented locally aggregated inflammatory cells. INF3 and INF4 comprised inflammatory cells that formed relatively small aggregates. The other clusters, INF2, INF5, INF8, INF9, encompassed scattered distributed inflammatory cells (refer to Figure 2a).

We included more immune cell subsets in the correlation analysis but still found that only INF0, INF1, INF6, INF7, INF3 cells were positively associated with the abundance of immune cell subsets (Fig. R14, also see Fig. 3a) (Revised Manuscript, page 8, line 24-26). This result may be interpreted as: the high abundance of spatially aggregated inflammatory cells indicated the overall activation of the tumor immune microenvironment, as manifested by elevated levels of various immune cell subsets.

Fig R14. Correlation between the abundance of inflammatory cell clusters and abundance of microenvironment immune cells estimated using ssGSEA based on the RNA-seq data.

Spearman correlation analysis is performed with *P* values corrected for multiple testing. Red/blue dots represent positive/negative correlation respectively. The size of the dots is proportional to the Spearman correlation coefficient. The outlined dots indicate false discovery rate corrected *P* values < 0.05. Rows are ordered by hierarchical clustering.

Comment 7. It seems that the ecotypes and the MEMs are very correlated (Fig7a), while the authors associate the ecotypes with prognosis it would had been interesting to see the same for the individual MEMs and single cell subtypes, it could help to understand how the ecotypes are linked to prognosis, indeed while the cox analysis show an independent prognostic value, it is not explained or hypothesized how/why that may be.

Response: Thanks for your suggestions. We performed univariate Cox regression analysis on the individual MEMs and the single cell clusters and have incorporated the results in our Revised Manuscript (Fig. R15; also see Supplementary Fig. S3b, Supplementary Fig. S6d, Supplementary Fig. S7d):

High abundance of INF0 and INF7 was associated with favorable prognosis. (Revised Manuscript, page 8, line 14-15). High abundance of TUM0 and TUM5 was correlated with poor prognosis. (Revised Manuscript, page 11, line 2-3). High abundance of STR2, STR4 and STR5 was associated with poor prognosis. (Revised Manuscript, page 11, line 24-25). High percentage of Module8_IA was associated with favorable prognosis, while high percentage of Module4_TS was correlated with poor prognosis. (Revised Manuscript, page 15, line 17-19)

Besides, as you mentioned, these results provided valuable insights into how ecotypes are linked to prognosis:

The poor prognosis of Ecotype 1 patients may be explained by their enrichment of Module4_TS and the corresponding TUM0, TUM5, STR5, STR4 and STR2 cells, which correlated with poor prognosis. The better prognosis of Ecotype 2 patients may be attributed to their enrichment of Module8_IA and the corresponding INF0 and INF7 cells, which were associated with favorable prognosis. (Revised Manuscript, page 17, line 7-12)

Fig. R15. Forest plot showing the univariate Cox regression analysis of relapse-free survival for a) tumor cell clusters, b) stroma cell clusters, c) inflammatory cell clusters and d) MEMs. Cell clusters and MEMs are modeled as log-transformed proportion data. Squares and whiskers represent point estimates and the 95% confidence interval of hazard ratios.

Comment 8. Is it complicated to link ecotypes to IFN, TUM and STR? would it be possible to retrieve the cellular subtypes composition of each ecotype?

Response: We presented the abundance of IFN, TUM and STR clusters according to the ecotype (Fig. R16, also see Supplementary Fig. S11c). **The cell clusters enriched in each ecotype coincide with the MEMs enriched in that ecotype.** Ecotype 1 was enriched for TUM0, TUM5, TUM8, STR2, STR4, STR5, which belong to Module4_TS; Ecotype 2 was enriched for INF0, INF1, INF6, INF7, TUM7 which belong to Module8_IA, and INF3, TUM1, STR1 which belongs to Module7_TSI. Ecotype 3 was enriched for INF4, INF5, STR0, STR3 which belong to Module6_SI, and INF9, STR6, STR7, STR8 which belong to Module5_LC. Ecotype 4 was enriched for TUM2, TUM3 and TUM4, which belong to Module1_TC. We have incorporated

these data to our Revised Manuscript (Supplementary Fig. S11c).

Figure R16. Cell cluster profiles according to the breast cancer ecotypes.
Rows are ordered according to the micro-ecological modules.

Comment 9. Is the step of MEM necessary to obtain ecotypes? Going from 8 modules to 4 ecotypes through unsupervised clustering almost seem unnecessary?

Response: We tried to omit the MEM step and performed clustering on the samples' cell cluster profiles using hierarchical clustering method, which is also used to derive the MEM-based ecotypes. The recommended number of clusters was reduced to two according to the Calinski-Harabasz index and Silhouette coefficient (Fig. R17a-b, refer to Supplementary Fig. S11a for comparison). We then compared this cell cluster-based ecotype classification to our MEM-based one. While cell cluster-based Ecotype 1 largely correspond to the MEM-based Ecotype 2, cell cluster-based Ecotype 2 encompassed a mixture of MEM-based Ecotype 2, 3 and 4 (Fig. R17c). In survival analysis, patients of cell cluster-based Ecotype 1 exhibited a trend toward preferable prognosis, which was consistent with the favorable prognosis of those of MEM-based Ecotype 2 (Fig. R17d, refer to Fig. 7d-e for comparison). However, the cell cluster-based ecotype classification did not identify a group of patients with poor prognosis such as those in MEM-based Ecotype 1. Based on these data, **we believe that our MEM-based ecotype classification was superior to cell cluster-based ecotype classification in characterizing the ecotype diversity**

and stratifying patient prognosis.

Fig. R17. Cell cluster-based ecotype classification and its comparison to MEM-based ecotype classification.

a. Selection of cluster number for hierarchical clustering based on Calinski-Harabasz index and Silhouette coefficient.

b. Hierarchical clustering of cell cluster profiles across 637 patients identifies two feature-based ecotypes.

c. Sankey plot showing the correspondence between the cell cluster-based ecotype classification and MEM-based ecotype classification.

d. Kaplan-Meier curves of recurrence-free survival according to the cell cluster-based ecotypes.

Abbreviations: MEM, micro-ecological module; RFS, recurrence-free survival.

Comment 10. It would had been super interesting to deconvolute the RNAseq data with CIBERSORTx and a reference of well annotated scRNA-seq data to understand if the IFN, STR and TUM correspond/correlate with to some more granular cell phenotypes, if it is the case all the biological downstream interpretation of prognosis and response to treatment would had been easier and had made much more sense.

Response: Following your advice, we deconvolute the RNA-seq data using CIBERSORTx using with a reference of annotated scRNA-seq data from Wu et al. Based on the results, we performed correlation analysis between our IFN, STR and TUM clusters and the biologically-defined cell

subsets. The results are summarized as follow:

TUM0, TUM3 and TUM5 cells that were enrich in HR+HER2- breast cancers were correlated with luminal cancer cells. TUM2 and TUM11 cells were positively associated with HER2-enriched cancer cells, which was in line with their enrichment in HER2+ breast cancers. TUM4 and TUM6 that were enrich in TNBCs were positively correlated with basal-like cancer cells (Fig. R18a). For stroma cells, STR0 and STR1 were positively correlated with differentiated perivascular cells, while STR2, STR3, STR4, STR6 and STR7 were positively associated with myofibroblast-like cancer associated fibroblasts. STR3, STR6 and STR7 were positively correlated with ACKR1+endothelial but showed a negative association with LYVE1+ lymphatic endothelial cells. For inflammatory cells, similar to the correlation analysis results though ssGSEA (Fig. R14), only INF0, INF1, INF6 and INF7 cells showed positive correlation with several immune cell subsets.

Figure R18. Correlation between the sc-MTOP-based clusters and cell subsets estimated through CIBERSORTx for a) tumor cells, b) stroma cells and c) inflammatory cells respectively.

Spearman correlation analysis is performed with *P* values corrected for multiple testing. Red/blue dots represent positive/negative correlation respectively. The size of the dots is proportional to the Spearman correlation coefficient. The outlined dots indicate false discovery rate corrected *P* values < 0.05. Rows are ordered by hierarchical clustering.

Abbreviations: CC, correlation coefficient; PVL, perivascular cells; CAF, cancer associated fibroblast.

We appreciate your suggestions to correlate our cell clusters (INF, TUM and STR) to the biologically-defined cell subsets and have performed several analyses including correlation of STR clusters with endothelial and adipocytes (response to your Comment 2), correlation of INF clusters with a variety of immune cells (response to your Comment 2 and 6), and a comprehensive correlation analysis between INF, TUM, STR clusters with CIBERSORTx-based cell subsets (response to your Comment 10). **Overall, the results indicated that there are indeed some mild to moderate correlations between our spatially-defined cell clusters and the biologically defined cell subsets, and, as you mentioned, a few of these correlations may explain the downstream findings concerning prognosis and treatment response.** For example, the positive correlation of STR4 and STR5 with myofibroblast-like CAFs may explained their association with poor prognosis. The positive correlation of INF0, INF1, INF6 and INF7 with immune cell subsets may interpret why a high AIC score (abundance sum of these four clusters) indicated a better response to immunotherapy of TNBC. However, with the current data, we have not yet been able to reach a definite conclusion on the correspondence between the spatially-define cell clusters and the biologically-defined cell subsets. Since these two data provided different information, a better perspective may be integrating our data or methods with spatially-resolved molecular data for the multimodal characterization of tumor ecosystem and the discovery of novel biologically meaningful tissue structures. We have incorporated this point of view in our Revised Manuscript (page 23, line 8-11).

Comment 11. I would not call, malignant or cancer cells tumor cells, is not any cell coming for a tumor a tumor cell?

Response: We really appreciate your comments and highly value this terminology question. We first surveyed the studies using the Hover-Net model trained on the PanNuke dataset and found that many of them also used the word “tumor cells”^{6,30,31}. Second, to avoid misunderstanding, we

have revised the sentence describing Hover-Net as follow (Revised Manuscript, page 26, line 18-23):

Hover-Net is a deep convolutional neural network for simultaneous nuclear segmentation and classification on WSIs. The model pre-trained on the public PanNuke dataset was used, which can identify five nuclear types including tumor (specifically refers to breast cancer cells in this study), inflammatory, stromal, normal (non-neoplastic epithelial) and dead (necrotic) nuclei²⁵.

Response:

1. Ciga, O., Xu, T. & Martel, A.L. Self supervised contrastive learning for digital histopathology. *Machine Learning with Applications* **7**, 100198 (2022).
2. Hwang, W.L., *et al.* Single-nucleus and spatial transcriptome profiling of pancreatic cancer identifies multicellular dynamics associated with neoadjuvant treatment. *Nat Genet* **54**, 1178-1191 (2022).
3. Wagner, J., *et al.* A Single-Cell Atlas of the Tumor and Immune Ecosystem of Human Breast Cancer. *Cell* **177**, 1330-1345 e1318 (2019).
4. Failmezger, H., *et al.* Topological Tumor Graphs: A Graph-Based Spatial Model to Infer Stromal Recruitment for Immunosuppression in Melanoma Histology. *Cancer Res* **80**, 1199-1209 (2020).
5. Wang, H., *et al.* Single-Cell Spatial Analysis of Tumor and Immune Microenvironment on Whole-Slide Image Reveals Hepatocellular Carcinoma Subtypes. *Cancers (Basel)* **12**(2020).
6. Fremond, S., *et al.* Interpretable deep learning model to predict the molecular classification of endometrial cancer from haematoxylin and eosin-stained whole-slide images: a combined analysis of the PORTEC randomised trials and clinical cohorts. *Lancet Digit Health* **5**, e71-e82 (2023).
7. Graham, S., *et al.* Hover-Net: Simultaneous segmentation and classification of nuclei in multi-tissue histology images. *Med Image Anal* **58**, 101563 (2019).
8. Traag, V.A., Waltman, L. & van Eck, N.J. From Louvain to Leiden: guaranteeing well-connected communities. *Sci Rep* **9**, 5233 (2019).
9. Heumos, L., *et al.* Best practices for single-cell analysis across modalities. *Nat Rev Genet*, 1-23 (2023).
10. Corredor, G., *et al.* An Imaging Biomarker of Tumor-Infiltrating Lymphocytes to Risk-Stratify Patients With HPV-Associated Oropharyngeal Cancer. *Journal of the National Cancer Institute* **114**, 609-617 (2022).
11. Newman, A.M., *et al.* Robust enumeration of cell subsets from tissue expression profiles. *Nat Methods* **12**, 453-457 (2015).
12. Becht, E., *et al.* Estimating the population abundance of tissue-infiltrating immune and stromal cell populations using gene expression. *Genome Biol* **17**, 218 (2016).
13. Lim, B., Lin, Y. & Navin, N. Advancing Cancer Research and Medicine with Single-Cell Genomics. *Cancer Cell* **37**, 456-470 (2020).
14. Bandura, D.R., *et al.* Mass cytometry: technique for real time single cell multitarget immunoassay based on inductively coupled plasma time-of-flight mass spectrometry. *Anal Chem* **81**, 6813-6822 (2009).

15. Black, S., *et al.* CODEX multiplexed tissue imaging with DNA-conjugated antibodies. *Nat Protoc* **16**, 3802-3835 (2021).
16. Merritt, C.R., *et al.* Multiplex digital spatial profiling of proteins and RNA in fixed tissue. *Nat Biotechnol* **38**, 586-599 (2020).
17. Danenberg, E., *et al.* Breast tumor microenvironment structures are associated with genomic features and clinical outcome. *Nat Genet* **54**, 660-669 (2022).
18. Schürch, C.M., *et al.* Coordinated Cellular Neighborhoods Orchestrate Antitumoral Immunity at the Colorectal Cancer Invasive Front. *Cell* **182**, 1341-1359.e1319 (2020).
19. Yamawaki, T.M., *et al.* Systematic comparison of high-throughput single-cell RNA-seq methods for immune cell profiling. *BMC Genomics* **22**, 66 (2021).
20. Newman, A.M., *et al.* Determining cell type abundance and expression from bulk tissues with digital cytometry. *Nat Biotechnol* **37**, 773-782 (2019).
21. Lu, C., *et al.* A prognostic model for overall survival of patients with early-stage non-small cell lung cancer: a multicentre, retrospective study. *Lancet Digit Health* **2**, e594-e606 (2020).
22. Desbois, M., *et al.* Integrated digital pathology and transcriptome analysis identifies molecular mediators of T-cell exclusion in ovarian cancer. *Nat Commun* **11**, 5583 (2020).
23. Xun, D., *et al.* Scellseg: A style-aware deep learning tool for adaptive cell instance segmentation by contrastive fine-tuning. *IScience* **25**, 105506 (2022).
24. Oza, P., Sindagi, V.A., Sharmini, V.V. & Patel, V.M. Unsupervised domain adaptation of object detectors: A survey. *IEEE Transactions on Pattern Analysis and Machine Intelligence* (2023).
25. Gamper, J., Alemi Koohbanani, N., Benet, K., Khuram, A. & Rajpoot, N. Pannuke: an open pan-cancer histology dataset for nuclei instance segmentation and classification. in *European congress on digital pathology* 11-19 (Springer, 2019).
26. Mroz, E.A., Tward, A.D., Hammon, R.J., Ren, Y. & Rocco, J.W. Intra-tumor genetic heterogeneity and mortality in head and neck cancer: analysis of data from the Cancer Genome Atlas. *PLoS Med* **12**, e1001786 (2015).
27. Álvarez-Fernández, M. & Malumbres, M. Mechanisms of Sensitivity and Resistance to CDK4/6 Inhibition. *Cancer Cell* **37**, 514-529 (2020).
28. Vilgelm, A.E., *et al.* MDM2 antagonists overcome intrinsic resistance to CDK4/6 inhibition by inducing p21. *Sci Transl Med* **11**(2019).
29. Lang, G.T., *et al.* Characterization of the genomic landscape and actionable mutations in Chinese breast cancers by clinical sequencing. *Nat Commun* **11**, 5679 (2020).
30. Chen, R.J., *et al.* Pan-cancer integrative histology-genomic analysis via multimodal deep learning. *Cancer Cell* **40**, 865-878 e866 (2022).
31. Nederlof, I., *et al.* Spatial interplay of lymphocytes and fibroblasts in estrogen receptor-positive HER2-negative breast cancer. *NPJ Breast Cancer* **8**, 56 (2022).

REVIEWER COMMENTS

Reviewer #1 (Remarks to the Author):

I would like to thank the authors for including more data and performing comprehensive analysis to address all reviewers' comments. I have two additional suggestions that will need authors' attention.

1- With respect to this reviewer's initial comment regarding the comparison with deep features, this reviewer appreciates the extra analyses that have been performed. These results show that deep features did not yield to satisfactory results (not even partially interesting results). I believe this observation needs to be further investigated. What was the patch size and magnification that was used and was the patch size (spot size as per authors' naming) the same for both MEM calculation and deep features? Why did the authors pick 9 clusters? Can the authors do the same analysis task with KimiaNet features? In case of negative results, a discussion touching on the possible reasons would be beneficial.

2- With regards to the identified cell clusters (e.g., inflammatory cell clusters (INF0-9)), in Figure 2c, the authors have shown representative patches containing various INF clusters. What is missing from the manuscript is the connection between cell features and patches. How are these patches selected? For example, Fig 2c INF0 patch, did the authors select an area that was enriched with such cell types? Or, are they clustering the patches based on cell representations as well? This needs to be super clear in the manuscript. Furthermore, the clarity of the whole analysis process can be substantially enhanced if a workflow figure from slide to cell clusters (INF, TUM, STR) and MEMs and then patient clusters is added. This information can be found in various figures (eg Fig 1a,b, Fig 6a etc) but a consolidated figure (could be supplemental) will greatly improve clarity and reproducibility.

Reviewer #2 (Remarks to the Author):

All my reviewer comments has been addressed satisfactory.

Reviewer #3 (Remarks to the Author):

Firstly, it would be beneficial for the authors to provide explicit details about the method used to derive the PDOs (Patient-Derived Organoids). Specifically, it would be helpful to know if the PDOs were passaged before exposure to drugs and the duration of their cultivation in culture. While the authors mention the utilization of the method described by Sachs et al for deriving the organoids, they later refer to the use of agarose as a matrix instead of Cultrex® BME, which was originally described in Sachs et al. If an improved version of the Sachs et al method was employed, it would be essential for the authors to explicitly mention and elaborate on these modifications for clarity.

Additionally, a more thorough description of the method employed for PDO viability assays is warranted. Clear information regarding the experimental design, duration of the assay, and any specific conditions or controls used would greatly enhance the reproducibility and understanding of the results.

Regarding the TUM0 and TUM5 subtypes, which predominantly consist of Luminal A and Luminal B cells and are generally associated with HR+ positive cells, it appears counterintuitive that they exhibit the worst hazard ratios. This contradiction within the context of breast cancer research needs to be addressed and clarified. Is it possible that the abundance of TUM5 and TUM0 cells in HR+ samples correlates with a worse prognosis? Moreover, TUM3, which is associated with luminal subtypes, demonstrates a neutral hazard ratio, while TUM1, with the best hazard ratio, exhibits a negative correlation with luminal subtypes. These findings require further discussion and supporting data to elucidate the potential explanations behind these seemingly contradictory results. It would be valuable to explore if the authors have any hypotheses or explanations for these observations.

Similar concerns arise regarding the Ecotypes, where Ecotype 2, characterized by the most adverse subtype of breast cancer (TNBC and HER2), surprisingly demonstrates the best prognosis, while Ecotype 1, predominantly composed of HR+HER2- subtypes, exhibits the worst prognosis. These observations also require thorough discussion and data analysis to better comprehend the reasons behind these counterintuitive results. Perhaps an explanation could be that HR+ with Ecotype 1 have a worse prognosis than HR+ with other ecotype and that TNBC with Ecotype 2 have a better prognosis than TNBC with other ecotypes?. If such insights are available, it would be beneficial to present them in the manuscript.

Reviewer #1 (Remarks to the Author): Expert in artificial intelligence, digital pathology, and computational genomics

I would like to thank the authors for including more data and performing comprehensive analysis to address all reviewers' comments. I have two additional suggestions that will need authors' attention.

Comment 1. With respect to this reviewer's initial comment regarding the comparison with deep features, this reviewer appreciates the extra analyses that have been performed. These results show that deep features did not yield to satisfactory results (not even partially interesting results). I believe this observation needs to be further investigated. What was the patch size and magnification that was used and was the patch size (spot size as per authors' naming) the same for both MEM calculation and deep features? Why did the authors pick 9 clusters? Can the authors do the same analysis task with KimiaNet features? In case of negative results, a discussion touching on the possible reasons would be beneficial.

Response: Thanks for your constructive suggestions. We have redone the analysis comparing our sc-MTOP method and the patch-based deep learning method following your recommendations. We examined two deep learning models: a) self-supervised learning model by Ciga et al.¹, b) KimiaNet by Riasatian et al.². For deep feature extraction, we used the same image patches (termed "spots" in our method) as those used in our original MEM analysis, keeping patch size, coordinates and magnification consistent. We performed K-means clustering on the patches' deep features and set the cluster number to 9, matching the same as the number of categories of our sc-MTOP-based MEM classification. The yielded self-supervised learning model-based MEM (sslMEM) classification and KimiaNet-based MEM (KimiaMEM) classification were compared with our sc-MTOP-based MEM classification in terms of the ability to characterize the locoregional cellular structures, congruency with histological patterns and the prognostic value (Fig. R1a).

Fig. R1. Comparison between the deep learning-based method and our sc-MTOP-based method in characterizing the breast cancer ecosystem and providing clinically relevant information.

- a. Diagram for the comparison between the deep learning-based method and our sc-MTOP-based method.
 - b. The intra- and inter-MEM variations of the tumor, stroma and inflammatory cell proportions for the sc-MTOP-based MEMs, sslMEMs and KimiaMEMs. A random nine-class classification was used as a negative control for this analysis.
 - c. Thumbnail of a whole slide image with manually histological annotation, sc-MTOP-based MEM mapping, sslMEM mapping and KimiaMEM mapping. In histological annotation, a pathologist manually delineated the regions of different histological patterns including two types of tumor regions (red and purple), stroma (blue), immune cell aggregation in stroma (green) and immune infiltration in tumor (yellow). Scale bar: 4 mm.
 - d. MEM composition of different histological regions according to the sc-MTOP-based MEM classification, sslMEM classification and KimiaMEM classification.
 - e. Forest plot showing the univariate Cox regression analysis of relapse-free survival for sc-MTOP-based MEMs, sslMEMs and KimiaMEMs modeled as log-transformed module percentages. Squares and whiskers represent point estimates and the 95% confidence interval of hazard ratios.
- Abbreviations: WSI, whole slide image; sc-MTOP, single-cell morphological and topological profiling; MEM, micro-ecological module; sslMEM, self-supervised learning model-based micro-ecological module; KimiaMEM, KimiaNet-based micro-ecological module; RFS, recurrence-free survival.

a) sslMEMs and KimiaMEMs can characterize the local cellular structures to some extent, but underperformed our sc-MTOP-based MEMs.

We continued to use the inter-MEM and intra-MEM variations (calculation methods described in Revised Manuscript, page 35, line 4-19) to evaluate the ability of sc-MTOP-based MEMs, sslMEMs and KimiaMEMs in characterizing the local multicellular structures in the tumor ecosystem. An ideal classification should have both lower intra-MEM variation, indicating they can identify and summarize the recurrent patterns of local multicellular structures, and higher inter-MEM variation, indicating they can clearly distinguish different local multicellular structures. A random nine-class classification was used as a negative control for this analysis.

The results demonstrated that the sslMEM and KimiaMEM classifications exhibited higher inter-MEM variations and lower intra-MEM variations compared to a random classification, suggesting that they can inform the locoregional cellular compositions to some extent, but their performance did not attain that of our sc-MTOP-based MEM classification (Fig. R1b).

b) sslMEMs and KimiaMEMs showed less congruency with histological regions than our sc-MTOP-based MEMs

Our previous analysis demonstrated high consistency between our sc-MTOP-based MEMs and histological patterns (Fig. 6e-g). We performed the same analysis for sslMEMs and KimiaMEMs.

It was shown that both of them displayed some congruency with histological patterns, such as the tumor region 1 corresponding to the sslModule1 and KimiaModule1 (Fig. R1c-d). However, there was no specific sslMEM or KimiaMEM corresponding clearly to the regions of immune aggregates or tumor & immune, which corresponded to Module8_IA and Module7_TSI of our sc-MTOP-based MEMs respectively (Fig. R1c-d).

c) Only sc-MTOP-based MEMs were prognostic relevant.

In sc-MTOP-based MEMs, high percentage of Module8_IA was significantly associated with a better prognosis, while high percentage of Module4_TS was significantly associated with a worse one. By comparison, no sslMEMs or KimiaMEMs showed a significant association with patient prognosis (Fig. R1e).

We have revised our manuscript to incorporate these data (Revised Manuscript, page 15, line 9-24; Supplementary Fig. S10) and have discussed them in Discussion of our Revised Manuscript (page 23, line 3-15):

We identified nine recurrent MEMs based on the spatial relationship of different cell clusters. These MEMs represented distinct local multicellular structures and showed good congruency with histological patterns. We also tried to use the pre-trained deep learning models to extract features of local image patches and identify deep feature-based MEMs through unsupervised clustering. Although the models were developed under weakly supervised or unsupervised settings, their generated feature-based MEMs can also indicate the cellular composition of patches and showed partial congruency with the histological patterns. This reflected the good robustness and generalizability of these deep learning models. However, without specific labels for supervised learning, convolutional neural network models tend to focus more on image texture features rather than the shape or topological features³. This may explain why they underperformed our sc-MTOP-based method in characterizing the local multicellular structures.

Comment 2.1. With regards to the identified cell clusters (e.g., inflammatory cell clusters (INF0-9)), in Figure 2c, the authors have shown representative patches containing various INF clusters. What is missing from the manuscript is the connection between cell features and patches. How are these patches selected? For example, Fig 2c INF0 patch, did the authors select an area that was enriched with such cell types? Or, are they clustering the patches based on cell representations as well? This needs to be super clear in the manuscript.

Response: Our study used the sc-MTOP data to characterize the diversity of breast cancer ecosystem at three levels: single-cell level (cell clusters), locoregional level (micro-ecological modules), and the whole slide level (ecotypes). The analysis at the cell cluster level did not involve the image patches or the local enrichment of cell clusters. The local visualizations for the cell clusters in figures such as Fig. 2c do not involve patch selection. Instead, they present a representative cell and the corresponding nuclear graph to provide readers with intuitive understanding of the features of this cell cluster.

Comment 2.2. Furthermore, the clarity of the whole analysis process can be substantially enhanced if a workflow figure from slide to cell clusters (INF, TUM, STR) and MEMs and then patient clusters is added. This information can be found in various figures (eg Fig 1 a,b, Fig 6a etc) but a consolidated figure (could be supplemental) will greatly improve clarity and reproducibility.

Response: Thanks for your suggestions. We have presented a workflow diagram illustrating our whole analysis process in Fig. 8 and provided a summarized description of this process in Discussion of our Revised Manuscript (page 21, line 5-18).

Fig. R2 (also see Fig. 8). Summary of the present study.

Left panel: The analytic workflow of generating and using sc-MTOP data to dissect the ecosystem diversity of human breast cancer at multiple levels. Right panel: The ecosystem features with therapeutic and prognostic implications.

Abbreviations: sc-MTOP, single-cell morphological and topological profiling; MEM, micro-ecological module; AIC score, aggregated inflammatory cell abundance score; MITH, morphological intratumor heterogeneity.

Reviewer #3 (Remarks to the Author): Expert in breast cancer tumour microenvironment, digital pathology, and single-cell and spatial omics

Comment 1.1 Firstly, it would be beneficial for the authors to provide explicit details about the method used to derive the PDOs (Patient-Derived Organoids). Specifically, it would be helpful to know if the PDOs were passaged before exposure to drugs and the duration of their cultivation in culture.

Response: To maintain the biological traits of the primary tumor in the PDOs, we did not perform passaging of the PDOs before drug treatment. PDOs were cultured for 5-7 days in the growth medium before drug treatment. We have provided the detailed methods to derive the PDOs in our Revised Manuscript (page 32, line 19-28):

PDO collection and culture

PDOs were established from the breast tumor surgical specimens according to the protocol in the previous study⁴. Breast cancer tissue was cut into 1-3 mm³ pieces and was digested in collagenase (Sigma). The digested tissue suspension was sequentially sheared and strained over 100 µm filter. The strained suspension was centrifuged, suspended and seeded in 24-well plates with breast

cancer organoid medium containing cold basement membrane extract (BME) (prepared according to Sachs et al.⁴). Plates were incubated at 37°C/5% CO₂ in ambient O₂ conditions. Organoids in good condition were split, strained over 70 µm filter, and cultured for 5-7 days in the growth medium before drug treatment.

Comment 1.2 While the authors mention the utilization of the method described by Sachs et al for deriving the organoids, they later refer to the use of agarose as a matrix instead of Cultrex® BME, which was originally described in Sachs et al. If an improved version of the Sachs et al method was employed, it would be essential for the authors to explicitly mention and elaborate on these modifications for clarity.

Response: The agarose (mentioned in Supplementary Fig. S9a) was only used to wrap the PDOs prior to formalin fixation, paraffin embedding and making the PDO slides for the MITH consistency analysis between primary tumors and PDOs. It was not used for deriving or culturing PDOs. We have revised Supplementary Fig. S9a to avoid this misunderstanding (Fig. R3, also see Supplementary Fig. S9a).

Fig. R3 (also see Supplementary Fig. S9a). Diagram of examining the consistency between the primary tumors and the corresponding PDOs in genomic alterations and MITH.

Related to Reviewer #3 Comment 1.2. Changes are marked in red.

Abbreviations: PDO, patient-derived organoids; MITH, morphological intratumor heterogeneity.

Comment 2. Additionally, a more thorough description of the method employed for PDO viability assays is warranted. Clear information regarding the experimental design, duration of the assay,

and any specific conditions or controls used would greatly enhance the reproducibility and understanding of the results.

Response: We have provided the detailed methods for PDO viability assays including the experimental procedure, conditions, duration of the assay and controls (page 33, line 2-10):

Drug response testing of PDOs

For drug response testing, organoids were harvested and diluted to 75 organoids/ μL in the medium. 384-well plates were coated with 10 μL BME per well using a multidrop dispenser. 30 μL of the organoid suspension was then added to each well. Abemaciclib (1 μM final concentration, S5716, Selleck), PF-06873600 (0.4 μM final concentration, S8816, Selleck) and DMSO controls were then added in duplicate. After 7 days of drug treatment, cell viability was measured by adding 40 μL of CellTiter-Glo 3D Reagent (G9683, Promega) per well and reading luminescence after 5 minutes of shaking and 25 minutes of incubation in darkness at room temperature.

Comment 4. Similar concerns arise regarding the Ecotypes, where Ecotype 2, characterized by the most adverse subtype of breast cancer (TNBC and HER2), surprisingly demonstrates the best prognosis, while Ecotype 1, predominantly composed of HR+HER2- subtypes, exhibits the worst prognosis. These observations also require thorough discussion and data analysis to better comprehend the reasons behind these counterintuitive results. Perhaps an explanation could be that HR+ with Ecotype 1 have a worse prognosis than HR+ with other ecotype and that TNBC with Ecotype 2 have a better prognosis than TNBC with other ecotypes?. If such insights are available, it would be beneficial to present them in the manuscript.

Response: Thanks very much for your insightful comments. To facilitate your understanding of our response to your Comment 3, we kindly request addressing Comment 4 here first.

We analyzed the prognostic relevance of ecotype classification in the context of breast cancer subtypes and the results were highly consistent with your speculation. Within HR+ group, the

prognosis of cases with Ecotype 1 was significantly worse than those with other ecotypes (Fig. R4a). Within HR- group, the prognosis of cases with Ecotype 2 was significantly better than those with other ecotypes (Fig. R4b). These subgroup analysis data reasonably explained why Ecotype 1 had more HR+ cases but demonstrated poorer prognosis, and why Ecotype 2 had more HR- cases but showed better prognosis. We have incorporated these data in our Revised Manuscript (page 17, line 10-13; Supplementary Fig. 12d).

Fig. R4. Subgroup analysis for the prognostic relevance of ecotype classification according to the HR status.
a. Kaplan-Meier curves of RFS of HR+ cases with Ecotype 1 and HR+ cases with other ecotypes.
b. Kaplan-Meier curves of RFS of HR- cases with Ecotype 2 and HR- cases with other ecotypes.
Abbreviations: RFS, recurrence-free survival.

Comment 3. Regarding the TUM0 and TUM5 subtypes, which predominantly consist of Luminal A and Luminal B cells and are generally associated with HR+ positive cells, it appears counterintuitive that they exhibit the worst hazard ratios. This contradiction within the context of breast cancer research needs to be addressed and clarified. Is it possible that the abundance of TUM5 and TUM0 cells in HR+ samples correlates with a worse prognosis? Moreover, TUM3, which is associated with luminal subtypes, demonstrates a neutral hazard ratio, while TUM1, with the best hazard ratio, exhibits a negative correlation with luminal subtypes. These findings require further discussion and supporting data to elucidate the potential explanations behind these

seemingly contradictory results. It would be valuable to explore if the authors have any hypotheses or explanations for these observations.

Response: In line with the ecotypes, the prognostic relevance of cell clusters was also not only determined by their associations with breast cancer subtypes. TUM0 and TUM5 cells, which were enriched in Ecotype 1 breast cancers, were associated with a worse prognosis in HR+ cases (Fig. R5a-b). TUM1 cells, enriched in Ecotype 2 breast cancers, was associated with a better prognosis in HR- cases (Fig. R5a, R5c). These data explained why TUM0 and TUM5 cells was enriched in HR+ cases but associated with a poorer prognosis, and why TUM1 cells was enriched in HR- cases but associated with a better prognosis.

Fig. R5. Data related to Reviewer 3 Comment 3.

a. Cell cluster profiles according to the breast cancer ecotypes (also see Supplementary Fig. S11c).

b-c. Forest plot showing the univariate Cox regression analysis of relapse-free survival for tumor cell clusters modeled as log-transformed proportion data in the b) HR+ group and c) HR- group. Squares and whiskers represent point estimates and the 95% confidence interval of hazard ratios.

Reference

1. Ciga, O., Xu, T. & Martel, A.L. Self supervised contrastive learning for digital histopathology. *Machine Learning with Applications* **7**, 100198 (2022).
2. Riasatian, A., *et al.* Fine-Tuning and training of densenet for histopathology image representation using TCGA diagnostic slides. *Med Image Anal* **70**, 102032 (2021).
3. Hermann, K., Chen, T. & Kornblith, S. The origins and prevalence of texture bias in convolutional neural networks. *Advances in Neural Information Processing Systems* **33**, 19000-19015 (2020).
4. Sachs, N., *et al.* A Living Biobank of Breast Cancer Organoids Captures Disease Heterogeneity. *Cell* **172**, 373-386.e310 (2018).

REVIEWERS' COMMENTS

Reviewer #1 (Remarks to the Author):

I would like to thank the authors for addressing all my comments. Congratulations on the great work.

Reviewer #3 (Remarks to the Author):

The authors have addressed all of my comments throughout the review process. I hope that some of the comments have helped the authors. I would like to thank them for the thorough evaluation of my comments and suggestions.

Reviewer #1 (Remarks to the Author): Expert in artificial intelligence, digital pathology, and computational genomics

I would like to thank the authors for addressing all my comments. Congratulations on the great work.

Response: We sincerely appreciate your positive feedback and recognition of our work. Thank you for your valuable comments and suggestions to help improve our manuscript through the review process.

Reviewer #3 (Remarks to the Author): Expert in breast cancer tumour microenvironment, digital pathology, and single-cell and spatial omics

The authors have addressed all of my comments throughout the review process. I hope that some of the comments have helped the authors. I would like to thank them for the thorough evaluation of my comments and suggestions.

Response: We sincerely thank the reviewers for reviewing our work. Your comments have been of important value in improving our research and manuscript.